# This Too Shall Pass: Removing Stale Observations in Dynamic Bayesian Optimization

**Anthony Bardou**
IC, EPFL
Lausanne, Switzerland
anthony.bardou@epfl.ch

**Patrick Thiran**
IC, EPFL
Lausanne, Switzerland
patrick.thiran@epfl.ch

**Giovanni Ranieri**
IC, EPFL
Lausanne, Switzerland
giovanni.ranieri@epfl.ch

## Abstract

Bayesian Optimization (BO) has proven to be very successful at optimizing a static, noisy, costly-to-evaluate black-box function $f : \mathcal{S} \to \mathbb{R}$. However, optimizing a black-box which is also a function of time (*i.e.*, a *dynamic* function) $f : \mathcal{S} \times \mathcal{T} \to \mathbb{R}$ remains a challenge, since a dynamic Bayesian Optimization (DBO) algorithm has to keep track of the optimum over time. This changes the nature of the optimization problem in at least three aspects: (i) querying an arbitrary point in $\mathcal{S} \times \mathcal{T}$ is impossible, (ii) past observations become less and less relevant for keeping track of the optimum as time goes by and (iii) the DBO algorithm must have a high sampling frequency so it can collect enough relevant observations to keep track of the optimum through time. In this paper, we design a Wasserstein distance-based criterion able to quantify the relevancy of an observation with respect to future predictions. Then, we leverage this criterion to build W-DBO, a DBO algorithm able to remove irrelevant observations from its dataset on the fly, thus maintaining simultaneously a good predictive performance and a high sampling frequency, even in continuous-time optimization tasks with unknown horizon. Numerical experiments establish the superiority of W-DBO, which outperforms state-of-the-art methods by a comfortable margin.

## 1 Introduction

Many real-world problems require the optimization of a costly-to-evaluate objective function $f : \mathcal{S} \subseteq \mathbb{R}^d \to \mathbb{R}$ with an unknown closed form (*i.e.*, either the closed form expression of $f$ exists but remains unknown to the user, or it does not exist). Such a setting occurs frequently, and examples can be found in hyperparameters tuning [1], networking [2, 3], robotics [4] or computational biology [5]. In such applications, $f$ can be seen as a black-box and cannot be optimized by usual first-order approaches. Bayesian Optimization (BO) is an effective framework for black-box optimization. Its core idea is to leverage a surrogate model, usually a Gaussian Process (GP), to query $f$ at specific inputs. By doing so, a BO algorithm is able to simultaneously discover and optimize the objective function.

Since its inception, BO has proven to be very effective at optimizing black-boxes in a variety of contexts, such as high-dimensional input spaces [6, 7, 8], batch mode [9] or multi-objective optimization [10]. However, few works study BO in dynamic contexts (*i.e.*, with a time-varying objective function), despite its critical importance. Indeed, dynamic black-box optimization problems arise whenever an optimization task is conducted within an environment that comprises exogenous factors that may vary with time and significantly impact the objective function. Dynamic black-boxes are found in network management [11], unmanned aerial vehicles tasks [12], hyperparameter tuning in online deep learning [13], online clustering [14] or crossing waypoints location in air routes [15].

In a dynamic context, $f : \mathcal{S} \times \mathcal{T} \to \mathbb{R}$ is a time-varying, black-box, costly-to-evaluate objective function with spatial domain $\mathcal{S} \subseteq \mathbb{R}^d$ and temporal domain $\mathcal{T} \subseteq \mathbb{R}$. Unlike common black-box

objective functions that take as input a point $\boldsymbol{x} \in \mathcal{S}$ in a spatial domain only, the function $f$ takes as input a point $(\boldsymbol{x}, t) \in \mathcal{S} \times \mathcal{T}$ in space and time. Since the problem of optimizing $f$ is still addressed under the BO framework, the framework is called *dynamic Bayesian optimization (DBO)*.

Taking time into account does not boil down to merely adding an extra dimension to $\mathcal{S}$. It changes the nature of the optimization problem on at least three aspects: (i) at present time $t_0$, a DBO algorithm can query any arbitrary point $\boldsymbol{x} \in \mathcal{S}$ but cannot query past (*i.e.*, $t < t_0$) nor future[1] (*i.e.*, $t > t_0$) points in time, (ii) as time (only) moves forward, a previously collected observation $((\boldsymbol{x}, t), f(\boldsymbol{x}, t))$ becomes less and less informative about the future values of $f$ as time goes by and (iii) the *response time* (*i.e.*, the time required for hyperparameters inference and acquisition function maximization) becomes a key feature of the DBO algorithm since it has a direct impact on the sampling frequency of the algorithm and, consequently, on its ability to track the position of the optimum as $f$ changes.

Interestingly, (ii) and (iii) imply that each observation eventually becomes stale and, as such, a computational burden if kept in the dataset. Since a DBO task might require the optimization of an objective function over arbitrarily long periods of time, the ability to remove observations from the dataset on the fly, as soon as they become irrelevant for future predictions, is essential to prevent a prohibitive growth of the response time of DBO algorithms. Overall, (i), (ii) and (iii) require to address dynamic (*i.e.*, space-time) problems differently from usual (*i.e.*, space-only) problems.

The main contributions of this article are twofold. First, we propose a fast and efficient method able to quantify the relevancy of an observation. Second, we leverage this method to build a DBO algorithm able to identify and delete irrelevant observations in an online fashion[2].

## 2 Background

Let us start by describing the BO framework with a GP prior, as introduced by [16]. Given an objective function $f : \mathcal{S} \subseteq \mathbb{R}^d \to \mathbb{R}$, BO assumes that $f$ is a $\mathcal{GP}(0, k(\boldsymbol{x}, \boldsymbol{x}'))$. For any $\boldsymbol{x} \in \mathcal{S}$, and given a dataset of observations $\mathcal{D} = \{(\boldsymbol{x}_i, y_i)\}_{i \in [\![1, n]\!]}$, where $\boldsymbol{x}_i \in \mathcal{S}$ is a previously queried input with (noisy) function value $y_i = f(\boldsymbol{x}_i) + \epsilon$, $\epsilon \sim \mathcal{N}(0, \sigma^2)$, the posterior distribution of $f(\boldsymbol{x})$ is $\mathcal{N}(\mu(\boldsymbol{x}), \sigma^2(\boldsymbol{x}))$, where

$$\mu(\boldsymbol{x}) = \boldsymbol{k}^\top(\boldsymbol{x}, \boldsymbol{X})\boldsymbol{\Delta}^{-1}\boldsymbol{y}, \tag{1}$$

$$\sigma^2(\boldsymbol{x}) = k(\boldsymbol{x}, \boldsymbol{x}) - \boldsymbol{k}^\top(\boldsymbol{x}, \boldsymbol{X})\boldsymbol{\Delta}^{-1}\boldsymbol{k}(\boldsymbol{x}, \boldsymbol{X}) \tag{2}$$

with $\boldsymbol{X} = (\boldsymbol{x}_1, \cdots, \boldsymbol{x}_n)$, $\boldsymbol{y} = (y_1, \cdots, y_n)$, $\boldsymbol{\Delta} = \boldsymbol{k}(\boldsymbol{X}, \boldsymbol{X}) + \sigma^2 \boldsymbol{I}$ and $\boldsymbol{k}(\mathcal{X}, \mathcal{Y}) = (k(\boldsymbol{x}_i, \boldsymbol{x}_j))_{\boldsymbol{x}_i \in \mathcal{X}, \boldsymbol{x}_j \in \mathcal{Y}}$.

To find $\boldsymbol{x}_{n+1}$, the input to query at the $(n+1)$th iteration, a BO algorithm exploits an acquisition function $\varphi : \mathcal{S} \to \mathbb{R}$. The acquisition function $\varphi$ quantifies the benefits of querying the input $\boldsymbol{x}$ in terms of (i) exploration (*i.e.*, how much it improves the GP regression of $f$) and (ii) exploitation (*i.e.*, how close it is to the optimum of $f$ according to the GP). A large variety of BO acquisition functions have been proposed, such as GP-UCB [17], Expected Improvement [18] or Probability of Improvement [19]. Formally, the BO algorithm determines its next queried input by finding $\boldsymbol{x}_{n+1} = \arg\max_{\boldsymbol{x} \in \mathcal{S}} \varphi(\boldsymbol{x})$.

BO extends naturally to dynamic problems, by adapting the covariance function $k$ to properly capture temporal correlations (discussed later in this section). The resulting inference formulas are very similar to (1) and (2). Given a dataset of observations $\mathcal{D} = \{((\boldsymbol{x}_i, t_i), y_i)\}_{i \in [\![1, n]\!]}$, where $(\boldsymbol{x}_i, t_i) \in \mathcal{S} \times \mathcal{T}$ is a previously queried input with the (noisy) function value $y_i = f(\boldsymbol{x}_i, t_i) + \epsilon$, the posterior distribution of $f(\boldsymbol{x}, t)$ is $\mathcal{N}(\mu(\boldsymbol{x}, t), \sigma^2(\boldsymbol{x}, t))$ for any $(\boldsymbol{x}, t) \in \mathcal{S} \times \mathcal{T}$, with

$$\mu(\boldsymbol{x}, t) = \boldsymbol{k}^\top((\boldsymbol{x}, t), \boldsymbol{X})\boldsymbol{\Delta}^{-1}\boldsymbol{y}, \tag{3}$$

$$\sigma^2(\boldsymbol{x}, t) = k((\boldsymbol{x}, t), (\boldsymbol{x}, t)) - \boldsymbol{k}^\top((\boldsymbol{x}, t), \boldsymbol{X})\boldsymbol{\Delta}^{-1}\boldsymbol{k}((\boldsymbol{x}, t), \boldsymbol{X}) \tag{4}$$

with $\boldsymbol{X} = ((\boldsymbol{x}_1, t_1), \cdots, (\boldsymbol{x}_n, t_n))$, $\boldsymbol{y} = (y_1, \cdots, y_n)$, $\boldsymbol{\Delta} = \boldsymbol{k}(\boldsymbol{X}, \boldsymbol{X}) + \sigma^2 \boldsymbol{I}$ and $\boldsymbol{k}(\mathcal{X}, \mathcal{Y}) = (k((\boldsymbol{x}_i, t_i), (\boldsymbol{x}_j, t_j)))_{(\boldsymbol{x}_i, t_i) \in \mathcal{X}, (\boldsymbol{x}_j, t_j) \in \mathcal{Y}}$.

---

[1]At least, not immediately.

[2]Its Python documented implementation can be found at `https://github.com/WDBO-ALGORITHM/wdbo_algo`. A PyPI package can be quickly installed with the command `pip install wdbo-algo`.

Exploiting the usual acquisition functions in a dynamic context is also straightforward. Since a DBO algorithm can only query an input at the current running time $t_0$, the next queried input is simply $(\boldsymbol{x}_{n+1}, t_0)$, with $\boldsymbol{x}_{n+1} = \arg\max_{\boldsymbol{x} \in \mathcal{S}} \varphi(\boldsymbol{x}, t_0)$. Some DBO algorithms (e.g., [20]) extend the querying horizon to the near future, that is, from $t_0$ to a time interval $[t_0, t_0 + \delta_t]$. In that case, the next queried input is $(\boldsymbol{x}_{n+1}, t_{n+1}) = \arg\max_{(\boldsymbol{x}, t) \in \mathcal{S} \times [t_0, t_0 + \delta_t]} \varphi(\boldsymbol{x}, t)$.

BO is an active field of research, but relatively few works address DBO, despite the natural extension of BO to dynamic problems described above. We conclude this section by reviewing them. In [21], the objective function is allowed to evolve in time according to a simple Markov model, controlled by a hyperparameter $\epsilon \in [0, 1]$. On the one hand, the authors propose R-GP-UCB, which handles data staleness by resetting the dataset every $N(\epsilon)$ iterations. On the other hand, the authors also propose TV-GP-UCB that incorporates data staleness by weighing the covariance of two queries $\boldsymbol{q}_i = (\boldsymbol{x}_i, t_i)$ and $\boldsymbol{q}_j = (\boldsymbol{x}_j, t_j)$ by $(1 - \epsilon)^{|i-j|/2}$. In [22], the authors use the same model with an event-triggered reset of the dataset. Although less relevant to this work, let us mention [23] and [24] for the sake of completeness. Under frequentist assumptions, they also propose DBO algorithms that forget irrelevant observations by either resetting their datasets or by using decreasing covariance weights. However, they assume that the variational budget of the objective function is fixed, which has the drawback of requiring the objective function to become asymptotically static. This is a very different setting than the one of interest in this paper, which does not make this requirement.

The aforementioned algorithms all work with discrete, evenly-spaced time steps. This setting simplifies the regret analysis of DBO algorithms through the use of proof techniques similar to the ones used for static BO. However, it also overlooks a critical effect of the response times of their algorithms. In fact, the response time of a BO algorithm heavily depends on its dataset size $n$, since BO inference is in $\mathcal{O}(n^3)$. Although it is reasonable to ignore this for classical BO because the objective function $f$ is static, DBO algorithms cannot make this simplification as it directly impacts their ability to track the optimal argument of the objective function through time. Many algorithms (e.g., see [21, 23, 24]) recommend to keep all the collected observations in their datasets, whereas in practice, their response times would asymptotically become prohibitive. Other algorithms (e.g., see [21, 22, 23]) propose to reset their datasets, either periodically or once an event is triggered. This probably deletes some relevant observations in the process. More importantly, these algorithms necessarily estimate their covariance function hyperparameters beforehand and keep them fixed during the optimization. This lack of adaptivity of the estimation might lead to severely under-optimal characterization of the function by the hyperparameters, especially when optimizing an ever-changing objective function on an infinite time horizon.

To the best of our knowledge, only one work acknowledges these problems. It proposes ABO [20], an algorithm that uses a decomposable spatio-temporal covariance function $k((\boldsymbol{x}, t), (\boldsymbol{x}', t')) = k_S(||\boldsymbol{x} - \boldsymbol{x}'||_2) k_T(|t - t'|)$ to accurately model complex spatio-temporal correlations and samples the objective function only when deemed necessary. Although this reduces the size of ABO's dataset, ABO does not propose a way to remove stale observations, it only adds new observations less frequently. Therefore, using ABO will still become prohibitive in the long run.

The most relevant methods to quantify the relevancy of an observation can be found in the sparse GPs literature (e.g., see [25, 26]). However, they require non-trivial adjustments to account for the particular nature of the time dimension. As far as we know, there is no method in the DBO literature able to quantify the relevancy of an observation in an online setting. As mentioned before, such a method is much needed as it would allow a DBO algorithm to remove stale data on the fly while preserving the predictive performance of the algorithm. We bridge this gap by providing a sound criterion to measure the relevancy of an observation and an algorithm exploiting this criterion to remove stale data from its dataset.

## 3 A Wasserstein Distance-Based Criterion

### 3.1 Core Assumptions

To address the DBO problem under suitable smoothness conditions, let us make the usual assumption of BO, using a Gaussian Process (GP) as a surrogate model for $f$ (see [16]).

**Assumption 3.1.** $f$ is a $\mathcal{GP}(0, k((\boldsymbol{x}, t), (\boldsymbol{x}', t')))$, whose mean is 0 (without loss of generality) and whose covariance function is denoted by $k : \mathcal{S} \times \mathcal{T} \times \mathcal{S} \times \mathcal{T} \to \mathbb{R}_+$.

In order to accurately model complex spatio-temporal dynamics, we make the same assumption on the decomposition and isotropy in time and space of the covariance function $k$ as in [20].

**Assumption 3.2.**

$$k((\boldsymbol{x}, t), (\boldsymbol{x}', t')) = \lambda k_S(||\boldsymbol{x} - \boldsymbol{x}'||_2, l_S) k_T(|t - t'|, l_T), \tag{5}$$

with $\lambda > 0$, $k_S : \mathbb{R}_+ \to [0, 1]$ and $k_T : \mathbb{R}_+ \to [0, 1]$ two positive correlation functions, parameterized by lengthscales $l_S > 0$ and $l_T > 0$, respectively. The factor $\lambda > 0$ scales the product of the two correlation functions and hence, controls the magnitude of the covariance function $k$. The lengthscales $l_S$ and $l_T$ control the correlation lengths of the GP (see [27] for more details) in space and in time, respectively.

Although the covariance function $k$ is able to model temporal correlations with $k_T$, it does not accurately measure the relevancy of an observation. The next section addresses this question.

## 3.2 Measuring the Relevancy of an Observation

By definition, when an irrelevant observation gets removed from the dataset, the GP posterior experiences hardly any change. Therefore, we propose to measure the relevancy of an observation $\boldsymbol{o}_i = ((\boldsymbol{x}_i, t_i), y_i)$ by measuring the impact that the removal of $\boldsymbol{o}_i$ has on the GP posterior.

Let $\mathcal{GP}_{\mathcal{D}}$ be the GP conditioned on the dataset $\mathcal{D} = \{((\boldsymbol{x}_i, t_i), y_i)\}_{i \in [\![1,n]\!]}$, with $(\boldsymbol{x}_i, t_i) \in \mathcal{S} \times \mathcal{T}$ and $y_i = f(\boldsymbol{x}_i, t_i) + \epsilon, \epsilon \sim \mathcal{N}(0, \sigma^2)$. Without loss of generality, let us measure the impact of removing $((\boldsymbol{x}_1, t_1), y_1)$ from the dataset on the GP posterior. Clearly, the measure should be defined on the domain of future predictions at time $t_0$, denoted by $\mathcal{F}_{t_0}$, which must include the whole space $\mathcal{S}$ and only the future time interval $[t_0, +\infty)$:

$$\mathcal{F}_{t_0} = \mathcal{S} \times [t_0, +\infty). \tag{6}$$

We compare a GP conditioned on the whole dataset, denoted by $\mathcal{GP}_{\mathcal{D}}$, with a GP conditioned on $\tilde{\mathcal{D}}$, the dataset without $(\boldsymbol{x}_1, t_1, y_1)$, denoted by $\mathcal{GP}_{\tilde{\mathcal{D}}}$. For an arbitrary point $(\boldsymbol{x}, t) \in \mathcal{F}_{t_0}$, $\mathcal{GP}_{\mathcal{D}}$ provides a posterior distribution $\mathcal{N}_{\mathcal{D}}(\boldsymbol{x}, t) = \mathcal{N}\left(\mu_{\mathcal{D}}(\boldsymbol{x}, t), \sigma_{\mathcal{D}}^2(\boldsymbol{x}, t)\right)$, and so does $\mathcal{GP}_{\tilde{\mathcal{D}}}$ with $\mathcal{N}_{\tilde{\mathcal{D}}}(\boldsymbol{x}, t) = \mathcal{N}\left(\mu_{\tilde{\mathcal{D}}}(\boldsymbol{x}, t), \sigma_{\tilde{\mathcal{D}}}^2(\boldsymbol{x}, t)\right)$. We compare these two distributions by using the 2-Wasserstein distance [28], given by

$$W_2\left(\mathcal{N}_{\mathcal{D}}(\boldsymbol{x}, t), \mathcal{N}_{\tilde{\mathcal{D}}}(\boldsymbol{x}, t)\right) = \left((\mu_{\mathcal{D}}(\boldsymbol{x}, t) - \mu_{\tilde{\mathcal{D}}}(\boldsymbol{x}, t))^2 + (\sigma_{\mathcal{D}}(\boldsymbol{x}, t) - \sigma_{\tilde{\mathcal{D}}}(\boldsymbol{x}, t))^2\right)^{\frac{1}{2}}. \tag{7}$$

A natural extension of the 2-Wasserstein distance from a point $(\boldsymbol{x}, t) \in \mathcal{F}_{t_0}$ to the domain $\mathcal{F}_{t_0}$ is

$$W_2\left(\mathcal{GP}_{\mathcal{D}}, \mathcal{GP}_{\tilde{\mathcal{D}}}\right) = \left(\oint_{\mathcal{S}} \int_{t_0}^{\infty} W_2^2\left(\mathcal{N}_{\mathcal{D}}(\boldsymbol{x}, t), \mathcal{N}_{\tilde{\mathcal{D}}}(\boldsymbol{x}, t)\right) d\boldsymbol{x} dt\right)^{\frac{1}{2}}. \tag{8}$$

Observe that (8) is a criterion that effectively captures the impact of removing the observation $\boldsymbol{o}_1 = ((\boldsymbol{x}_1, t_1), y_1)$ from the dataset on the GP posterior. However, as discussed in Appendix F, the covariance function hyperparameters $\boldsymbol{\theta} = (\lambda, l_S, l_T)$ control the magnitude of (8). This is illustrated by Figure 1, which depicts two couples of GP posteriors that achieve the same value (8). Depending on the lengthscale magnitude, the posteriors may be quite different or, conversely, very similar. As a result, (8) cannot be directly used as a gauge of observation relevancy.

To remove this ambiguity, we normalize (8) by $W_2(\mathcal{GP}_{\mathcal{D}}, \mathcal{GP}_{\emptyset})$ (*i.e.*, the 2-Wasserstein distance between the GP conditioned on $\mathcal{D}$ and the prior GP), and we obtain the ratio

$$R(\mathcal{GP}_{\mathcal{D}}, \mathcal{GP}_{\tilde{\mathcal{D}}}) = \frac{W_2(\mathcal{GP}_{\mathcal{D}}, \mathcal{GP}_{\tilde{\mathcal{D}}})}{W_2(\mathcal{GP}_{\mathcal{D}}, \mathcal{GP}_{\emptyset})}. \tag{9}$$

Intuitively, $W_2(\mathcal{GP}_{\mathcal{D}}, \mathcal{GP}_{\emptyset})$ measures the impact of resetting the whole dataset $\mathcal{D}$ on the GP posterior and serves as a baseline that puts into perspective the distance measured by (8). Technically, taking the ratio (9) successfully cancels out the influence of the covariance function hyperparameters on the magnitude of the Wasserstein distances, as further discussed in Appendix F. As a direct consequence,

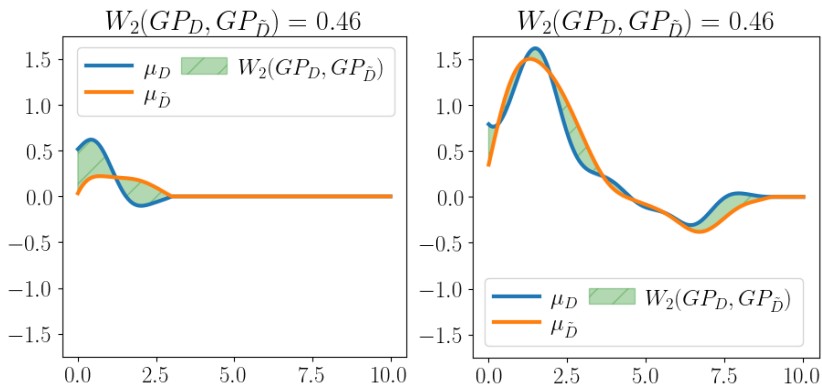

Figure 1: Similar values of Wasserstein distance, different effect on posteriors. For visualization purposes, only the posterior means of two posterior GPs (blue for $\mu_{\mathcal{D}}$ and orange for $\mu_{\tilde{\mathcal{D}}}$) are depicted, along a single dimension (e.g., time). The Wasserstein distance between the two posteriors is shown by the green shaded area. The GPs have a small lengthscale (left) or, conversely, a large lengthscale (right) for the chosen dimension.

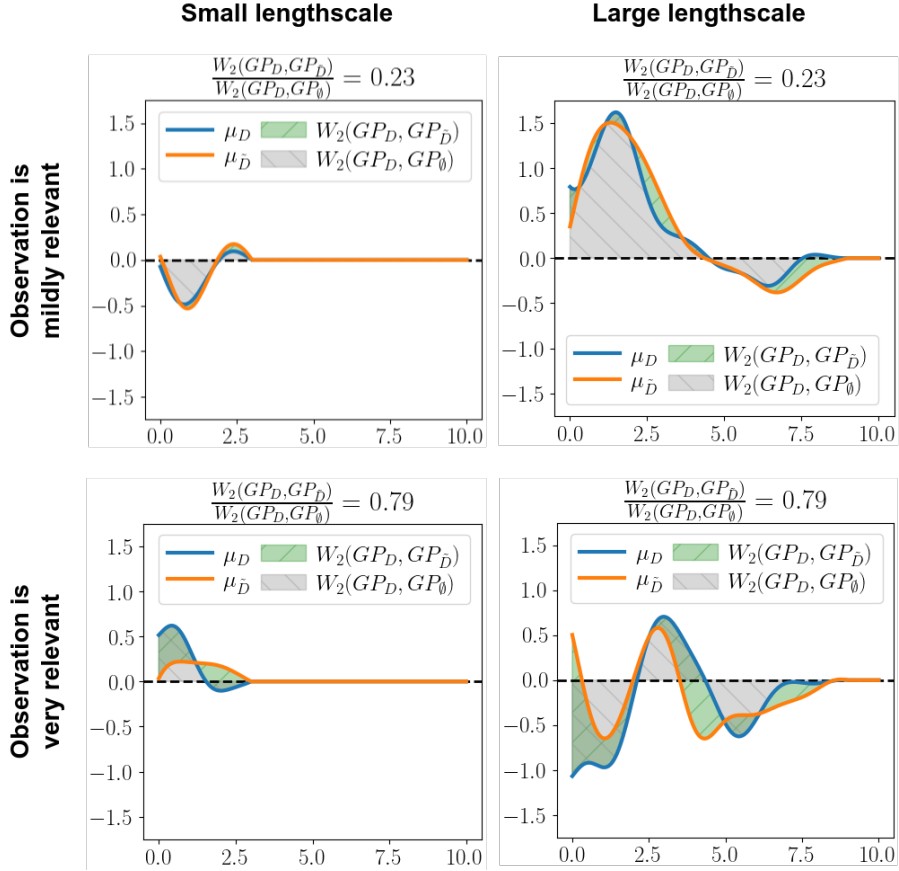

Figure 2: Normalized Wasserstein distances. Similarly to Figure 1, a few couples of GP posterior means $(\mu_{\mathcal{D}}, \mu_{\tilde{\mathcal{D}}})$ are depicted. The top (resp., bottom) row depicts couples of posteriors that yield a small (resp., large) ratio (9). The left (resp., right) column depicts couples of posteriors controlled by a small (resp., large) lengthscale. The prior GP mean $\mu_{\emptyset} = 0$ is shown as a black dashed line, and the Wasserstein distance between the posterior and the prior as a gray shaded area.

(9) is an unambiguous indication of how relevant an observation is. This is illustrated by Figure 2, which depicts couples of GP posteriors under different contexts. When (9) is small (resp., large), the posteriors are similar (resp., dissimilar) regardless of the magnitude of the lengthscale.

Exploiting this criterion is straightforward. When (9) is small, one can infer that the observation $o_1$ can be safely removed from the dataset since it will have virtually no impact on the posterior. Conversely, when (9) is large, one can infer that removing $o_1$ would alter the posterior too much, and conclude that it is a relevant observation that must remain in the dataset. The exploitation of (9) is discussed in more details in Section 4.2.

The criterion (9) is useful for a DBO algorithm if and only if it can be computed on the fly, in an online setting. In the next section, we show that (9) can be approximated efficiently, and we describe a DBO algorithm able to exploit the criterion.

## 4 Using the Criterion in Practice

### 4.1 Computational Tractability

In [29], the authors provide an algorithm to approximate the 2-Wasserstein distance between two GPs up to an arbitrary precision level. However, the computational cost of this algorithm is too expensive in an online setting, where it is crucial to keep the per-iteration cost as small as possible to ensure a high sampling frequency. In this section, we put this issue to rest by deriving an explicit approximation of (9). These formulas are computationally cheap enough to be exploited on the fly.

In Appendix A, we show that (7) can be computed efficiently. Next, in Appendix B, we apply these results to obtain an upper bound of (8). The key observation for deriving this result is to approximate the integrals in (8) by a convolution of the covariance functions with themselves in space and time. The same trick can be used for approximating $W_2(\mathcal{GP}_\mathcal{D}, \mathcal{GP}_\emptyset)$.

**Theorem 4.1.** *Let $t_0$ be the present time and $\mathcal{D} = \{((\boldsymbol{x}_i, t_i), y_i)\}_{i \in [\![1,n]\!]}$ be a dataset of observations made before $t_0$. Let $\tilde{\mathcal{D}} = \{((\boldsymbol{x}_i, t_i), y_i)\}_{i \in [\![2,n]\!]}$ be the dataset without the first observation and $\mathcal{F}_{t_0} = \mathcal{S} \times [t_0, +\infty)$ be the domain of future predictions. Then, an upper bound for $W_2^2(\mathcal{GP}_\mathcal{D}, \mathcal{GP}_{\tilde{\mathcal{D}}})$ on $\mathcal{F}_{t_0}$ is*

$$
\hat{W}_2^2\left(\mathcal{GP}_\mathcal{D}, \mathcal{GP}_{\tilde{\mathcal{D}}}\right) = \lambda^2(a^2 + \boldsymbol{E})C((\boldsymbol{x}_1, t_1),(\boldsymbol{x}_1, t_1)) + \lambda^2(2a\boldsymbol{b} + \boldsymbol{c})C((\boldsymbol{x}_1, t_1), \tilde{\mathcal{D}}) \\
+ \lambda^2 \operatorname{tr}\left(\left(\boldsymbol{b}\boldsymbol{b}^\top + \boldsymbol{M}\right) C(\tilde{\mathcal{D}}, \tilde{\mathcal{D}})\right) \tag{10}
$$

*where $C(\mathcal{X}, \mathcal{Y}) = \left((k_S * k_S)(\boldsymbol{x}_j - \boldsymbol{x}_i) \cdot (k_T * k_T)_{t_0 - t_i}^{+\infty}(t_j - t_i)\right)_{\substack{(\boldsymbol{x}_i, t_i) \in \mathcal{X} \\ (\boldsymbol{x}_j, t_j) \in \mathcal{Y}}}$, where $(f * g)$ denotes the convolution between $f$ and $g$, and $(f * g)_a^b$ denotes the convolution between $f$ and $g$ restricted to the interval $[a, b]$. The terms $a$, $\boldsymbol{b}$, $\boldsymbol{c}$, $\boldsymbol{E}$ and $\boldsymbol{M}$ are explicited in Appendices A and B.*

*Moreover, an upper bound for $W_2^2(\mathcal{GP}_\mathcal{D}, \mathcal{GP}_\emptyset)$ on $\mathcal{F}_{t_0}$ is*

$$
\hat{W}_2^2\left(\mathcal{GP}_\mathcal{D}, \mathcal{GP}_\emptyset\right) = \lambda^2\left(\boldsymbol{y}^\top \boldsymbol{\Delta}^{-\top} C\left(\mathcal{D}, \mathcal{D}\right) \boldsymbol{\Delta}^{-1} \boldsymbol{y} + \operatorname{tr}\left(\boldsymbol{\Delta}^{-1} C\left(\mathcal{D}, \mathcal{D}\right)\right)\right). \tag{11}
$$

This theorem provides the analytic form of an upper bound for the Wasserstein distance between $\mathcal{GP}_\mathcal{D}$ and $\mathcal{GP}_{\tilde{\mathcal{D}}}$ and the Wasserstein distance between $\mathcal{GP}_\mathcal{D}$ and $\mathcal{GP}_\emptyset$ on the domain of future predictions $\mathcal{F}_{t_0}$. Using it, we can compute an approximation $\hat{R}$ of the relative criterion (9), that is

$$
\hat{R}(\mathcal{GP}_\mathcal{D}, \mathcal{GP}_{\tilde{\mathcal{D}}}) = \frac{\hat{W}_2(\mathcal{GP}_\mathcal{D}, \mathcal{GP}_{\tilde{\mathcal{D}}})}{\hat{W}_2(\mathcal{GP}_\mathcal{D}, \mathcal{GP}_\emptyset)}. \tag{12}
$$

In Appendix C, we study the error between the criterion (9) and the approximation (12). In essence, we bound the approximation error caused by estimating the integrals in (8) by a self-convolution of the covariance functions $k_S$ and $k_T$ (i.e., $k_S * k_S$ and $k_T * k_T$). Furthermore, we provide numerical evidence that the approximation errors in the numerator and the denominator of (12) compensate each other at least in part, making (12) a decent approximation for (9).

In practice, the upper bounds (10) and (11) can only be computed efficiently if the convolutions of the covariance functions can themselves be computed efficiently. The analytic forms for the convolution

Table 1: Usual covariance functions. $\Gamma$ is the Gamma function and $K_\nu$ is a modified Bessel function of the second kind of order $\nu$.

| Covariance Function | $k(\boldsymbol{x})$ |
|---|---|
| Squared-Exponential $(l)$ | $e^{-\frac{\lvert\lvert\boldsymbol{x}\rvert\rvert_2^2}{2l^2}}$ |
| Matérn $(\nu, l)$ | $\frac{2^{1-\nu}}{\Gamma(\nu)} \left( \frac{\sqrt{2\nu}\lvert\lvert\boldsymbol{x}\rvert\rvert_2}{l_S} \right)^\nu K_\nu \left( \frac{\sqrt{2\nu}\lvert\lvert\boldsymbol{x}\rvert\rvert_2}{l_S} \right)$ |

of two usual covariance functions listed in Table 1, namely Squared-Exponential (SE) and Matérn [30], are provided in Appendix D together with Tables 3 and 4 that list the self-convolutions for the spatial (resp., temporal) covariance function. Their detailed computations are also provided in this appendix. In a nutshell, the formulas are obtained first in the Fourier domain by computing the square of the spectral densities of the covariance functions, and next by computing their inverse Fourier transform.

Together, Tables 3, 4 in Appendix D and Theorem 4.1 show that the approximation of (9) given by (12) can be computed efficiently in an online setting. In an effort to generalize our results to a class of covariance functions that extends beyond Assumption 3.2, we also discuss how to compute the self-convolution of an anisotropic spatial SE covariance function in Appendix E. We now leverage (12) to propose a DBO algorithm able to pinpoint and remove irrelevant observations in its dataset.

## 4.2 W-DBO

The metric (9) and its approximation (12) can be seen as a relative error (or drift), expressed as a percentage, that separates $\mathcal{GP}_\mathcal{D}$ and $\mathcal{GP}_{\tilde{\mathcal{D}}}$. Indeed, the Wasserstein distance $W\left(\mathcal{GP}_\mathcal{D}, \mathcal{GP}_{\tilde{\mathcal{D}}}\right)$ is scaled by the Wasserstein distance $W\left(\mathcal{GP}_\mathcal{D}, \mathcal{GP}_\emptyset\right)$, that is, the distance between $\mathcal{GP}_\mathcal{D}$ and the prior. In other words, (9) and its approximation (12) measure the relative drift from $\mathcal{GP}_\mathcal{D}$ to $\mathcal{GP}_{\tilde{\mathcal{D}}}$ caused by the removal of one observation. When removing multiple observations, the relative drifts naturally accumulate in a multiplicative way (similarly to the way relative errors accumulate). As a consequence, removing multiple observations could, in the worst case, make $\mathcal{GP}_{\tilde{\mathcal{D}}}$ drift exponentially fast from $\mathcal{GP}_\mathcal{D}$. To keep this exponential drift under control, one can use a removal budget $b(t) = (1 + \alpha)^t$ that allows a maximal relative drift from $\mathcal{GP}_\mathcal{D}$ of $\alpha$ per unit of time (e.g., if $\alpha = 0.1$, the allowed maximal drift is 10 % per unit of time). The cost of removing an observation is given by (12).

Algorithm 1 describes W-DBO, a DBO algorithm exploiting (12) to remove irrelevant observations on the fly. As described above, the removal budget is controlled by a single hyperparameter $\alpha$ and grows exponentially as time goes by (see line 24). At each iteration, (12) is used to compute the relevancy of each observation in the dataset (see lines 10-13). The relevancy of the least relevant observation is then compared to the removal budget, and the observation is removed if the budget allows it (see lines 14-17). This process is repeated until all the budget is consumed. Such a greedy observation removal policy causes W-DBO to overestimate the impact of removing multiple observations[3]. We discuss and motivate the expression of the removal budget in Appendix G. The sensitivity analysis conducted in Section 5.1 supports this removal budget, by showing that the same value of the hyperparameter $\alpha$ is valid for a large set of different objective functions.

Finally, note that using (12) to remove irrelevant observations on the fly can be performed in conjunction with any BO algorithm, because it can be appended at the end of each optimization step as a simple post-processing stage. This agnostic property of W-DBO is supported by the ability of Algorithm 1 to take as inputs any GP model $\mathcal{GP}$ and any acquisition function $\varphi$. Similarly, observe that lines 5-8 in Algorithm 1 describe the usual BO optimization loop, without any modification.

## 5 Numerical Results

In this section, we study the empirical performance of W-DBO. To measure the quality of the queries made by the DBO solutions, we compute the average regret (lower is better). For the sake of realistic evaluation, two iterations of a solution are seperated by its response time (*i.e.*, the time taken to infer its hyperparameters and optimize its acquisition function). Furthermore, all covariance function

---

[3]To prevent this, the criterion (12) could be computed on every element of $2^{\lvert\mathcal{D}\rvert}$ at each iteration. Unfortunately, this policy does not scale well with the dataset size $\lvert\mathcal{D}\rvert$.

**Algorithm 1** W-DBO

1: **Input**: $\mathcal{GP}$, acquisition function $\varphi$, hyperparameter $\alpha$, clock $\mathcal{C}$, initial observations $\mathcal{D}$.
2: Get current time $t$ from clock $\mathcal{C}$
3: $b = 1$
4: **while true do**
5:     Find $\boldsymbol{x}_t = \arg\max_{\boldsymbol{x}} \varphi(\boldsymbol{x}, t)$
6:     Observe $y_t = f(\boldsymbol{x}_t, t) + \epsilon, \epsilon \sim \mathcal{N}(0, \sigma^2)$
7:     $\mathcal{D} = \mathcal{D} \cup \{(\boldsymbol{x}_t, t, y_t)\}$
8:     Condition $\mathcal{GP}$ on $\mathcal{D}$ and get the MLE parameters $(\lambda, l_S, l_T, \hat{\sigma}^2)$
9:     **while** $b > 1$ **do**
10:         **for all** $(\boldsymbol{x}_i, t_i, y_i) \in \mathcal{D}$ [in parallel] **do**
11:             $\tilde{\mathcal{D}} = \mathcal{D} \setminus \{(\boldsymbol{x}_i, t_i, y_i)\}$
12:             Compute $\hat{R}_i$ using (12)
13:         **end for**
14:         $i^* = \arg\min_{i \in [\![1, |\mathcal{D}|]\!]} \hat{R}_i$
15:         **if** $b > 1 + \hat{R}_{i^*}$ **then**
16:             $\mathcal{D} = \mathcal{D} \setminus \{(\boldsymbol{x}_{i^*}, t_{i^*}, y_{i^*})\}$
17:             $b = b/(1 + \hat{R}_{i^*})$
18:         **else**
19:             **break**
20:         **end if**
21:     **end while**
22:     $t' = t$
23:     Get current time $t$ from clock $\mathcal{C}$
24:     $b = b(1 + \alpha)^{(t-t')/l_T}$
25: **end while**

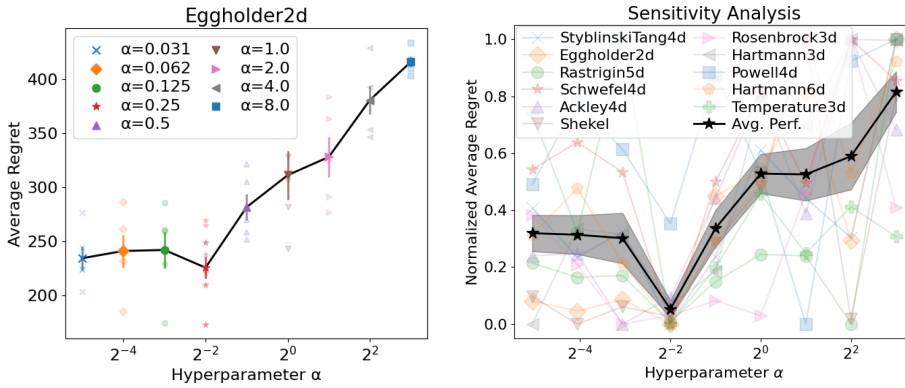

Figure 3: (Left) Sensitivity analysis on the Eggholder function. (Right) Aggregation of sensitivity analyses of W-DBO made on 10 synthetic functions and a real-world experiment. For aggregation purposes, the average regrets in each experiment have been normalized between 0 (lowest average regret) and 1 (largest average regret). The average performance of W-DBO over all the experiments is shown in black. Standard errors are depicted with colored bars (left) and shaded areas (right).

parameters and the noise level are estimated on the fly. Please refer to Appendix H.1 for further experimental details, and to Appendix H.2 for a detailed description of the dynamic benchmarks.

## 5.1 Sensitivity Analysis

We start by studying the impact of the W-DBO hyperparameter $\alpha$ on the average regret. Recall that we take into account the response time of the algorithm. This evaluation protocol reveals that a trade-off must be achieved between having an accurate model of the objective function (which requires a large dataset) and being able to track the optimal argument of the function as it evolves (which requires a high sampling frequency, thus a moderately-sized dataset).

Table 2: Comparison of W-DBO with competing methods. The average regret over 10 independent replications is reported (lower is better). The performance of the best algorithm is written in **bold text**. The performance of algorithms whose confidence intervals overlap the best performing algorithm's confidence interval is underlined.

| Experiment ($d+1$) | GP-UCB | R-GP-UCB | TV-GP-UCB | ET-GP-UCB | ABO | W-DBO |
|---|---|---|---|---|---|---|
| Rastrigin (5) | **17.81** | 53.67 | 26.50 | 19.54 | 36.16 | 18.54 |
| Schwefel (4) | 469.10 | 954.03 | 520.40 | 428.97 | 662.32 | **290.34** |
| StyblinskiTang (4) | 18.83 | 45.82 | 15.74 | 22.16 | 58.40 | **13.04** |
| Eggholder (2) | 542.53 | 273.60 | 287.01 | 559.61 | 256.92 | **225.68** |
| Ackley (4) | 4.10 | 4.45 | 3.27 | 3.96 | 3.63 | **2.24** |
| Rosenbrock (3) | 31.37 | 25.99 | 17.55 | 28.79 | 171.04 | **3.81** |
| Shekel (4) | 2.56 | 2.21 | 2.03 | 2.70 | 2.06 | **1.72** |
| Hartmann3 (3) | 1.17 | **0.26** | 0.82 | 1.06 | 0.55 | 0.35 |
| Hartmann6 (6) | 1.33 | 1.25 | 0.44 | 1.46 | 0.61 | **0.32** |
| Powell (4) | 1992.1 | 1167.6 | 1223.4 | 534.2 | 9888.6 | **428.1** |
| Temperature (3) | 1.02 | 0.69 | 1.36 | 1.25 | 1.21 | **0.68** |
| WLAN (5) | 1.46 | 4.84 | 1.33 | 4.98 | 12.94 | **1.19** |
| Avg. Perf. | 0.48 | 0.47 | 0.29 | 0.54 | 0.62 | **0.01** |

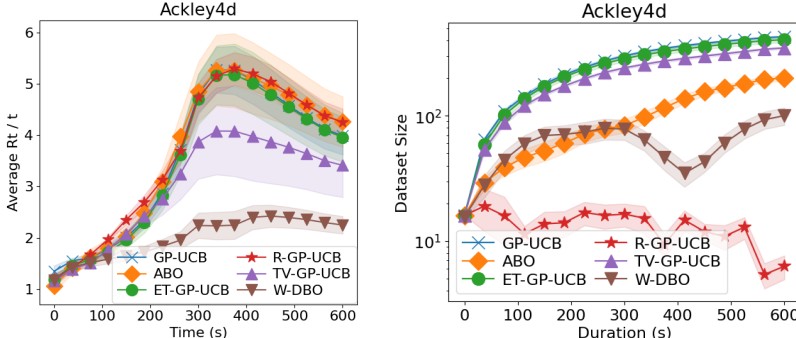

Figure 4: (Left) Average regrets of the DBO solutions during the optimization of the Ackley synthetic function. (Right) Dataset sizes of the DBO solutions during the optimization of the Ackley function.

To study this trade-off, we compare the performance of W-DBO with several values of its hyper-parameter $\alpha$, as illustrated by the left side of Figure 3. We apply this protocol on 11 different benchmarks (described in Appendix H.2). The aggregated results (see the right side of Figure 3) show that achieving a trade-off between the size of the dataset and the sampling frequency can significantly improve the performance of W-DBO. Clearly, the sweet spot is reached for $\alpha = \frac{1}{4}$. This hyperparameter value is used to evaluate W-DBO in the next section.

## 5.2 Comparison with Baselines

The competing baselines are the relevant algorithms reported in Section 2, namely R-GP-UCB and TV-GP-UCB [21], ET-GP-UCB [22] and ABO [20]. We also consider vanilla BO with the GP-UCB acquisition function, which only considers spatial correlations. For comparison purposes, all the results are gathered in Table 2. The benchmarks comprise ten synthetic functions and two real-world experiments. All figures (including standard errors) are provided in Appendix H.2, and the performance of each DBO solution is discussed at length in Appendix H.3. Furthermore, the provided supplementary animated visualizations are discussed in Appendix H.4.

In this section. we only depict the performance of the DBO solutions on the Ackley synthetic function in Figure 4, because it illustrates best the singularity of W-DBO. The Ackley function is known for its almost flat outer region (with lots of local minima) and its deep hole at the center of its domain. Observe that most DBO solutions miss that hole, as their average regrets skyrocket between 200 and

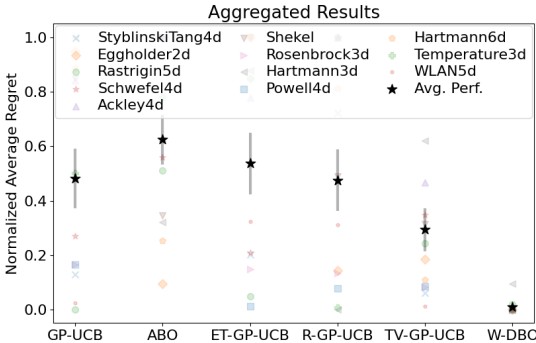

Figure 5: Visual summary of the results reported in Table 2. For aggregation purposes, the average regrets in each experiment have been normalized between 0 (lowest average regret) and 1 (largest average regret). The average performance of the DBO solutions is shown in black.

400 seconds. In contrast, W-DBO manages to rapidly exploit the hole at the center of the function domain, thereby maintaining a low average regret.

This performance gap can be explained by studying the dataset size of W-DBO (see the right side of Figure 4). At first, the dataset size increases since most collected observations are relevant to predict the outer region of the Ackley function. After 200 seconds, the dataset size plateaus as W-DBO begins to realize that some previously collected observations are irrelevant to predict the shape of the hole that lies ahead. Between 300 and 400 seconds, the dataset size is halved because most previously collected observations are deemed irrelevant. Eventually, after 400 seconds, W-DBO explores the flatter outer region of the Ackley function again. Consequently, its dataset size increases again.

For a summary of the performance of the DBO solutions across all our benchmarks, please refer to the last row of Table 2, and to the visual summary in Figure 5. In our experimental setting, ABO and ET-GP-UCB obtain roughly the same performance as vanilla BO. R-GP-UCB shows slightly better average performance than GP-UCB, while TV-GP-UCB appears significantly better than the aforementioned algorithms. Remarkably, W-DBO shows significantly better performance than TV-GP-UCB and outperforms the other DBO solutions by a comfortable margin. In fact, it obtains the lowest average regret on almost every benchmark.

## 6   Conclusion

The ability to remove irrelevant observations from the dataset of a DBO algorithm is essential to ensure a high sampling frequency while preserving its predictive performance. To address this difficult problem, we have proposed (i) a criterion based on the Wasserstein distance to measure the relevancy of an observation, (ii) a computationally tractable approximation of this criterion to allow its use in an online setting and (iii) a DBO algorithm, W-DBO, that exploits this approximation. We have evaluated W-DBO against the state-of-the-art of DBO on a variety of benchmarks comprising synthetic functions and real-world experiments. The evaluation was conducted in the most challenging settings, where time is continuous, the time horizon is unknown, as well as the covariance functions hyperparameters. We observe that W-DBO outperforms the state-of-the-art of DBO by a comfortable margin. We explain this significant performance gap by the ability of W-DBO to quantify the relevancy of each of its observations, which is not shared with any other DBO algorithm, to the best of our knowledge. As a result, W-DBO can remove irrelevant observations in a smoother and more appropriate way than by simply triggering the erasure of the whole dataset. By doing so, W-DBO simultaneously ensures a high sampling frequency and a very good predictive performance.

In addition to its impact on DBO itself, we believe that W-DBO can have a significant impact on the fields that make heavy use of DBO (e.g., computer networks, robotics). As a future work, we plan on exploring these applications of W-DBO. Furthermore, we plan on better understanding the excellent performance of W-DBO by addressing the difficult problem of deriving a regret bound that holds in a continuous time setting and incorporates the effect of the sampling frequency of the DBO algorithm as well as the deletion of irrelevant observations.

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

# A   Wasserstein Distance at a Point in $\mathcal{F}_{t_0}$

At the core of (8) lies (7). In this appendix, we provide explicit expressions for (7). Let us start by proving the following lemma.

**Lemma A.1.**

$$\boldsymbol{\Delta}_{\mathcal{D}}^{-1} = \begin{pmatrix} \boldsymbol{E} & \boldsymbol{G} \\ \boldsymbol{H} & \boldsymbol{F} \end{pmatrix} \tag{13}$$

with $\boldsymbol{\Delta}_{\mathcal{D}} = \boldsymbol{k}(\mathcal{D}, \mathcal{D}) + \sigma^2 \boldsymbol{I}$ and $\boldsymbol{E}, \boldsymbol{F}, \boldsymbol{G}, \boldsymbol{H}$ defined as

$$\boldsymbol{E} = \left( \lambda + \sigma^2 - \boldsymbol{k}^\top \left( (\boldsymbol{x}_1, t_1), \tilde{\mathcal{D}} \right) \boldsymbol{\Delta}_{\tilde{\mathcal{D}}}^{-1} \boldsymbol{k} \left( (\boldsymbol{x}_1, t_1), \tilde{\mathcal{D}} \right) \right)^{-1}, \tag{14}$$

$$\boldsymbol{F} = \left( \boldsymbol{\Delta}_{\tilde{\mathcal{D}}} - \frac{1}{\lambda + \sigma^2} \boldsymbol{k} \left( (\boldsymbol{x}_1, t_1), \tilde{\mathcal{D}} \right) \boldsymbol{k}^\top \left( (\boldsymbol{x}_1, t_1), \tilde{\mathcal{D}} \right) \right)^{-1}, \tag{15}$$

$$\boldsymbol{G} = -\boldsymbol{E} \boldsymbol{k}^\top \left( (\boldsymbol{x}_1, t_1), \tilde{\mathcal{D}} \right) \boldsymbol{\Delta}_{\tilde{\mathcal{D}}}^{-1}, \tag{16}$$

$$\boldsymbol{H} = -\frac{1}{\lambda + \sigma^2} \boldsymbol{F} \boldsymbol{k} \left( (\boldsymbol{x}_1, t_1), \tilde{\mathcal{D}} \right), \tag{17}$$

where $\boldsymbol{k}\left( \mathcal{X}, \mathcal{Y} \right) = \left( k \left( (\boldsymbol{x}_i, t_i), (\boldsymbol{x}_j, t_j) \right) \right)_{\substack{(\boldsymbol{x}_i, t_i) \in \mathcal{X} \\ (\boldsymbol{x}_j, t_j) \in \mathcal{Y}}}$ and $\boldsymbol{\Delta}_{\tilde{\mathcal{D}}} = \boldsymbol{k}(\tilde{\mathcal{D}}, \tilde{\mathcal{D}}) + \sigma^2 \boldsymbol{I}$.

*Proof.* The proof is trivial using the inverse of a block matrix:

$$\begin{pmatrix} \boldsymbol{A} & \boldsymbol{B} \\ \boldsymbol{C} & \boldsymbol{D} \end{pmatrix}^{-1} = \begin{pmatrix} (\boldsymbol{A} - \boldsymbol{B} \boldsymbol{D}^{-1} \boldsymbol{C})^{-1} & \boldsymbol{0} \\ \boldsymbol{0} & (\boldsymbol{D} - \boldsymbol{C} \boldsymbol{A}^{-1} \boldsymbol{B})^{-1} \end{pmatrix} \begin{pmatrix} \boldsymbol{I} & -\boldsymbol{B} \boldsymbol{D}^{-1} \\ -\boldsymbol{C} \boldsymbol{A}^{-1} & \boldsymbol{I} \end{pmatrix}$$
$$= \begin{pmatrix} (\boldsymbol{A} - \boldsymbol{B} \boldsymbol{D}^{-1} \boldsymbol{C})^{-1} & -(\boldsymbol{A} - \boldsymbol{B} \boldsymbol{D}^{-1} \boldsymbol{C})^{-1} \boldsymbol{B} \boldsymbol{D}^{-1} \\ -(\boldsymbol{D} - \boldsymbol{C} \boldsymbol{A}^{-1} \boldsymbol{B})^{-1} \boldsymbol{C} \boldsymbol{A}^{-1} & (\boldsymbol{D} - \boldsymbol{C} \boldsymbol{A}^{-1} \boldsymbol{B})^{-1} \end{pmatrix}. \tag{18}$$

Note that $\boldsymbol{A}$ and $\boldsymbol{D}$ must be invertible.

We can use (18) to write $\boldsymbol{\Delta}_{\mathcal{D}}^{-1}$ as a function of $\boldsymbol{\Delta}_{\tilde{\mathcal{D}}}^{-1}$, since

$$\boldsymbol{\Delta}_{\mathcal{D}}^{-1} = \begin{pmatrix} k((\boldsymbol{x}_1, t_1), (\boldsymbol{x}_1, t_1)) + \sigma^2 & \boldsymbol{k}^\top((\boldsymbol{x}_1, t_1), \tilde{\mathcal{D}}) \\ \boldsymbol{k}((\boldsymbol{x}_1, t_1), \tilde{\mathcal{D}}) & \boldsymbol{\Delta}_{\tilde{\mathcal{D}}} \end{pmatrix}^{-1}$$
$$= \begin{pmatrix} \lambda + \sigma^2 & \boldsymbol{k}^\top((\boldsymbol{x}_1, t_1), \tilde{\mathcal{D}}) \\ \boldsymbol{k}((\boldsymbol{x}_1, t_1), \tilde{\mathcal{D}}) & \boldsymbol{\Delta}_{\tilde{\mathcal{D}}} \end{pmatrix}^{-1}. \tag{19}$$

Note that, in (19), $\lambda + \sigma^2$ and $\boldsymbol{\Delta}_{\tilde{\mathcal{D}}}$ are invertible. Therefore, (18) can be applied to (19), and this yields the desired result. $\square$

To compute (7), we provide the following results.

**Proposition A.2.**

$$\mu_{\mathcal{D}}(\boldsymbol{x}, t) - \mu_{\tilde{\mathcal{D}}}(\boldsymbol{x}, t) = a k((\boldsymbol{x}, t), (\boldsymbol{x}_1, t_1)) + \boldsymbol{b} \boldsymbol{k} \left( (\boldsymbol{x}, t), \tilde{\mathcal{D}} \right), \tag{20}$$

with $a = \boldsymbol{E} y_1 + \boldsymbol{G} \tilde{\boldsymbol{y}}$, $\boldsymbol{b}^\top = \boldsymbol{H}^\top y_1 + \tilde{\boldsymbol{y}}^\top \left( \boldsymbol{F} - \boldsymbol{\Delta}_{\tilde{\mathcal{D}}}^{-1} \right)$ and $\tilde{\boldsymbol{y}} = (y_2, \cdots, y_n)$.

**Proposition A.3.**

$$\left( \sigma_{\mathcal{D}}(\boldsymbol{x}, t) - \sigma_{\tilde{\mathcal{D}}}(\boldsymbol{x}, t) \right)^2 \leq \boldsymbol{E} k((\boldsymbol{x}, t), (\boldsymbol{x}_1, t_1))^2 + k((\boldsymbol{x}, t), (\boldsymbol{x}_1, t_1)) \boldsymbol{c} \boldsymbol{k}((\boldsymbol{x}, t), \tilde{\mathcal{D}})$$
$$+ \boldsymbol{k}^\top((\boldsymbol{x}, t), \tilde{\mathcal{D}}) \boldsymbol{M} \boldsymbol{k}((\boldsymbol{x}, t), \tilde{\mathcal{D}}), \tag{21}$$

with $\boldsymbol{c} = \boldsymbol{G} + \boldsymbol{H}^\top$ and $\boldsymbol{M} = \boldsymbol{F} - \boldsymbol{\Delta}_{\tilde{\mathcal{D}}}^{-1}$.

We now prove Proposition A.2.

*Proof.* According to (3)

$$\mu_{\tilde{\mathcal{D}}}(\boldsymbol{x}, t) = \boldsymbol{k}^{\top}((\boldsymbol{x}, t), \tilde{\mathcal{D}})\boldsymbol{\Delta}_{\tilde{\mathcal{D}}}^{-1}\tilde{\boldsymbol{y}}. \tag{22}$$

Applying the same definition for $\mu_{\mathcal{D}}(\boldsymbol{x}, t)$ we have

$$\begin{aligned}
\mu_{\mathcal{D}}(\boldsymbol{x}, t) &= \left(k((\boldsymbol{x}, t), (\boldsymbol{x}_1, t_1)), \boldsymbol{k}^{\top}((\boldsymbol{x}, t), \tilde{\mathcal{D}})\right)\boldsymbol{\Delta}_{\mathcal{D}}^{-1}\boldsymbol{y} \\
&= \left(k((\boldsymbol{x}, t), (\boldsymbol{x}_1, t_1)), \boldsymbol{k}^{\top}((\boldsymbol{x}, t), \tilde{\mathcal{D}})\right)\begin{pmatrix} \boldsymbol{E} & \boldsymbol{G} \\ \boldsymbol{H} & \boldsymbol{F} \end{pmatrix}\boldsymbol{y} \tag{23} \\
&= \left(k((\boldsymbol{x}, t), (\boldsymbol{x}_1, t_1)), \boldsymbol{k}^{\top}((\boldsymbol{x}, t), \tilde{\mathcal{D}})\right)\begin{pmatrix} \boldsymbol{E}y_1 + \boldsymbol{G}\tilde{\boldsymbol{y}} \\ \boldsymbol{H}y_1 + \boldsymbol{F}\tilde{\boldsymbol{y}} \end{pmatrix} \\
&= k((\boldsymbol{x}, t), (\boldsymbol{x}_1, t_1))\left(\boldsymbol{E}y_1 + \boldsymbol{G}\tilde{\boldsymbol{y}}\right) + \boldsymbol{k}^{\top}((\boldsymbol{x}, t), \tilde{\mathcal{D}})\left(\boldsymbol{H}y_1 + \boldsymbol{F}\tilde{\boldsymbol{y}}\right), \tag{24}
\end{aligned}$$

where (23) follows from Lemma A.1.

Finally, we have

$$\begin{aligned}
\mu_{\mathcal{D}}(\boldsymbol{x}, t) - \mu_{\tilde{\mathcal{D}}}(\boldsymbol{x}, t) = k((\boldsymbol{x}, t), (\boldsymbol{x}_1, t_1))\left(\boldsymbol{E}y_1 + \boldsymbol{G}\tilde{\boldsymbol{y}}\right) + \boldsymbol{k}^{\top}((\boldsymbol{x}, t), \tilde{\mathcal{D}})\left(\boldsymbol{H}y_1 + \boldsymbol{F}\tilde{\boldsymbol{y}}\right) \\
- \boldsymbol{k}^{\top}((\boldsymbol{x}, t), \tilde{\mathcal{D}})\boldsymbol{\Delta}_{\tilde{\mathcal{D}}}^{-1}\tilde{\boldsymbol{y}}
\end{aligned}$$

which can be reduced to

$$\mu_{\mathcal{D}}(\boldsymbol{x}, t) - \mu_{\tilde{\mathcal{D}}}(\boldsymbol{x}, t) = ak((\boldsymbol{x}, t), (\boldsymbol{x}_1, t_1)) + \boldsymbol{bk}((\boldsymbol{x}, t), \tilde{\mathcal{D}}) \tag{25}$$

with $a = \boldsymbol{E}y_1 + \boldsymbol{G}\tilde{\boldsymbol{y}}$ and $\boldsymbol{b}^{\top} = \boldsymbol{H}^{\top}y_1 + \tilde{\boldsymbol{y}}^{\top}\left(\boldsymbol{F} - \boldsymbol{\Delta}_{\tilde{\mathcal{D}}}^{-1}\right)$. This concludes the proof. $\square$

Finally, we prove Proposition A.3.

*Proof.* As $(\sigma_{\mathcal{D}}(\boldsymbol{x}, t) - \sigma_{\tilde{\mathcal{D}}}(\boldsymbol{x}, t))^2$ is hard to integrate, to get (8), we upper bound it by

$$\begin{aligned}
(\sigma_{\mathcal{D}}(\boldsymbol{x}, t) - \sigma_{\tilde{\mathcal{D}}}(\boldsymbol{x}, t))^2 &= \sigma_{\mathcal{D}}^2(\boldsymbol{x}, t) + \sigma_{\tilde{\mathcal{D}}}^2(\boldsymbol{x}, t) - 2\sigma_{\mathcal{D}}(\boldsymbol{x}, t)\sigma_{\tilde{\mathcal{D}}}(\boldsymbol{x}, t) \\
&\leq \sigma_{\mathcal{D}}^2(\boldsymbol{x}, t) + \sigma_{\tilde{\mathcal{D}}}^2(\boldsymbol{x}, t) - 2\sigma_{\mathcal{D}}^2(\boldsymbol{x}, t) \tag{26} \\
&= \sigma_{\tilde{\mathcal{D}}}^2(\boldsymbol{x}, t) - \sigma_{\mathcal{D}}^2(\boldsymbol{x}, t) \tag{27}
\end{aligned}$$

where (26) follows from $\sigma_{\mathcal{D}}(\boldsymbol{x}, t) \leq \sigma_{\tilde{\mathcal{D}}}(\boldsymbol{x}, t)$.

Now, (4) yields that for $\boldsymbol{X} = \tilde{\mathcal{D}}$,

$$\sigma_{\tilde{\mathcal{D}}}^2(\boldsymbol{x}, t) = \lambda - \boldsymbol{k}^{\top}((\boldsymbol{x}, t), \tilde{\mathcal{D}})\boldsymbol{\Delta}_{\tilde{\mathcal{D}}}^{-1}\boldsymbol{k}((\boldsymbol{x}, t), \tilde{\mathcal{D}}). \tag{28}$$

and for $\boldsymbol{X} = \mathcal{D} = \tilde{\mathcal{D}} \cup \{(\boldsymbol{x}_1, t_1)\}$

$$\begin{aligned}
\sigma_{\mathcal{D}}^2(\boldsymbol{x}, t) &= \lambda - \left(k((\boldsymbol{x}, t), (\boldsymbol{x}_1, t_1)), \boldsymbol{k}^{\top}((\boldsymbol{x}, t), \tilde{\mathcal{D}})\right)\boldsymbol{\Delta}_{\mathcal{D}}^{-1}\begin{pmatrix} k((\boldsymbol{x}, t), (\boldsymbol{x}_1, t_1)) \\ \boldsymbol{k}((\boldsymbol{x}, t), \tilde{\mathcal{D}}) \end{pmatrix} \\
&= \lambda - \left(k((\boldsymbol{x}, t), (\boldsymbol{x}_1, t_1)), \boldsymbol{k}^{\top}((\boldsymbol{x}, t), \tilde{\mathcal{D}})\right)\begin{pmatrix} \boldsymbol{E} & \boldsymbol{G} \\ \boldsymbol{H} & \boldsymbol{F} \end{pmatrix}\begin{pmatrix} k((\boldsymbol{x}, t), (\boldsymbol{x}_1, t_1)) \\ \boldsymbol{k}((\boldsymbol{x}, t), \tilde{\mathcal{D}}) \end{pmatrix} \tag{29} \\
&= \lambda - \left(k((\boldsymbol{x}, t), (\boldsymbol{x}_1, t_1)), \boldsymbol{k}^{\top}((\boldsymbol{x}, t), \tilde{\mathcal{D}})\right)\begin{pmatrix} \boldsymbol{E}k((\boldsymbol{x}, t), (\boldsymbol{x}_1, t_1)) + \boldsymbol{G}\boldsymbol{k}((\boldsymbol{x}, t), \tilde{\mathcal{D}}) \\ \boldsymbol{H}k((\boldsymbol{x}, t), (\boldsymbol{x}_1, t_1)) + \boldsymbol{F}\boldsymbol{k}((\boldsymbol{x}, t), \tilde{\mathcal{D}}) \end{pmatrix} \\
& \tag{30}
\end{aligned}$$

where (29) follows from Lemma A.1. Developing the dot product in (30), we have

$$\begin{aligned}
\sigma_{\mathcal{D}}^2(\boldsymbol{x}, t) = \lambda - \boldsymbol{E}k^2((\boldsymbol{x}, t), (\boldsymbol{x}_1, t_1)) - k((\boldsymbol{x}, t), (\boldsymbol{x}_1, t_1))\boldsymbol{G}\boldsymbol{k}((\boldsymbol{x}, t), \tilde{\mathcal{D}}) \\
- k((\boldsymbol{x}, t), (\boldsymbol{x}_1, t_1))\boldsymbol{H}^{\top}\boldsymbol{k}((\boldsymbol{x}, t), \tilde{\mathcal{D}}) - \boldsymbol{k}^{\top}((\boldsymbol{x}, t), \tilde{\mathcal{D}})\boldsymbol{F}\boldsymbol{k}((\boldsymbol{x}, t), \tilde{\mathcal{D}}).
\end{aligned} \tag{31}$$

Combining (28) and (31), we get

$$\begin{aligned}
\sigma_{\tilde{\mathcal{D}}}^2(\boldsymbol{x}, t) - \sigma_{\mathcal{D}}^2(\boldsymbol{x}, t) = \boldsymbol{E}k((\boldsymbol{x}, t), (\boldsymbol{x}_1, t_1))^2 + k((\boldsymbol{x}, t), (\boldsymbol{x}_1, t_1))\boldsymbol{c}\boldsymbol{k}((\boldsymbol{x}, t), \tilde{\mathcal{D}}) \\
+ \boldsymbol{k}^{\top}((\boldsymbol{x}, t), \tilde{\mathcal{D}})\boldsymbol{M}\boldsymbol{k}((\boldsymbol{x}, t), \tilde{\mathcal{D}})
\end{aligned} \tag{32}$$

where $\boldsymbol{c} = \boldsymbol{G} + \boldsymbol{H}^{\top}$ and $\boldsymbol{M} = \boldsymbol{F} - \boldsymbol{\Delta}_{\tilde{\mathcal{D}}}^{-1}$.

Combining (27) and (32) concludes the proof. $\square$

# B    Wasserstein Distance on $\mathcal{F}_{t_0}$

In this appendix, we extend the computation of the Wasserstein distance between two posterior Gaussian distributions at an arbitrary point in $\mathcal{F}_{t_0}$ (see Propositions A.2 and A.3) to the Wasserstein distance between $\mathcal{GP}_\mathcal{D}$ and $\mathcal{GP}_{\tilde{\mathcal{D}}}$ on $\mathcal{F}_{t_0}$. We also provide an upper bound for the Wasserstein distance between a posterior GP conditioned on $\mathcal{D}$ (*i.e.*, $\mathcal{GP}_\mathcal{D}$) and the prior GP (*i.e.*, $\mathcal{GP}_\emptyset$).

Let us start by proving the following lemma.

**Lemma B.1.** *Let $t_0$ be the present time and $\mathcal{D} = \{((\boldsymbol{x}_i, t_i), y_i)\}_{i\in[\![1,n]\!]}$ be a dataset of observations before $t_0$. Let $\tilde{\mathcal{D}} = \{((\boldsymbol{x}_i, t_i), y_i)\}_{i\in[\![2,n]\!]}$ and $\mathcal{F}_{t_0} = \mathcal{S}\times[t_0, +\infty)$ be the domain of future predictions. Then, for any pair of observations $(\boldsymbol{x}_i, t_i, \cdot), (\boldsymbol{x}_j, t_j, \cdot)$ from $\mathcal{D}$ we have*

$$\oint_\mathcal{S} \int_{t_0}^\infty k\left((\boldsymbol{x}, t), (\boldsymbol{x}_i, t_i)\right) k\left((\boldsymbol{x}, t), (\boldsymbol{x}_j, t_j)\right) d\boldsymbol{x}dt \leq \lambda^2 (k_S * k_S)(\boldsymbol{x}_j - \boldsymbol{x}_i)(k_T * k_T)_{t_0 - t_i}^{+\infty}(t_j - t_i) \tag{33}$$

*with $(f * g)$ the convolution between $f$ and $g$ and $(f * g)_a^b$ the convolution between $f$ and $g$ restricted to the interval $[a, b]$.*

*Proof.* For the sake of brevity, let us denote by $I$ the LHS of (33). According to Assumption 3.2, Fubini's theorem and since $\mathcal{F}_{t_0} = \mathcal{S} \times [t_0, +\infty)$, we have

$$I = \lambda^2 \oint_\mathcal{S} k_S(||\boldsymbol{x} - \boldsymbol{x}_i||_2)k_S(||\boldsymbol{x} - \boldsymbol{x}_j||_2)d\boldsymbol{x} \int_{t_0}^{+\infty} k_T(|t - t_i|)k_T(|t - t_j|)dt. \tag{34}$$

Let us focus on the integral on the spatial domain $\mathcal{S}$. Depending on the expression of the covariance function, this integral can be quite difficult to compute. Let us turn this expression into a more pleasant upper bound

$$\oint_\mathcal{S} k_S(||\boldsymbol{x} - \boldsymbol{x}_i||_2)k_S(||\boldsymbol{x} - \boldsymbol{x}_j||_2)d\boldsymbol{x} \leq \oint_{\mathbb{R}^d} k_S(||\boldsymbol{x} - \boldsymbol{x}_i||_2)k_S(||\boldsymbol{x} - \boldsymbol{x}_j||_2)d\boldsymbol{x} \tag{35}$$

$$= \oint_{\mathbb{R}^d} k_S(||\boldsymbol{u}||_2)k_S(||\boldsymbol{x}_j - \boldsymbol{x}_i - \boldsymbol{u}||_2)d\boldsymbol{u} \tag{36}$$

where (35) holds since $k_S$ is positive and (36) comes from the change of variable $\boldsymbol{u} = \boldsymbol{x} - \boldsymbol{x}_i$. Obviously, this upper bound remains interesting because $k_S$ is usually an exponentially decreasing function (see Table 1). We discuss this point in more details in Appendix C.

Observe that (36) is actually the convolution $(k_S * k_S)(\boldsymbol{x}_j - \boldsymbol{x}_i)$. A similar change of variable can be made regarding the time integral, with $v = t - t_i$:

$$\int_{t_0}^{+\infty} k_T(|t - t_i|)k_T(|t - t_j|)dt = \int_{t_0-t_i}^{+\infty} k_T(|v|)k_T(|t_j - t_i - v|)dv. \tag{37}$$

Note that (37) is also a convolution, but restricted to the interval $[t_0 - t_i, +\infty)$. We denote this convolution on a restricted interval $(k_T * k_T)_{t_0-t_i}^{+\infty}(t_j - t_i)$.

Combining (34), (36) and (37) yields Lemma B.1. $\qquad\square$

We now prove the following lemma.

**Lemma B.2.** *Let $t_0$ be the present time and $\mathcal{D} = \{((\boldsymbol{x}_i, t_i), y_i)\}_{i\in[\![1,n]\!]}$ be a dataset of observations before $t_0$. Let $\tilde{\mathcal{D}} = \{((\boldsymbol{x}_i, t_i), y_i)\}_{i\in[\![2,n]\!]}$ and $\mathcal{F}_{t_0} = \mathcal{S} \times [t_0, +\infty)$ the domain of future predictions. Then, we have*

$$\oint_\mathcal{S} \int_{t_0}^\infty (\mu_\mathcal{D}(\boldsymbol{x}, t) - \mu_{\tilde{\mathcal{D}}}(\boldsymbol{x}, t))^2 d\boldsymbol{x}dt \leq \lambda^2 a^2 C((\boldsymbol{x}_1, t_1), (\boldsymbol{x}_1, t_1)) + 2\lambda^2 a\boldsymbol{b}C((\boldsymbol{x}_1, t_1), \tilde{\mathcal{D}})$$
$$+ \lambda^2 \operatorname{tr}\left(\boldsymbol{b}\boldsymbol{b}^\top C(\tilde{\mathcal{D}}, \tilde{\mathcal{D}})\right), \tag{38}$$

*with $a = \boldsymbol{E}y_1 + \boldsymbol{G}\tilde{\boldsymbol{y}}$ and $\boldsymbol{b}^\top = \boldsymbol{H}^\top y_1 + \tilde{\boldsymbol{y}}^\top\left(\boldsymbol{F} - \boldsymbol{\Delta}_{\tilde{\mathcal{D}}}^{-1}\right)$, $C(\mathcal{X}, \mathcal{Y}) = \left((k_S * k_S)(\boldsymbol{x}_j - \boldsymbol{x}_i)(k_T * k_T)_{t_0-t_i}^{+\infty}(t_j - t_i)\right)_{\substack{(\boldsymbol{x}_i, t_i)\in\mathcal{X}, \\ (\boldsymbol{x}_j, t_j)\in\mathcal{Y}}}$, $\boldsymbol{1}$ the conformable vector of ones,*

$(f * g)$ the convolution between $f$ and $g$ and $(f * g)_a^b$ the convolution between $f$ and $g$ restricted to the interval $[a, b]$.

*Proof.* By Proposition A.2,

$$\oint_{\mathcal{S}} \int_{t_0}^{\infty} (\mu_{\mathcal{D}}(\boldsymbol{x}, t) - \mu_{\tilde{\mathcal{D}}}(\boldsymbol{x}, t))^2 d\boldsymbol{x}dt = \oint_{\mathcal{S}} \int_{t_0}^{\infty} \left( ak((\boldsymbol{x}, t), (\boldsymbol{x}_1, t_1)) + \boldsymbol{bk}\left((\boldsymbol{x}, t), \tilde{\mathcal{D}}\right) \right)^2 d\boldsymbol{x}dt \tag{39}$$

with $a = \boldsymbol{E}y_1 + \boldsymbol{G}\tilde{\boldsymbol{y}}$ and $\boldsymbol{b}^\top = \boldsymbol{H}^\top y_1 + \tilde{\boldsymbol{y}}^\top \left( \boldsymbol{F} - \boldsymbol{\Delta}_{\tilde{\mathcal{D}}}^{-1} \right)$.

Expanding the square, we get three different integrals, denoted $A$, $B$ and $C$ with

$$A = \oint_{\mathcal{S}} \int_{t_0}^{\infty} a^2 k^2((\boldsymbol{x}, t), (\boldsymbol{x}_1, t_1)) d\boldsymbol{x}dt,$$

$$B = \oint_{\mathcal{S}} \int_{t_0}^{\infty} 2a\boldsymbol{b}k((\boldsymbol{x}, t), (\boldsymbol{x}_1, t_1))\boldsymbol{k}((\boldsymbol{x}, t), \tilde{\mathcal{D}}) d\boldsymbol{x}dt,$$

$$C = \oint_{\mathcal{S}} \int_{t_0}^{\infty} \boldsymbol{bk}\left((\boldsymbol{x}, t), \tilde{\mathcal{D}}\right) \boldsymbol{k}^\top\left((\boldsymbol{x}, t), \tilde{\mathcal{D}}\right) \boldsymbol{b}^\top d\boldsymbol{x}dt.$$

Using Lemma B.1, computing an upper bound for $A$, $B$ and $C$ is immediate. In fact,

$$A \le \lambda^2 a^2 C((\boldsymbol{x}_1, t_1), (\boldsymbol{x}_1, t_1)) \tag{40}$$

$$B \le 2\lambda^2 a\boldsymbol{b}C((\boldsymbol{x}_1, t_1), \tilde{\mathcal{D}}) \tag{41}$$

$$C \le \lambda^2 \boldsymbol{1}^\top \left( \boldsymbol{bb}^\top \odot C(\tilde{\mathcal{D}}, \tilde{\mathcal{D}}) \right) \boldsymbol{1}$$

$$= \lambda^2 \operatorname{tr}\left( \boldsymbol{bb}^\top C(\tilde{\mathcal{D}}, \tilde{\mathcal{D}}) \right), \tag{42}$$

where $\odot$ is the Hadamard product and $\boldsymbol{1}$ the conformable vector of ones.

Adding (40), (41) and (42) together concludes our proof. $\qquad\square$

To get the first part of Theorem 4.1, we prove the following lemma.

**Lemma B.3.** *Let $t_0$ be the present time and $\mathcal{D} = \{((\boldsymbol{x}_i, t_i), y_i)\}_{i \in [\![1,n]\!]}$ be a dataset of observations before $t_0$. Let $\tilde{\mathcal{D}} = \{((\boldsymbol{x}_i, t_i), y_i)\}_{i \in [\![2,n]\!]}$ and $\mathcal{F}_{t_0} = \mathcal{S} \times [t_0, +\infty)$ the domain of future predictions. Then, we have*

$$\oint_{\mathcal{S}} \int_{t_0}^{\infty} (\sigma_{\mathcal{D}}(\boldsymbol{x}, t) - \sigma_{\tilde{\mathcal{D}}}(\boldsymbol{x}, t))^2 d\boldsymbol{x}dt \le \lambda^2 \boldsymbol{E}C((\boldsymbol{x}_1, t_1), (\boldsymbol{x}_1, t_1)) + \lambda^2 \boldsymbol{c}C((\boldsymbol{x}_1, t_1), \tilde{\mathcal{D}})$$

$$+ \lambda^2 \operatorname{tr}\left( \boldsymbol{M}C(\tilde{\mathcal{D}}, \tilde{\mathcal{D}}) \right), \tag{43}$$

*with* $\boldsymbol{c} = \boldsymbol{G} + \boldsymbol{H}^\top$, $\boldsymbol{M} = \boldsymbol{F} - \boldsymbol{\Delta}_{\tilde{\mathcal{D}}}^{-1}$, $C(\mathcal{X}, \mathcal{Y}) = ((k_S * k_S)(\boldsymbol{x}_j - \boldsymbol{x}_i)(k_T * k_T)_{t_0 - t_i}^{+\infty}(t_j - t_i))_{\substack{(\boldsymbol{x}_i, t_i) \in \mathcal{X} \\ (\boldsymbol{x}_j, t_j) \in \mathcal{Y}}}$, $(f * g)$ *the convolution between $f$ and $g$ and $(f * g)_a^b$ the convolution between $f$ and $g$ restricted to the interval $[a, b]$.*

*Proof.* This proof is conceptually identical to the previous one. By Proposition A.3,

$$\oint_{\mathcal{S}} \int_{t_0}^{\infty} (\sigma_{\mathcal{D}}(\boldsymbol{x}, t) - \sigma_{\tilde{\mathcal{D}}}(\boldsymbol{x}, t))^2 d\boldsymbol{x}dt \le \oint_{\mathcal{S}} \int_{t_0}^{\infty} (\boldsymbol{E}k((\boldsymbol{x}, t), (\boldsymbol{x}_1, t_1))^2 + k((\boldsymbol{x}, t), (\boldsymbol{x}_1, t_1))\boldsymbol{c}\boldsymbol{k}((\boldsymbol{x}, t), \tilde{\mathcal{D}})$$

$$+ \boldsymbol{k}^\top((\boldsymbol{x}, t), \tilde{\mathcal{D}})\boldsymbol{M}\boldsymbol{k}((\boldsymbol{x}, t), \tilde{\mathcal{D}}))d\boldsymbol{x}dt \tag{44}$$

where $\boldsymbol{c} = \boldsymbol{G} + \boldsymbol{H}^\top$ and $\boldsymbol{M} = \boldsymbol{F} - \boldsymbol{\Delta}_{\tilde{\mathcal{D}}}^{-1}$.

By linearity of the integral, the RHS of (44) can be split into three different integrals $A$, $B$ and $C$ where

$$A = \oint_{\mathcal{S}} \int_{t_0}^{\infty} \boldsymbol{E}k((\boldsymbol{x},t),(\boldsymbol{x}_1,t_1))^2 d\boldsymbol{x}dt$$

$$B = \oint_{\mathcal{S}} \int_{t_0}^{\infty} k((\boldsymbol{x},t),(\boldsymbol{x}_1,t_1))\boldsymbol{c}\boldsymbol{k}((\boldsymbol{x},t),\tilde{\mathcal{D}}) d\boldsymbol{x}dt$$

$$C = \oint_{\mathcal{S}} \int_{t_0}^{\infty} \boldsymbol{k}^{\top}((\boldsymbol{x},t),\tilde{\mathcal{D}})\boldsymbol{M}\boldsymbol{k}((\boldsymbol{x},t),\tilde{\mathcal{D}}) d\boldsymbol{x}dt$$

Once again, Lemma B.1 can be used to compute an upper bound for $A$, $B$ and $C$. In fact,

$$A \leq \lambda^2 \boldsymbol{E}C((\boldsymbol{x}_1,t_1),(\boldsymbol{x}_1,t_1)) \tag{45}$$

$$B \leq \lambda^2 \boldsymbol{c}C((\boldsymbol{x}_1,t_1),\tilde{\mathcal{D}}) \tag{46}$$

$$C \leq \lambda^2 \mathbf{1}^{\top}\left(\boldsymbol{M}\odot C(\tilde{\mathcal{D}},\tilde{\mathcal{D}})\right)\mathbf{1}$$

$$= \operatorname{tr}\left(\boldsymbol{M}C(\tilde{\mathcal{D}},\tilde{\mathcal{D}})\right), \tag{47}$$

where $\odot$ is the Hadamard product and $\mathbf{1}$ is the conformable vector of ones.

Adding (45), (46) and (47) together concludes the proof. $\qquad\square$

Together, Lemmas B.1, B.2 and B.3 yield the first part of Theorem 4.1. For the second part of Theorem 4.1, we prove the following lemma.

**Lemma B.4.** *Let $t_0$ be the present time and $\mathcal{D} = \{((\boldsymbol{x}_i,t_i),y_i)\}_{i\in[\![1,n]\!]}$ be a dataset of observations before $t_0$. Let $\mathcal{F}_{t_0} = \mathcal{S}\times[t_0,+\infty)$ the domain of future predictions. Then, we have*

$$W_2^2(\mathcal{GP}_{\mathcal{D}},\mathcal{GP}_{\emptyset}) \leq \lambda^2\left(\boldsymbol{a}^{\top}C(\mathcal{D},\mathcal{D})\boldsymbol{a} + \operatorname{tr}\left(\boldsymbol{\Delta}^{-1}C(\mathcal{D},\mathcal{D})\right)\right) \tag{48}$$

*with $\boldsymbol{a} = \boldsymbol{\Delta}^{-1}\boldsymbol{y}$ and $C(\mathcal{X},\mathcal{Y}) = \left((k_S*k_S)(\boldsymbol{x}_j-\boldsymbol{x}_i)(k_T*k_T)_{t_0-t_i}^{+\infty}(t_j-t_i)\right)_{\substack{(\boldsymbol{x}_i,t_i)\in\mathcal{X},\\(\boldsymbol{x}_j,t_j)\in\mathcal{Y}}}$, $(f*g)$ the convolution between $f$ and $g$ and $(f*g)_a^b$ the convolution between $f$ and $g$ restricted to the interval $[a,b]$.*

*Proof.* Recall that, according to $\mathcal{GP}_{\emptyset}$, $f(\boldsymbol{x},t)\sim\mathcal{N}(0,\lambda)$ for any point $(\boldsymbol{x},t)\in\mathcal{F}_{t_0}$. Consequently,

$$W_2(\mathcal{GP}_{\mathcal{D}},\mathcal{GP}_{\emptyset}) = \left(\oint_{\mathcal{S}}\int_{t_0}^{\infty}\mu_{\mathcal{D}}^2(\boldsymbol{x},t)d\boldsymbol{x}dt + \oint_{\mathcal{S}}\int_{t_0}^{\infty}\left(\sqrt{\lambda}-\sigma_{\mathcal{D}}(\boldsymbol{x},t)\right)^2 d\boldsymbol{x}dt\right)^{\frac{1}{2}}. \tag{49}$$

These two integrals in (49) can be computed with the same techniques as above. For the mean integral, we have

$$\oint_{\mathcal{S}}\int_{t_0}^{\infty}\mu_{\mathcal{D}}^2(\boldsymbol{x},t)d\boldsymbol{x}dt = \oint_{\mathcal{S}}\int_{t_0}^{\infty}\boldsymbol{y}^{\top}\boldsymbol{\Delta}^{-\top}\boldsymbol{k}^{\top}((\boldsymbol{x},t),\mathcal{D})\boldsymbol{k}((\boldsymbol{x},t),\mathcal{D})\boldsymbol{\Delta}^{-1}\boldsymbol{y}d\boldsymbol{x}dt$$

$$= \boldsymbol{y}^{\top}\boldsymbol{\Delta}^{-\top}\left(\oint_{\mathcal{S}}\int_{t_0}^{\infty}\boldsymbol{k}^{\top}((\boldsymbol{x},t),\mathcal{D})\boldsymbol{k}((\boldsymbol{x},t),\mathcal{D})d\boldsymbol{x}dt\right)\boldsymbol{\Delta}^{-1}\boldsymbol{y}$$

$$\leq \lambda^2\boldsymbol{y}^{\top}\boldsymbol{\Delta}^{-\top}C(\mathcal{D},\mathcal{D})\boldsymbol{\Delta}^{-1}\boldsymbol{y}. \tag{50}$$

Regarding the variance integral in (49), we have

$$\oint_{\mathcal{S}}\int_{t_0}^{\infty}\left(\sqrt{\lambda}-\sigma_{\mathcal{D}}(\boldsymbol{x},t)\right)^2 d\boldsymbol{x}dt = \oint_{\mathcal{S}}\int_{t_0}^{\infty}\left(\lambda-2\sqrt{\lambda}\sigma_{\mathcal{D}}(\boldsymbol{x},t)+\sigma_{\mathcal{D}}^2(\boldsymbol{x},t)\right)d\boldsymbol{x}dt$$

$$\leq \oint_{\mathcal{S}}\int_{t_0}^{\infty}\left(\lambda-\sigma_{\mathcal{D}}^2(\boldsymbol{x},t)\right)d\boldsymbol{x}dt \tag{51}$$

$$= \oint_{\mathcal{S}}\int_{t_0}^{\infty}\boldsymbol{k}^{\top}((\boldsymbol{x},t),\mathcal{D})\boldsymbol{\Delta}^{-1}\boldsymbol{k}((\boldsymbol{x},t),\mathcal{D})d\boldsymbol{x}dt \tag{52}$$

$$\leq \lambda^2\operatorname{tr}\left(\boldsymbol{\Delta}^{-1}C(\mathcal{D},\mathcal{D})\right) \tag{53}$$

where (51) holds because $\sqrt{\lambda}\sigma_{\mathcal{D}}(\boldsymbol{x}, t) \geq \sigma_{\mathcal{D}}^2(\boldsymbol{x}, t)$ and (52) holds because of (4).

Together, (50) and (53) conclude the proof. □

By combining Lemmas B.2 and B.3, the proof of Theorem 4.1 is immediate.

# C Approximation Error

In Appendix B, we provided a computationally tractable upper bound of $W_2(\mathcal{GP}_{\mathcal{D}}, \mathcal{GP}_{\tilde{\mathcal{D}}})$ and $W_2(\mathcal{GP}_{\mathcal{D}}, \mathcal{GP}_{\emptyset})$. In this appendix, we provide the expression of the corresponding approximation error and we study its magnitude with the Squared-Exponential (SE) covariance function.

First, and without loss of generality, assume $\mathcal{S} = [0, 1]^d$. In Appendix B, we have to approximate the Wasserstein distance because the integration over $\mathcal{S}$ in (34) is difficult to compute. The upper bound proposed in (35) is

$$\oint_{[0,1]^d} k_S(||\boldsymbol{x} - \boldsymbol{x}_i||_2)k_S(||\boldsymbol{x} - \boldsymbol{x}_j||_2)d\boldsymbol{x} \leq \oint_{\mathbb{R}^d} k_S(||\boldsymbol{x} - \boldsymbol{x}_i||_2)k_S(||\boldsymbol{x} - \boldsymbol{x}_j||_2)d\boldsymbol{x}. \tag{54}$$

Recall that the upper bounded quantity is a product of functions which are decreasing exponentially (see Table 1). As a consequence, their product decreases exponentially as well, so that extending the integration from $\mathcal{S} = [0, 1]^d$ to $\mathbb{R}^d$ has a bounded impact on the result. Clearly, a first absolute approximation error for (54) is

$$\oint_{(\mathbb{R}\setminus[0,1])^d} k_S(||\boldsymbol{x} - \boldsymbol{x}_i||_2)k_S(||\boldsymbol{x} - \boldsymbol{x}_j||_2)d\boldsymbol{x}.$$

Because the upper bounded quantity is a product of two correlation functions $k_S$ on a hypercube of volume 1, the upper bound can be capped to 1 as well. This leads to the more refined absolute approximation error:

$$A(\boldsymbol{x}_i, \boldsymbol{x}_j; l_S) = \min\left\{ \oint_{(\mathbb{R}\setminus[0,1])^d} k_S(||\boldsymbol{x} - \boldsymbol{x}_i||_2)k_S(||\boldsymbol{x} - \boldsymbol{x}_j||_2)d\boldsymbol{x}, \right.$$
$$\left. 1 - \oint_{[0,1]^d} k_S(||\boldsymbol{x} - \boldsymbol{x}_i||_2)k_S(||\boldsymbol{x} - \boldsymbol{x}_j||_2)d\boldsymbol{x} \right\}. \tag{55}$$

Obtaining a closed-form for the approximation error (55) is difficult. However, because the spatial lengthscale controls the correlation lengths in the spatial domain, it is clear that the left term in (55) is an increasing function with respect to $l_S$. Conversely, the right term in (55) is a decreasing function with respect to $l_S$. This observation allows us to derive the spatial lengthscale for which (55) is maximal.

**Proposition C.1.** *Let $(\boldsymbol{x}_i, \boldsymbol{x}_j) \in \mathcal{S}^2$, with $\mathcal{S} = [0, 1]^d$. Let $k_S$ be a SE kernel with lengthscale $l_S$. Then,*

$$\arg\max_{l_S \in \mathbb{R}^+} A(\boldsymbol{x}_i, \boldsymbol{x}_j; l_S) = \frac{1}{\sqrt{\pi}} e^{\frac{1}{2}W_0\left(\frac{\pi||\boldsymbol{x}_i - \boldsymbol{x}_j||_2^2}{2d}\right)}, \tag{56}$$

*with $W_0$ the principal branch of the Lambert function.*

*Proof.* Because the two terms in (55) are respectively increasing and decreasing with respect to the spatial lengthscale $l_S$, (55) is maximal when both terms are equal. Therefore, from (55), we have the following relation

$$\oint_{(\mathbb{R}\setminus[0,1])^d} k_S(||\boldsymbol{x} - \boldsymbol{x}_i||_2)k_S(||\boldsymbol{x} - \boldsymbol{x}_j||_2)d\boldsymbol{x} = 1 - \oint_{[0,1]^d} k_S(||\boldsymbol{x} - \boldsymbol{x}_i||_2)k_S(||\boldsymbol{x} - \boldsymbol{x}_j||_2)d\boldsymbol{x},$$

$$\oint_{\mathbb{R}^d} k_S(||\boldsymbol{x} - \boldsymbol{x}_i||_2)k_S(||\boldsymbol{x} - \boldsymbol{x}_j||_2)d\boldsymbol{x} = 1,$$

$$\pi^{\frac{d}{2}}l_S^d e^{\frac{-||\boldsymbol{x}_i - \boldsymbol{x}_j||_2^2}{4l_S^2}} = 1, \tag{57}$$

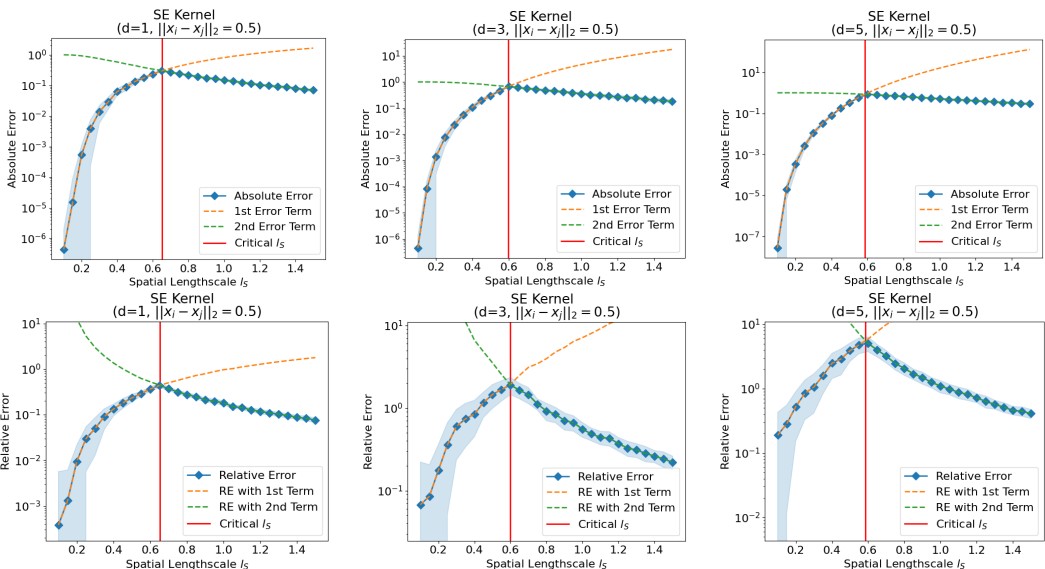

Figure 6: (Top row) Absolute approximation error (55) with respect to the spatial lengthscale $l_S$ for a 1, 3 and 5-dimensional spatial domain. Both error terms in (55) are shown in orange and green dashed lines, respectively. Finally, the critical lengthscale (56) is shown as a red vertical line. In this example, $k_S$ is a SE correlation function. (Bottom row) Relative approximation error with respect to the spatial lengthscale $l_S$. The color codes are the same.

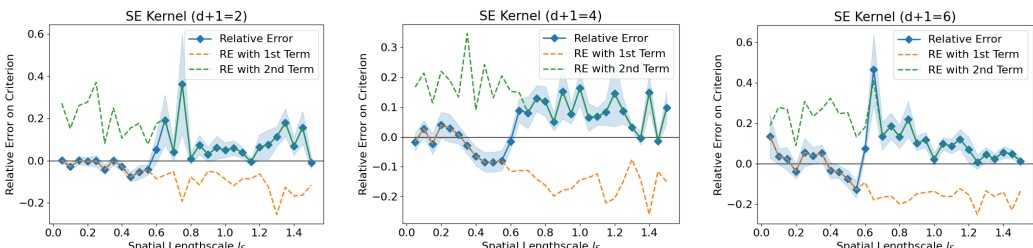

Figure 7: Relative error between the criterion (9) and its approximation (12), with respect to the spatial lengthscale $l_S$ for a 1, 3 and 5-dimensional spatial domain. The relative error computed with both terms in (55) are shown as orange and green dashed lines, respectively. In this example, $k_S$ is a SE correlation function.

where (57) uses a result derived in Appendix D and reported in Table 3. Solving for $l_S$ concludes the proof. $\qquad\square$

Let us illustrate (55) and Proposition C.1 by plotting the relative approximation error (55) with respect to the spatial lengthscale $l_S$. The integral over $\mathcal{S}$ is computed with a Monte-Carlo numerical integration technique. The results are shown in Figure 6. Looking at the top row, we see that the absolute approximation error peaks at a spatial lengthscale $l_S^*$ given by (56), as anticipated above. For $l_S < l_S^*$, the error is given by the first error term in (55), and conversely, the error is given by the second error term in (55) for $l_S > l_S^*$. The bottom row of Figure 6 shows that the same observations apply to the relative errors.

Figure 6 shows that even though it is bounded, the approximation error of the upper bound (54) is non-negligible. This is particularly noticeable when looking at the relative errors in the bottom row of Figure 6, which clearly increases in magnitude with the dimensionality of the spatial domain.

Nevertheless, recall that we seek to approximate the ratio (9) with a ratio of upper bounded Wasserstein distances (12) that involve (54) both in the numerator and in the denominator. Because the spatial lengthscale $l_S$ does not vary when computing the numerator and the denominator, the errors are of similar magnitude and point in the same direction (both the numerator and the denominator are upper

Table 3: Analytic forms for the convolution of usual spatial covariance functions. $\Gamma$ is the Gamma function, $J_\alpha$ is a Bessel function of the first kind of order $\alpha$, $K_\alpha$ is a modified Bessel function of the second kind of order $\alpha$.

| Covariance Function $k_S$ | $(k_S * k_S)(\boldsymbol{x})$ |
|---|---|
| Squared-Exponential ($l_S$) | $\pi^{\frac{d}{2}} l_S^d e^{\frac{-||\boldsymbol{x}||_2^2}{4l_S^2}}$ |
| Matérn ($\nu, l_S$) | $\frac{2^{\frac{d}{2}-2\nu+1}\pi^{\frac{d}{2}}\Gamma(\nu+\frac{d}{2})^2}{\Gamma(\nu)^2\Gamma(2\nu+d)}\left(\frac{\sqrt{2\nu}}{l_S}\right)^{2\nu-\frac{d}{2}}||\boldsymbol{x}||_2^{2\nu+\frac{d}{2}}K_{2\nu+\frac{d}{2}}\left(\frac{||\boldsymbol{x}||_2\sqrt{2\nu}}{l_S}\right)$ |

Table 4: Analytic forms for the convolution of usual temporal covariance functions on the interval $[t_0 - t_i, +\infty)$. Note that erf is the error function. Also, for the sake of brevity, the terms $C_{k_1 k_2}$, $P_{k_1 k_2}$ and $Q_{k_1 k_2}$ are defined in Appendix D.2.2.

| Covariance Function $k_T$ | $(k_T * k_T)_{t_0-t_i}^{+\infty}(t_j - t_i)$ |
|---|---|
| Squared-Exponential ($l_T$) | $\frac{\sqrt{\pi}l_T}{2}e^{\frac{-(t_i-t_j)^2}{2l_T^2}}\left(1 - \text{erf}\left(\frac{2t_0-t_i-t_j}{2l_T}\right)\right)$ |
| Matérn ($\nu = p + \frac{1}{2}, l_T$) | $\sum_{k_1=0}^{p}\sum_{k_2=0}^{p}C_{k_1 k_2}e^{\frac{-\sqrt{2p+1}(2t_0-t_i-t_j)}{l_T}}P_{k_1 k_2}(t_0, t_i, t_j)$ |

bounds). As a consequence, the approximation errors compensate each other (at least in part) when computing (12). To verify this observation numerically, we compute the relative approximation error between the criterion (9) and its approximation (12). The results are shown in Figure 7. Although the approximation errors in the numerator and the denominator do not entirely compensate each other, the approximation (12) appears to have lost most of its dependency to the dimensionality of the spatial domain and to the spatial lengthscale, making (12) a decent approximation of (9) regardless of $d$ or $l_S$. In the main paper, Section 5 corroborates this observation by demonstrating the usefulness of the approximation (12) in practice.

# D  Convolutions of Usual Covariance Functions

In this appendix, we derive the analytic forms of the convolution of usual covariance functions listed in Tables 3 and 4, which are used to compute the criterion (12).

## D.1  Spatial Covariance Functions

In this subsection, we compute specifically the analytic forms for the convolution of usual spatial covariance functions. We rely on a direct consequence of the convolution theorem, that is $(k*k)(\boldsymbol{x}) = \mathcal{F}^{-1}\left(\mathcal{F}^2(k)\right)(\boldsymbol{x})$, with $\mathcal{F}(f)$ denoting the Fourier transform of $f$ and $\mathcal{F}^{-1}(f)$ the inverse Fourier transform of $f$.

The Fourier transform $\mathcal{F}(k)$ of a stationary covariance function $k$ is called the spectral density of $k$, and is usually denoted $S$. Both functions are Fourier duals of each other (see [27] for more details). Furthermore, it is known that if $k$ is isotropic (*i.e.* it can be written as a function of $r = ||\boldsymbol{x}||_2$), then its spectral density $S(\boldsymbol{s})$ can be written as a function of $s = ||\boldsymbol{s}||_2$. In that case, the two functions are linked by the pair of transforms (see [27])

$$k(r) = \frac{2\pi}{r^{\frac{d}{2}-1}}\int_0^\infty S(s)J_{\frac{d}{2}-1}(2\pi rs)s^{\frac{d}{2}}\,ds \tag{58}$$

$$S(s) = \frac{2\pi}{s^{\frac{d}{2}-1}}\int_0^\infty k(r)J_{\frac{d}{2}-1}(2\pi rs)r^{\frac{d}{2}}\,dr \tag{59}$$

where $J_{\frac{d}{2}-1}$ is a Bessel function of the first kind and of order $d/2 - 1$.

As an immediate consequence of (58), (59) and the convolution theorem, we have the following corollary.

**Corollary D.1.** *Let $k$ be a stationary, isotropic covariance function with spectral density $S$. Let $r = ||\boldsymbol{x}||_2$ and $s = ||\boldsymbol{s}||_2$. Then,*

$$(k * k)(r) = \frac{2\pi}{r^{\frac{d}{2}-1}} \int_0^\infty S^2(s) J_{\frac{d}{2}-1}(2\pi rs) s^{\frac{d}{2}} ds \tag{60}$$

*where $J_{\frac{d}{2}-1}$ is a Bessel function of the first kind and of order $d/2 - 1$.*

We now derive the analytic forms for the convolutions of usual spatial covariance functions $k_S$.

### D.1.1 Squared-Exponential Covariance Function

**Lemma D.2.** *Let $k_S$ be a Squared-Exponential covariance function (see Table 1), with lengthscale $l_S > 0$. Then,*

$$(k_S * k_S)(\boldsymbol{x}) = \pi^{\frac{d}{2}} l_S^d e^{\frac{-||\boldsymbol{x}||_2^2}{4l_S^2}}. \tag{61}$$

*Proof.* The spectral density of a Squared-Exponential covariance function $k_S$ is (see [27])

$$S(s) = (2\pi l_S^2)^{\frac{d}{2}} e^{-2\pi^2 l_S^2 s^2}. \tag{62}$$

According to Corollary D.1, we have

$$\begin{aligned}
(k_S * k_S)(\boldsymbol{x}) &= \frac{2\pi}{r^{\frac{d}{2}-1}} \int_0^\infty S^2(s) J_{\frac{d}{2}-1}(2\pi rs) s^{\frac{d}{2}} ds \\
&= \frac{2\pi}{r^{\frac{d}{2}-1}} \int_0^\infty (2\pi l_S^2)^d e^{-4\pi^2 l_S^2 s^2} J_{\frac{d}{2}-1}(2\pi rs) s^{\frac{d}{2}} ds \\
&= \frac{(2\pi)^{d+1} l_S^{2d}}{r^{\frac{d}{2}-1}} \int_0^\infty e^{-4\pi^2 l_S^2 s^2} J_{\frac{d}{2}-1}(2\pi rs) s^{\frac{d}{2}} ds
\end{aligned} \tag{63}$$

where $r = ||\boldsymbol{x}||_2$ and $J_{\frac{d}{2}-1}$ is a Bessel function of the first kind of order $\frac{d}{2} - 1$.

It is known (see [31]) that

$$\int_0^\infty e^{-\alpha x^2} x^{\nu+1} J_\nu(\beta x) dx = \frac{\beta^\nu}{(2\alpha)^{\nu+1}} e^{\frac{-\beta^2}{4\alpha}}.$$

Therefore,

$$\begin{aligned}
(k_S * k_S)(\boldsymbol{x}) &= \frac{(2\pi)^{d+1} l_S^{2d}}{r^{\frac{d}{2}-1}} \frac{(2\pi r)^{\frac{d}{2}-1}}{(8\pi^2 l_S^2)^{\frac{d}{2}}} e^{\frac{-4\pi^2 r^2}{16\pi^2 l_S^2}} \\
&= \pi^{\frac{d}{2}} l_S^d e^{\frac{-r^2}{4l_S^2}}.
\end{aligned} \tag{64}$$

Replacing $r$ by $||\boldsymbol{x}||_2$ in (64) concludes the proof. $\qquad\square$

### D.1.2 Matérn Covariance Function

**Lemma D.3.** *Let $k_S$ be a Matérn covariance function (see Table 1), with smoothness parameter $\nu > 0$ and lengthscale $l_S > 0$. Then,*

$$(k_S * k_S)(\boldsymbol{x}) = \frac{2^{\frac{d}{2}-2\nu+1} \pi^{\frac{d}{2}} \Gamma(\nu + \frac{d}{2})^2}{\Gamma(\nu)^2 \Gamma(2\nu + d)} \left(\frac{\sqrt{2\nu}}{l_S}\right)^{2\nu-\frac{d}{2}} ||\boldsymbol{x}||_2^{2\nu+\frac{d}{2}} K_{2\nu+\frac{d}{2}}\left(\frac{||\boldsymbol{x}||_2\sqrt{2\nu}}{l_S}\right), \tag{65}$$

*where $\Gamma$ is the Gamma function and $K_\alpha$ is a modified Bessel function of the second kind of order $\alpha$.*

*Proof.* The spectral density of a Matérn covariance function $k_S$ is (see [27])

$$S(s) = \frac{2^d \pi^{\frac{d}{2}} \Gamma(\nu + \frac{d}{2})(2\nu)^\nu}{\Gamma(\nu) l_S^{2\nu}} \left(\frac{2\nu}{l_S^2} + 4\pi^2 s^2\right)^{-\nu-\frac{d}{2}} \tag{66}$$

where $\Gamma$ is the Gamma function.

According to Corollary D.1, we have

$$
\begin{aligned}
(k_S * k_S)(\boldsymbol{x}) &= \frac{2\pi}{r^{\frac{d}{2}-1}} \int_0^\infty S^2(s) J_{\frac{d}{2}-1}(2\pi rs) s^{\frac{d}{2}} ds \\
&= \frac{2\pi}{r^{\frac{d}{2}-1}} \int_0^\infty \frac{2^{2d}\pi^d \Gamma^2(\nu+\frac{d}{2})(2\nu)^{2\nu}}{\Gamma^2(\nu)l_S^{4\nu}} \left(\frac{2\nu}{l_S^2} + 4\pi^2 s^2\right)^{-2\nu-d} J_{\frac{d}{2}-1}(2\pi rs) s^{\frac{d}{2}} ds \\
&= \frac{2^{2d+1}\pi^{d+1}\Gamma^2(\nu+\frac{d}{2})(2\nu)^{2\nu}}{\Gamma^2(\nu)l_S^{4\nu} r^{\frac{d}{2}-1}} \int_0^\infty \left(\frac{2\nu}{l_S^2} + 4\pi^2 s^2\right)^{-2\nu-d} J_{\frac{d}{2}-1}(2\pi rs) s^{\frac{d}{2}} ds \\
&= \frac{2^{\frac{3d}{2}}\pi^{\frac{d}{2}}\Gamma^2(\nu+\frac{d}{2})(2\nu)^{2\nu}}{\Gamma^2(\nu)l_S^{4\nu} r^{\frac{d}{2}-1}} \int_0^\infty \left(\frac{2\nu}{l_S^2} + u^2\right)^{-2\nu-d} J_{\frac{d}{2}-1}(ru) u^{\frac{d}{2}} du \qquad (67)
\end{aligned}
$$

where $J_{\frac{d}{2}-1}$ is a Bessel function of the first kind of order $\frac{d}{2} - 1$, and (67) comes from the change of variable $u = 2\pi s$.

It is known (see [31]) that

$$
\int_0^\infty (a^2 + x^2)^{-(\mu+1)} J_\alpha(bx) x^{\alpha+1} dx = \frac{a^{\alpha-\mu} b^\mu}{2^\mu \Gamma(\mu+1)} K_{\mu-\alpha}(ab)
$$

where $K_{\mu-\alpha}$ is a modified Bessel function of the second kind of order $\mu - \alpha$.

Therefore,

$$
\begin{aligned}
(k_S * k_S)(\boldsymbol{x}) &= \frac{2^{\frac{3d}{2}}\pi^{\frac{d}{2}}\Gamma^2(\nu+\frac{d}{2})(2\nu)^{2\nu}}{\Gamma^2(\nu)l_S^{4\nu} r^{\frac{d}{2}-1}} \left(\frac{\sqrt{2\nu}}{l_S}\right)^{-2\nu-\frac{d}{2}} \frac{r^{2\nu+d-1}}{2^{2\nu+d-1}\Gamma(2\nu+d)} K_{2\nu+\frac{d}{2}}\left(\frac{r\sqrt{2\nu}}{l_S}\right) \\
&= \frac{2^{\frac{d}{2}-2\nu+1}\pi^{\frac{d}{2}}\Gamma(\nu+\frac{d}{2})^2}{\Gamma(\nu)^2\Gamma(2\nu+d)} \left(\frac{\sqrt{2\nu}}{l_S}\right)^{2\nu-\frac{d}{2}} r^{2\nu+\frac{d}{2}} K_{2\nu+\frac{d}{2}}\left(\frac{r\sqrt{2\nu}}{l_S}\right). \qquad (68)
\end{aligned}
$$

Replacing $r$ by $||\boldsymbol{x}||_2$ in (68) concludes the proof. $\qquad\square$

## D.2 Temporal Covariance Functions

In this section, we derive the analytic expression of the convolutions of the most popular temporal covariance functions $k_T$ restricted to the interval $[t_0 - t_i, +\infty)$. Therefore, we compute many integrals of the form

$$
\int_{t_0-t_i}^{+\infty} k_T(|t|) k_T(|t_j - t_i - t|) dt,
$$

which can be rewritten as

$$
\int_{t_0-t_i}^{+\infty} k_T(t) k_T(t + t_i - t_j) dt. \qquad (69)
$$

since $t \geq 0$ and $t_j - t_i - t \leq 0$ for all $t \in [t_0 - t_i, +\infty)$. The form (69) will be used in every proof of this section.

### D.2.1 Squared-Exponential Covariance Function

**Lemma D.4.** *Let $k_T$ be a Squared-Exponential covariance function (see Table 1), with lengthscale $l_S > 0$. Then,*

$$
(k_T * k_T)_{t_0-t_i}^{+\infty}(t_i - t_j) = \frac{\sqrt{\pi}l_T}{2} e^{\frac{-(t_i-t_j)^2}{2l_T^2}} \left(1 - erf\left(\frac{2t_0 - t_i - t_j}{2l_T}\right)\right) \qquad (70)
$$

*where erf is the error function.*

*Proof.* Since $k_T$ is a Squared-Exponential function, (69) becomes

$$\int_{t_0-t_i}^{+\infty} e^{\frac{-t^2}{2l_T^2}} e^{\frac{-(t-t_j+t_i)^2}{2l_T^2}} dt$$

$$= \int_{t_0-t_i}^{+\infty} e^{\frac{-(2t^2-2(t_j-t_i)t+t_j^2+t_i^2-2t_it_j)}{2l_T^2}} dt. \tag{71}$$

It is known (see [31]) that

$$\int e^{-(ax^2+2bx+c)} dx = \frac{1}{2}\sqrt{\frac{\pi}{a}} e^{\frac{b^2-ac}{a}} \operatorname{erf}\left(\sqrt{a}x + \frac{b}{\sqrt{a}}\right)$$

where erf is the error function.

Therefore, (71) becomes

$$\frac{\sqrt{\pi}l_T}{2} e^{\frac{-(t_i-t_j)^2}{2l_T^2}} \left(1 - \operatorname{erf}\left(\frac{2t_0-t_i-t_j}{2l_T}\right)\right).$$

$\square$

### D.2.2 Matérn Covariance Function

**Lemma D.5.** *Let $k_T$ be a Matérn covariance function (see Table 1), with smoothness parameter $\nu = p + \frac{1}{2}, p \in \mathbb{N}$ and lengthscale $l_T > 0$. Then,*

$$(k_T * k_T)_{t_0-t_i}^{l_T+t_0-t_i}(t_i - t_j) = \sum_{k_1=0}^{p} \sum_{k_2=0}^{p} C_{k_1k_2} e^{\frac{-\sqrt{2p+1}(2t_0-t_i-t_j)}{l_T}} P_{k_1k_2}(t_0, t_i, t_j) \tag{72}$$

*where*

$$C_{k_1k_2} = \left(\frac{p!}{(2p)!}\right)^2 \frac{(p+k_1)!(p+k_2)!}{k_1!k_2!(p-k_1)!(p-k_2)!} \left(\frac{2\sqrt{2p+1}}{l_T}\right)^{2p-k_1-k_2-1}, \tag{73}$$

$$P_{k_1k_2}(t_0, t_i, t_j) = \sum_{k_3=0}^{2p-k_1-k_2} \left(\frac{l_T}{2\sqrt{2p+1}}\right)^{k_3} P^{(k_3)}(t_0 - t_i), \tag{74}$$

$$P(t) = t^{p-k_1}(t - t_j + t_i)^{p-k_2} \tag{75}$$

*and $P^{(k)}$ the kth derivative of $P(t)$ with respect to t.*

*Proof.* The Matérn covariance function has a simpler form when its smoothness parameter $\nu$ is a half-integer, that is $\nu = p + \frac{1}{2}, p \in \mathbb{N}$ (see [30]). In that case,

$$k_T(t) = e^{\frac{-\sqrt{2p+1}t}{l_T}} \frac{p!}{(2p)!} \sum_{k_1=0}^{p} \frac{(p+k_1)!}{k_1!(p-k_1)!} \left(\frac{2\sqrt{2p+1}t}{l_T}\right)^{p-k_1}.$$

Therefore,

$$k_T(t)k_T(t + t_i - t_j) = \frac{2\sqrt{2p+1}}{l_T} \sum_{k_1=0}^{p} \sum_{k_2=0}^{p} C_{k_1k_2} e^{\frac{-\sqrt{2p+1}(2t-t_j+t_i)}{l_T}} P(t) \tag{76}$$

with $C_{k_1k_2}$ defined in (73) and $P(t)$ defined in (75).

Integrating (76), we get

$$\frac{2\sqrt{2p+1}}{l_T} \sum_{k_1=0}^{p} \sum_{k_2=0}^{p} C_{k_1k_2} e^{\frac{-\sqrt{2p+1}(t_i-t_j)}{l_T}} \int_{t_0-t_i}^{+\infty} e^{\frac{-2\sqrt{2p+1}t}{l_T}} P(t)dt \tag{77}$$

thanks to the linearity of the integral.

It is known (see [31]) that

$$\int P(x)e^{ax}dx = \frac{e^{ax}}{a}\sum_{k=0}^{m}(-1)^k\frac{P_m^{(k)}(x)}{a^k}$$

where $P_m$ is a polynomial of degree $m$ and $P_m^{(k)}$ is the $k$th derivative of $P_m$.

Therefore,

$$\int e^{\frac{-2\sqrt{2p+1}t}{l_T}}P(t)dt = -\frac{l_T}{2\sqrt{2p+1}}e^{-\frac{2\sqrt{2p+1}t}{l_T}}\sum_{k_3=0}^{2p-k_1-k_2}\frac{P^{(k_3)}(t)l_T^{k_3}}{\left(2\sqrt{2p+1}\right)^{k_3}} \tag{78}$$

Combining (77) and (78) we get

$$(k_T * k_T)_{t_0-t_i}^{+\infty}(t_i - t_j) = \sum_{k_1=0}^{p}\sum_{k_2=0}^{p}C_{k_1k_2}e^{\frac{-\sqrt{2p+1}(2t_0-t_i-t_j)}{l_T}}P_{k_1k_2}(t_0, t_i, t_j)$$

with $C_{k_1k_2}$ defined in (73) and $P_{k_1k_2}$ defined in (74).

This concludes the proof. $\qquad\square$

# E  Extension to Anisotropic Spatial Kernels

In this appendix, we illustrate how Theorem 4.1 could be extended to anisotropic spatial kernels by considering an Automatic Relevance Detection (ARD) Squared-Exponential (SE). It has the following form:

$$k_S(\boldsymbol{x}, \boldsymbol{y}) = e^{-\frac{1}{2}(\boldsymbol{x}-\boldsymbol{y})^\top \boldsymbol{M}^{-2}(\boldsymbol{x}-\boldsymbol{y})} \tag{79}$$

where $\boldsymbol{M} = \operatorname{diag}(l_1, \cdots, l_d)$ is a diagonal matrix that gathers a different lengthscale for each dimension. Observe that the isotropic SE with lengthscale $l_S$ is retrieved by setting $\boldsymbol{M} = l_S\boldsymbol{I}$.

Because the ARD SE kernel (79) is anisotropic, the convolution with itself

$$(k_S * k_S)(\boldsymbol{x} - \boldsymbol{y}) = \oint_{\mathbb{R}^d}k_S(\boldsymbol{x}, \boldsymbol{z})k_S(\boldsymbol{y}, \boldsymbol{z})d\boldsymbol{z} \tag{80}$$

cannot be simplified to a one-dimensional integral through a change to polar coordinates, as done in Corollary D.1. The integral becomes more complex, but can still be computed exactly for some kernel such as the ARD SE.

**Lemma E.1.** *Let $k_S$ be an ARD SE covariance function with parameter $\boldsymbol{M}$. Then,*

$$(k_S * k_S)(\boldsymbol{x} - \boldsymbol{y}) = \pi^{\frac{d}{2}}\det(\boldsymbol{M})e^{-\frac{1}{4}(\boldsymbol{x}-\boldsymbol{y})^\top \boldsymbol{M}^{-2}(\boldsymbol{x}-\boldsymbol{y})}. \tag{81}$$

*Proof.* For the ARD SE kernel with parameter $\boldsymbol{M} = \operatorname{diag}(l_1, \cdots, l_d)$, the convolution (80) is

$$\begin{aligned}
(k_S * k_S)(\boldsymbol{x} - \boldsymbol{y}) &= \oint_{\mathbb{R}^d}e^{-\frac{1}{2}(\boldsymbol{x}-\boldsymbol{z})^\top \boldsymbol{M}^{-2}(\boldsymbol{x}-\boldsymbol{z})}e^{-\frac{1}{2}(\boldsymbol{y}-\boldsymbol{z})^\top \boldsymbol{M}^{-2}(\boldsymbol{y}-\boldsymbol{z})}d\boldsymbol{z} \\
&= e^{-\frac{1}{2}\left(\boldsymbol{x}\boldsymbol{M}^{-2}\boldsymbol{x}+\boldsymbol{y}\boldsymbol{M}^{-2}\boldsymbol{y}\right)}\oint_{\mathbb{R}^d}e^{-\boldsymbol{z}^\top \boldsymbol{M}^{-2}\boldsymbol{z}+\boldsymbol{z}^\top \boldsymbol{M}^{-2}(\boldsymbol{x}+\boldsymbol{y})}d\boldsymbol{z} \\
&= e^{-\frac{1}{2}\left(\boldsymbol{x}\boldsymbol{M}^{-2}\boldsymbol{x}+\boldsymbol{y}\boldsymbol{M}^{-2}\boldsymbol{y}\right)}\oint_{\mathbb{R}^d}e^{\sum_{k=1}^d z_k(x_k+y_k-z_k)/l_k^2}dz_1\cdots dz_d \\
&= e^{-\frac{1}{2}\left(\boldsymbol{x}\boldsymbol{M}^{-2}\boldsymbol{x}+\boldsymbol{y}\boldsymbol{M}^{-2}\boldsymbol{y}\right)}\prod_{k=1}^d\int_{-\infty}^{+\infty}e^{z_k(x_k+y_k-z_k)/l_k^2}dz_k.
\end{aligned} \tag{82}$$

Integrating the $k$-th term in (82), we get

$$\begin{aligned}
\int_{-\infty}^{+\infty}e^{z_k(x_k+y_k-z_k)/l_k^2}dz_k &= \frac{1}{2}\sqrt{\pi}l_ke^{\frac{(x_k+y_k)^2}{4l_k^2}}\left[\operatorname{erf}\left(\frac{t}{l_k}+\frac{x_k+y_k}{2l_k}\right)\right]_{-\infty}^{+\infty} \\
&= \sqrt{\pi}l_ke^{\frac{(x_k+y_k)^2}{4l_k^2}}.
\end{aligned} \tag{83}$$

Injecting (83) into (82), we have

$$(k_S * k_S)(\boldsymbol{x} - \boldsymbol{y}) = e^{-\frac{1}{2}\left(\boldsymbol{x}\boldsymbol{M}^{-2}\boldsymbol{x} + \boldsymbol{y}\boldsymbol{M}^{-2}\boldsymbol{y}\right)} \prod_{k=1}^{d} \sqrt{\pi} l_k e^{\frac{(x_k + y_k)^2}{4l_k^2}}$$

$$= \pi^{\frac{d}{2}} \det(\boldsymbol{M}) e^{\frac{1}{4}(\boldsymbol{x}+\boldsymbol{y})\boldsymbol{M}^{-2}(\boldsymbol{x}+\boldsymbol{y}) - \frac{1}{2}\left(\boldsymbol{x}\boldsymbol{M}^{-2}\boldsymbol{x} + \boldsymbol{y}\boldsymbol{M}^{-2}\boldsymbol{y}\right)} \tag{84}$$

$$= \pi^{\frac{d}{2}} \det(\boldsymbol{M}) e^{-\frac{1}{4}(\boldsymbol{x}-\boldsymbol{y})^\top \boldsymbol{M}^{-2}(\boldsymbol{x}-\boldsymbol{y})} \tag{85}$$

where (84) holds because the determinant of a diagonal matrix is the product of its diagonal elements.
□

As a safety check, observe that Lemma D.2 is a special case of Lemma E.1 where $\boldsymbol{M} = l_S \boldsymbol{I}$, that is, when $k_S$ is an isotropic SE kernel.

# F    Relative Quantification of Relevancy

In this appendix, we discuss how (9) and its approximation (12) address the dependency on the covariance function hyperparameters introduced by (8). For the sake of this discussion, we take $k_S$ and $k_T$ as two Squared-Exponential (SE) covariance functions (see Table 1). A similar reasoning can be conducted with Matérn covariance functions.

Let us start by rewriting the product of spatial and temporal convolutions $C((\boldsymbol{x}, t), (\boldsymbol{x}', t'))$ with the formulas provided in Tables 3 and 4 for the SE covariance functions. We get

$$C((\boldsymbol{x}, t), (\boldsymbol{x}', t')) = \pi^{\frac{d}{2}} l_S^d e^{\frac{-||\boldsymbol{x}-\boldsymbol{x}'||_2^2}{4l_S^2}} \frac{\sqrt{\pi}}{2} l_T e^{\frac{-(t-t')^2}{2l_T^2}} \left(1 - \text{erf}\left(\frac{2t_0 - t - t'}{2l_T}\right)\right)$$

$$= \frac{1}{2} \pi^{\frac{d+1}{2}} l_S^d l_T e^{\frac{-||\boldsymbol{x}-\boldsymbol{x}'||_2^2}{4l_S^2} - \frac{-(t-t')^2}{2l_T^2}} \left(1 - \text{erf}\left(\frac{2t_0 - t - t'}{2l_T}\right)\right)$$

$$= \frac{1}{2} \pi^{\frac{d+1}{2}} l_S^d l_T C^*((\boldsymbol{x}, t), (\boldsymbol{x}', t')). \tag{86}$$

The dependency on the covariance function hyperparameters $\lambda, l_S, l_T$ appears clearly in (86). Both $l_S$ and $l_T$ are used, not only to scale the spatial distance $||\boldsymbol{x} - \boldsymbol{x}'||_2$ and the temporal distance $|t - t'|$ in $C^*$, but also as a scaling constant of the magnitude of the output of the product of convolutions itself. Because $C((\boldsymbol{x}, t), (\boldsymbol{x}', t'))$ is involved in every term of (10) and (11) in Theorem 4.1, $\frac{1}{2}\pi^{\frac{d+1}{2}} l_S^d l_T$ can be factored out of (10) and (11). Overall, both equations have in common the factor $\frac{1}{2}\pi^{\frac{d+1}{2}} \lambda^2 l_S^d l_T$. Clearly, this shows how the covariance function hyperparameters $\boldsymbol{\theta} = (\lambda, l_S, l_T)$ may control the magnitude of the Wasserstein distances.

To capture the intrinsic relevancy of an observation regardless of the hyperparameters values, one can compute (12), that is the ratio between (10) and (11). Doing so, the factors which are common to the two equations cancel out. Considering the application of Theorem 4.1 with $k_S$ and $k_T$ being SE covariance functions, the undesirable factor $\frac{1}{2}\pi^{\frac{d+1}{2}} \lambda^2 l_S^d l_T$ is removed. Clearly, (12) remains a function of $l_S$ and $l_T$, but the hyperparameters are only used to scale the spatial and temporal distances, that is to control correlation lengths. However, the undesirable scaling exposed in (86) no longer exists.

# G    Removal Budget

In this appendix, we discuss why the removal budget of W-DBO (see Algorithm 1), denoted $b_t$ at a given time $t$, has the form

$$\begin{cases} b_0 & = 1, \\ b_{t+\Delta t} & = b_t (1 + \alpha)^{\Delta t / l_T} \end{cases} \tag{87}$$

with $l_T$ the temporal lengthscale (see Assumption 3.2) and $\alpha$ the hyperparameter of W-DBO. Crucially, $l_T$ and $t$ must be expressed in the same unit of time.

Table 5: Comparison of removal budgets (87) and (88) when doing experiments of different durations on the Hartmann3d synthetic function. All experiments use the same time domain $[0, 1]$.

| Duration $D$ (seconds) | 1 Second (axis unit) | Lengthscale $l_T$ (axis unit) | Lengthscale $l_T$ (seconds) | Budget (87) $(1 + \alpha)^{D/l_T}$ | Budget (88) $(1 + \alpha)^D$ |
|---|---|---|---|---|---|
| 300 | 1/300 | 3/5 | 180 | $(1 + \alpha)^{5/3}$ | $(1 + \alpha)^{300}$ |
| 600 | 1/600 | 3/5 | 480 | $(1 + \alpha)^{5/3}$ | $(1 + \alpha)^{600}$ |
| 1800 | 1/1800 | 3/5 | 1080 | $(1 + \alpha)^{5/3}$ | $(1 + \alpha)^{1800}$ |

First, note that the expression of the budget is intuitive because (9) measures a ratio, expressed as a percentage. Therefore, the budget must accumulate in a multiplicative way, leading to the exponential form (87).

More interestingly, let us discuss the exponent $\Delta t/l_T$. Arguably, an alternative, more intuitive form of the removal budget would be

$$\begin{cases} b_0 & = 1, \\ b_{t+\Delta t} & = b_t(1 + \alpha)^{\Delta t} \end{cases}. \tag{88}$$

Although easier to understand, the budget (88) presents a major problem since it depends on arbitrary choices made by the user. This is illustrated by Table 5, where the same synthetic function (Hartmann3d) is optimized under three different durations. When the duration varies, the removal budget (88), which depends on the number of elapsed seconds only, also varies. Conversely, the budget (87) remains the same. This is because the number of temporal lengthscales elapsed during the experiment remains constant, regardless of the experiment duration.

Using the removal budget (88) becomes really troublesome when it comes to making a recommendation for the hyperparameter $\alpha$. If the analysis in Section 5.1 had used the budget (88), its recommendation $\alpha^*$ would have been a function of the temporal lengthscale, and it would have been valid only for experiments with the same duration (e.g., ten minutes). Any other experiment duration would have required another sensitivity analysis.

Conversely, the recommendation made in Section 5.1, using the budget (87), is a single number that is valid regardless of experiment duration. This is a much more general insight.

# H   Empirical Results

## H.1   Experimental Settings

In each experiment, the $d$-dimensional spatial domain is scaled in $\mathcal{S}' = [0, 1]^d$ and the temporal domain (viewed as the $(d + 1)$th dimension) is normalized in $[0, 1]$. Additionally, each optimization task lasts 600 seconds (10 minutes).

Unless stated otherwise, each DBO algorithm exploits a Matern-5/2 kernel as its spatial covariance function. GP-UCB, R-GP-UCB and ET-GP-UCB do not explicitly take into account temporal correlations, while TV-GP-UCB uses its own temporal covariance function. Eventually, ABO and W-DBO exploits a Matern-3/2 kernel as their temporal covariance function.

Each DBO algorithm begins its optimization task with 15 initial observations, uniformly sampled in $\mathcal{S}' \times \left[0, \frac{1}{40}\right]$. At each iteration (at time $t$), (i) the noise level as well as the kernel parameters are estimated, and (ii) the GP-UCB acquisition function is optimized to get the next query. The sum of the times taken to perform tasks (i) and (ii) is the *response time* of the DBO algorithm, denoted by $\Delta t$. Clearly, $\Delta t$ is a function of the dataset size of the DBO algorithm. Consequently, it varies throughout the optimization, getting larger when the DBO algorithm adds a new observation to its dataset, and getting smaller when the DBO algorithm removes at least one point. Once (i) and (ii) are performed, the objective function is immediately sampled (except for ABO which can decide to sample $f$ at a specific time in the future) and a Gaussian noise with variance equal to 5 % of the signal variance is added. Then, the next iteration begins at time $t + \Delta t$ (except for ABO if it decides to sample $f$ later).

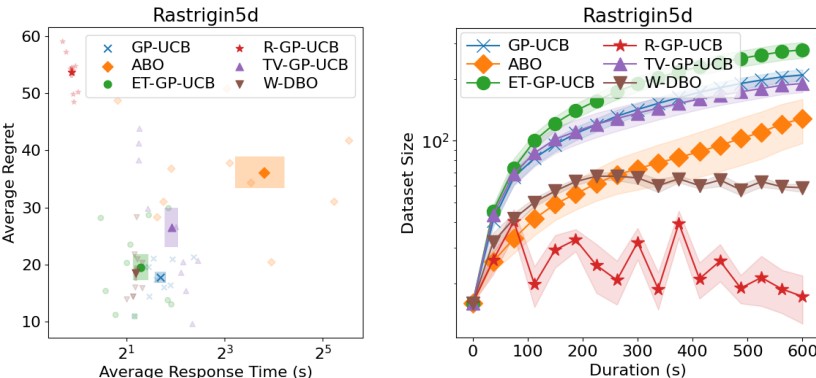

Figure 8: (Left) Average response time and average regrets of the DBO solutions during the optimization of the Rastrigin synthetic function. (Right) Dataset sizes of the DBO solutions during the optimization of the Rastrigin synthetic function.

For the sake of benchmarking fairness, all the solutions have been implemented using the same popular BO Python library, namely BOTorch [32] (MIT License). To comply with the technical choices (*i.e.*, Python front-end, C++ back-end), the computationally-heavy part of W-DBO (*i.e.*, the evaluation of the formulas in Section 4) have been implemented in C++ and bound to the Python code with PyBind11 [33] (BSD License). All experiments have been independently replicated 10 times on a laptop equipped with an Intel Core i9-9980HK @ 2.40 GHz with 8 cores (16 threads).

### H.2 Benchmarks and Figures

We provide here a detailed description of each implemented benchmark and the associated figures. There are two figures associated with each benchmark, showing their average regrets and the size of their datasets throughout the experiment.

In the following, the synthetic benchmarks will be described as functions of a point $z$ in the $d + 1$-dimensional spatio-temporal domain $\mathcal{S} \times \mathcal{T}$. More precisely, the point $z$ is explicitly given by $z = (x_1, \cdots, x_d, t)$. Also, we will write $d' = d + 1$ for the sake of brevity.

**Rastrigin.** The Rastrigin function is $d'$-dimensional, and has the form

$$f(z) = ad' + \sum_{i=1}^{d'} z_i^2 - a \cos\left(2\pi z_i\right).$$

For the numerical evaluation, we set $a = 10$, $d' = 5$ and we optimized the function on the domain $[-4, 4]^{d'}$. The results are provided in Figure 8.

**Schwefel.** The Schwefel function is $d'$-dimensional, and has the form

$$f(z) = 418.9829 d' - \sum_{i=1}^{d'} z_i \sin\left(\sqrt{|z_i|}\right).$$

For the numerical evaluation, we set $d' = 4$ and we optimized the function on the domain $[-500, 500]^{d'}$. The results are provided in Figure 9. This benchmark has also been used to replicate our results using the ARD covariance function studied in Appendix E. The results are provided in Figure 10.

**Styblinski-Tang.** The Syblinski-Tang function is $d'$-dimensional, and has the form

$$f(z) = \frac{1}{2} \sum_{i=1}^{d'} z_i^4 - 16 z_i^2 + 5 z_i.$$

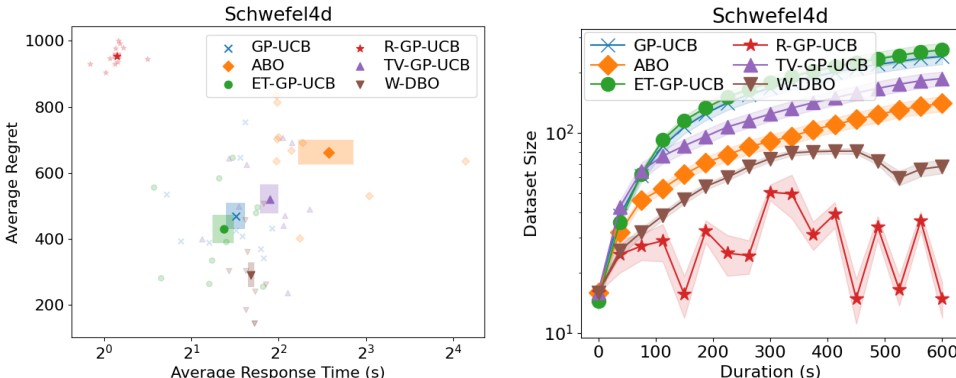

Figure 9: (Left) Average response time and average regrets of the DBO solutions during the optimization of the Schwefel synthetic function. (Right) Dataset sizes of the DBO solutions during the optimization of the Schwefel synthetic function.

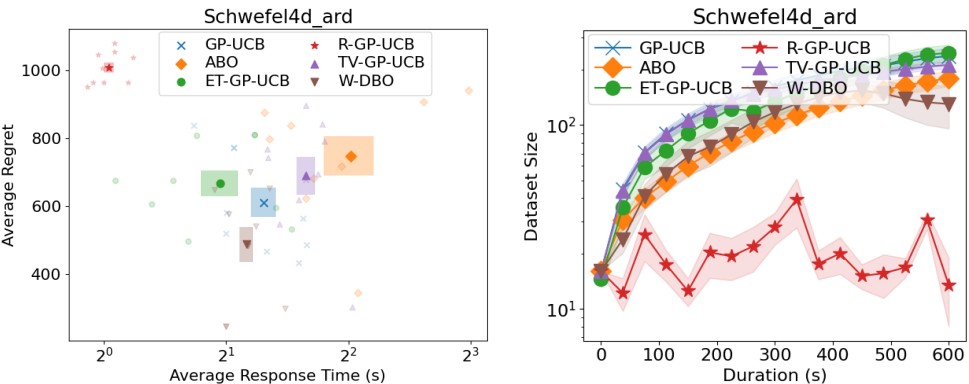

Figure 10: (Left) Average response time and average regrets of the DBO solutions using an ARD SE kernel during the optimization of the Schwefel synthetic function. (Right) Dataset sizes of the DBO solutions using an ARD SE kernel during the optimization of the Schwefel synthetic function.

For the numerical evaluation, we set $d' = 4$ and we optimized the function on the domain $[-5, 5]^{d'}$. The results are provided in Figure 11.

**Eggholder.** The Eggholder function is 2-dimensional, and has the form

$$f(\boldsymbol{z}) = -(z_2 + 47)\sin\left(\sqrt{\left|z_2 + \frac{z_1}{2} + 47\right|}\right) - z_1 \sin\left(\sqrt{|z_1 - z_2 - 47|}\right).$$

For the numerical evaluation, we optimized the function on the domain $[-512, 512]^2$. The results are provided in Figure 12.

**Ackley.** The Ackley function is $d'$-dimensional, and has the form

$$f(\boldsymbol{z}) = -a\exp\left(-b\sqrt{\frac{1}{d'}\sum_{i=1}^{d'} z_i^2}\right) - \exp\left(\frac{1}{d'}\sum_{i=1}^{d'}\cos(cz_i)\right) + a + \exp(1).$$

For the numerical evaluation, we set $a = 20$, $b = 0.2$, $c = 2\pi$, $d' = 4$ and we optimized the function on the domain $[-32, 32]^{d'}$. The results are provided in Figure 13. This benchmark has also been used to replicate our results using the ARD covariance function studied in Appendix E. The results are provided in Figure 14.

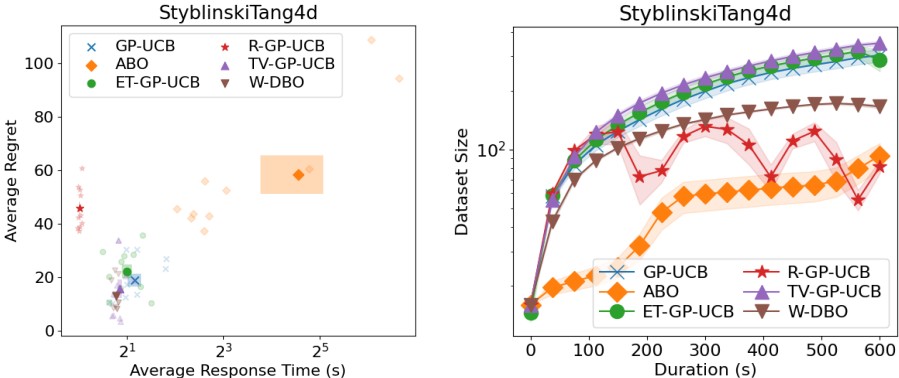

Figure 11: (Left) Average response time and average regrets of the DBO solutions during the optimization of the Styblinski-Tang synthetic function. (Right) Dataset sizes of the DBO solutions during the optimization of the Styblinski-Tang synthetic function.

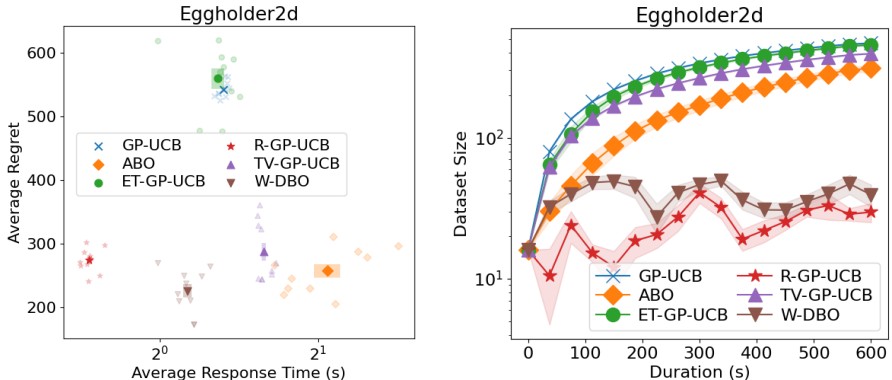

Figure 12: (Left) Average response time and average regrets of the DBO solutions during the optimization of the Eggholder synthetic function. (Right) Dataset sizes of the DBO solutions during the optimization of the Eggholder synthetic function.

**Rosenbrock.** The Rosenbrock function is $d'$-dimensional, and has the form

$$f(\boldsymbol{z}) = \sum_{i=1}^{d'-1} 100(z_{i+1} - z_i^2)^2 + (z_i - 1)^2.$$

For the numerical evaluation, we set $d' = 3$ and we optimized the function on the domain $[-1, 1.5]^{d'}$. The results are provided in Figure 15.

**Shekel.** The Shekel function is $4$-dimensional, and has the form

$$f(\boldsymbol{z}) = -\sum_{i=1}^{m} \left( \sum_{j=1}^{4} (z_j - C_{ji})^2 + \beta_i \right)^{-1}.$$

For the numerical evaluation, we set $m = 10$, $\boldsymbol{\beta} = \frac{1}{10}(1, 2, 2, 4, 4, 6, 3, 7, 5, 5)$,

$$\boldsymbol{C} = \begin{pmatrix} 4 & 1 & 8 & 6 & 3 & 2 & 5 & 8 & 6 & 7 \\ 4 & 1 & 8 & 6 & 7 & 9 & 3 & 1 & 2 & 3.6 \\ 4 & 1 & 8 & 6 & 3 & 2 & 5 & 8 & 6 & 7 \\ 4 & 1 & 8 & 6 & 7 & 9 & 3 & 1 & 2 & 3.6 \end{pmatrix},$$

and we optimized the function on the domain $[0, 10]^4$. The results are provided in Figure 16.

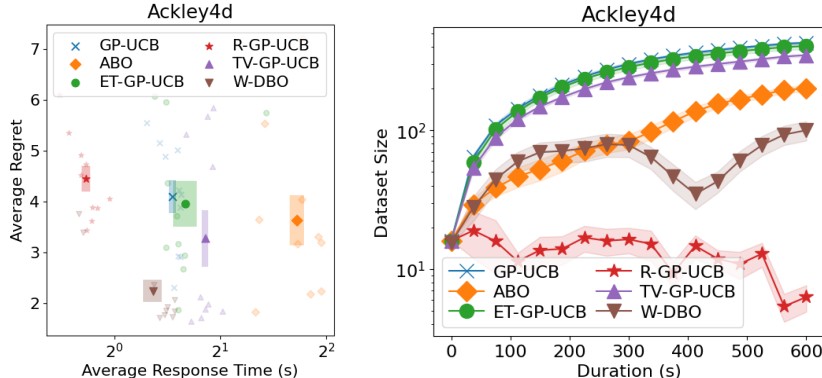

Figure 13: (Left) Average response time and average regrets of the DBO solutions during the optimization of the Ackley synthetic function. (Right) Dataset sizes of the DBO solutions during the optimization of the Ackley synthetic function.

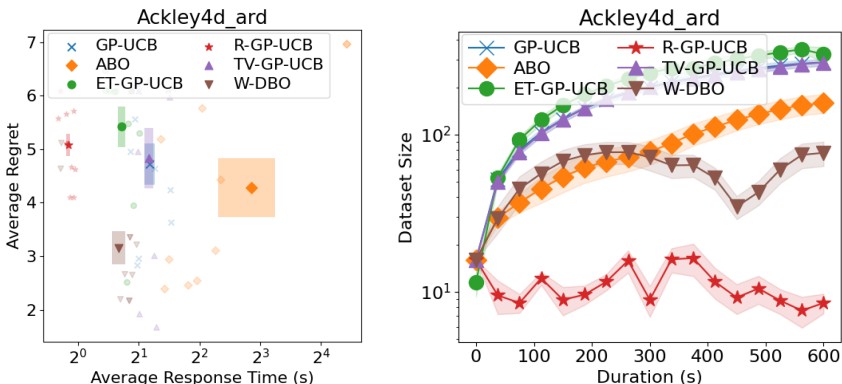

Figure 14: (Left) Average response time and average regrets of the DBO solutions using an ARD SE kernel during the optimization of the Ackley synthetic function. (Right) Dataset sizes of the DBO solutions using an ARD SE kernel during the optimization of the Ackley synthetic function.

**Hartmann-3.** The Hartmann-3 function is 3-dimensional, and has the form

$$f(\boldsymbol{z}) = -\sum_{i=1}^{4} \alpha_i \exp\left(-\sum_{j=1}^{3} A_{ij}(z_j - P_{ij})^2\right).$$

For the numerical evaluation, we set $\boldsymbol{\alpha} = (1.0, 1.2, 3.0, 3.2)$,

$$\boldsymbol{A} = \begin{pmatrix} 3 & 10 & 30 \\ 0.1 & 10 & 35 \\ 3 & 10 & 30 \\ 0.1 & 10 & 35 \end{pmatrix}, \boldsymbol{P} = 10^{-4} \begin{pmatrix} 3689 & 1170 & 2673 \\ 4699 & 4387 & 7470 \\ 1091 & 8732 & 5547 \\ 381 & 5743 & 8828 \end{pmatrix},$$

and we optimized the function on the domain $[0, 1]^3$. The results are provided in Figure 17.

**Hartmann-6.** The Hartmann-6 function is 6-dimensional, and has the form

$$f(\boldsymbol{z}) = -\sum_{i=1}^{4} \alpha_i \exp\left(-\sum_{j=1}^{6} A_{ij}(z_j - P_{ij})^2\right).$$

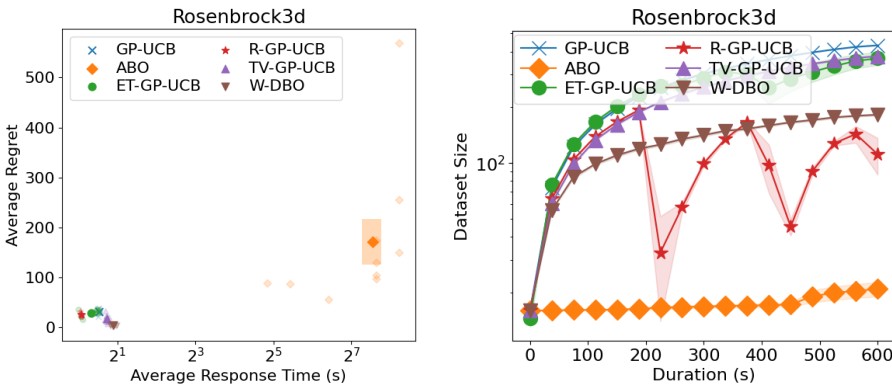

Figure 15: (Left) Average response time and average regrets of the DBO solutions during the optimization of the Rosenbrock synthetic function. (Right) Dataset sizes of the DBO solutions during the optimization of the Rosenbrock synthetic function.

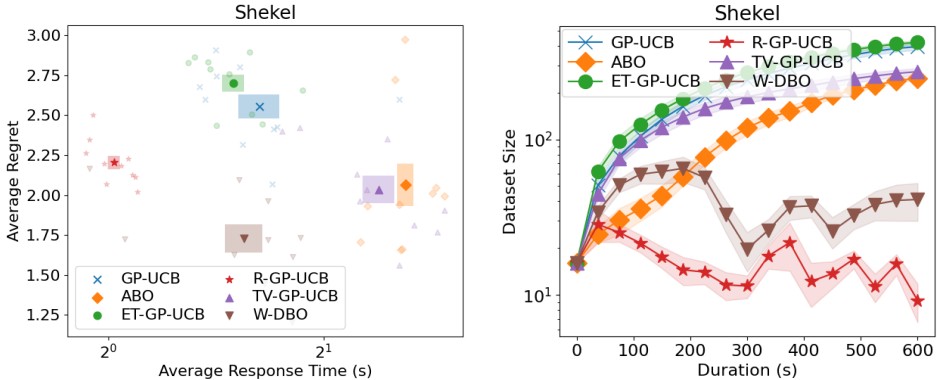

Figure 16: (Left) Average response time and average regrets of the DBO solutions during the optimization of the Shekel synthetic function. (Right) Dataset sizes of the DBO solutions during the optimization of the Shekel synthetic function.

For the numerical evaluation, we set $\boldsymbol{\alpha} = (1.0, 1.2, 3.0, 3.2)$,

$$\boldsymbol{A} = \begin{pmatrix} 10 & 3 & 17 & 3.50 & 1.7 & 8 \\ 0.05 & 10 & 17 & 0.1 & 8 & 14 \\ 3 & 3.5 & 1.7 & 10 & 17 & 8 \\ 17 & 8 & 0.05 & 10 & 0.1 & 14 \end{pmatrix}, \boldsymbol{P} = 10^{-4} \begin{pmatrix} 1312 & 1696 & 5569 & 124 & 8283 & 5886 \\ 2329 & 4135 & 8307 & 3736 & 1004 & 9991 \\ 2348 & 1451 & 3522 & 2883 & 3047 & 6650 \\ 4047 & 8828 & 8732 & 5743 & 1091 & 381 \end{pmatrix},$$

and we optimized the function on the domain $[0, 1]^6$. The results are provided in Figure 18.

**Powell.** The Powell function is $d'$-dimensional, and has the form

$$f(\boldsymbol{z}) = \sum_{i=1}^{d'/4} (z_{4i-3} + 10z_{4i-2})^2 + 5(z_{4i-1} - z_{4i})^2 + (z_{4i-2} - 2z_{4i-1})^4 + 10(z_{4i-3} - z_{4i})^4.$$

For the numerical evaluation, we set $d' = 4$ and we optimized the function on the domain $[-4, 5]^{d'}$. The results are provided in Figure 19.

**Temperature.** This benchmark comes from the temperature dataset collected from 46 sensors deployed at Intel Research Berkeley. It is a famous benchmark, used in other works such as [21, 22]. The goal of the DBO task is to activate the sensor with the highest temperature, which will vary with time. To make the benchmark more interesting, we interpolate the data in space-time. With this interpolation, the algorithms can activate any point in space-time, making it a 3-dimensional benchmark (2 spatial dimensions for a location in Intel Research Berkeley, 1 temporal dimension).

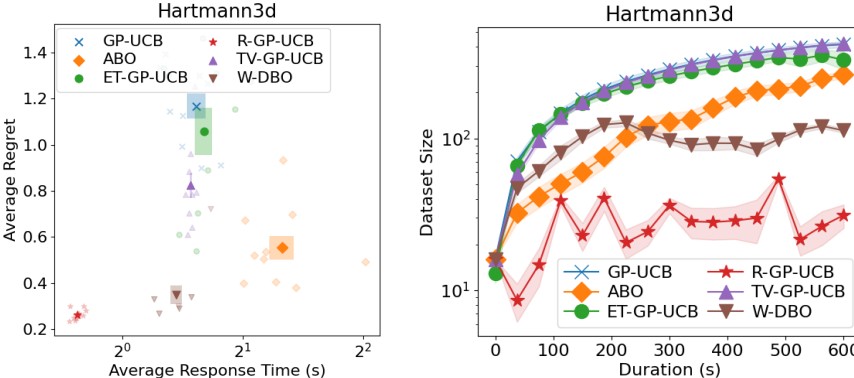

Figure 17: (Left) Average response time and average regrets of the DBO solutions during the optimization of the Hartmann-3 synthetic function. (Right) Dataset sizes of the DBO solutions during the optimization of the Hartmann-3 synthetic function.

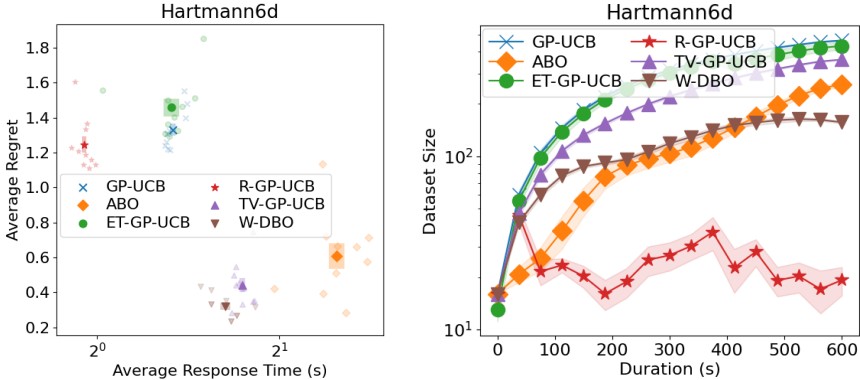

Figure 18: (Left) Average response time and average regrets of the DBO solutions during the optimization of the Hartmann-6 synthetic function. (Right) Dataset sizes of the DBO solutions during the optimization of the Hartmann-6 synthetic function.

For the numerical evaluation, we used the first day of data. The results are provided in Figure 20.

**WLAN.** This benchmark aims at maximizing the throughput of a Wireless Local Area Network (WLAN). 18 moving end-users are associated with one of 4 fixed nodes and continuously stream a large amount of data. As they move in space, they change the radio environment of the network, which should adapt accordingly to improve its performance. To do so, each node has a power level that can be tuned for the purpose of reaching the best trade-off between serving all its users and not causing interference for the neighboring nodes.

The performance of the network is computed as the sum of the Shannon capacities for each pair of node and associated end-users. The Shannon capacity [34] sets a theoretical upper bound on the throughput of a wireless communication. We denote it $C(i, j)$, we express it in bits per second (bps). It depends on $S_{ij}$ the Signal-to-Interference plus Noise Ratio (SINR) of the communication between node $i$ and end-user $j$, as well as on $W$, the bandwidth of the radio channel (in Hz):

$$C_{ij}(\boldsymbol{x}, t) = W \log_2(1 + S_{ij}(\boldsymbol{x}, t)).$$

Then, the objective function is

$$f(\boldsymbol{x}, t) = \sum_{i=1}^{4} \sum_{j \in \mathcal{N}_i} C_{ij}(\boldsymbol{x}, t),$$

with $\mathcal{N}_i$ the end-users associated with node $i$.

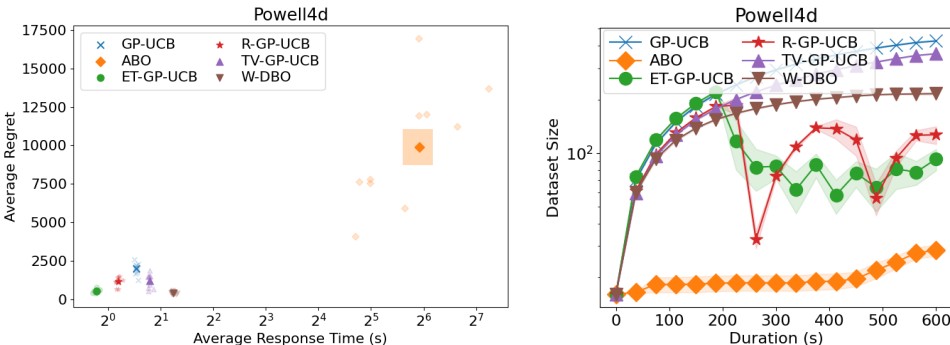

Figure 19: (Left) Average response time and average regrets of the DBO solutions during the optimization of the Powell synthetic function. (Right) Dataset sizes of the DBO solutions during the optimization of the Powell synthetic function.

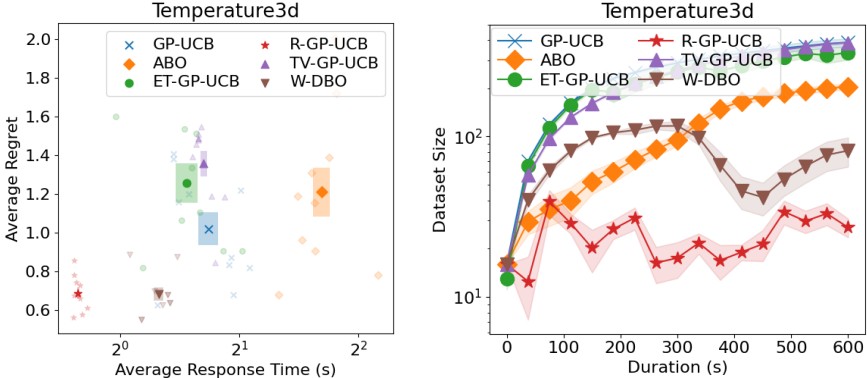

Figure 20: (Left) Average response time and average regrets of the DBO solutions during the Temperature real-world experiment. (Right) Dataset sizes of the DBO solutions during the Temperature real-world experiment.

For the numerical evaluation, we optimized the power levels $x$ in the domain $[10^{0.1}, 10^{2.5}]^4$. For this experiment, the DBO solutions were evaluated with a Matérn-5/2 for the spatial covariance function and a Matérn-1/2 for the temporal covariance function. The results are provided in Figure 21.

### H.3 Discussion on Empirical Performance

In this section, we discuss the performance achieved by all the DBO solutions on the benchmarks introduced in the previous section.

**GP-UCB.** This baseline, which does not take into account temporal correlations, obtains surprisingly good performance in this experimental setting (continuous time, hyperparameters estimated on the fly). Its simple behavior (*i.e.*, keep all observations in the dataset until the end of the experiment) hampers its response time, but this drawback is balanced by the fact that it has only three parameters to estimate with MLE (*i.e.*, $\lambda, l_S, \sigma^2$). Overall, it is dominated by R-GP-UCB, TV-GP-UCB and W-DBO, but behaves surprisingly well against ABO and ET-GP-UCB (see Figure 5).

**ABO.** ABO performs poorly in this experimental setting. We explain this poor performance by the fact that the hyperparameters (including the spatial and temporal lengthscales) have to be estimated on the fly. Since ABO can decide to postpone its next query to the near future (a fraction of the temporal lengthscale $l_T$ away), overestimating $l_T$ may cause ABO to wait for a long time before querying $f$ again. This interpretation is supported by the fact that the functions ABO performs the poorest on are the ones with the largest temporal lengthscales $l_T$, e.g., Rosenbrock (see Figure 15) and Powell (see Figure 19). Conversely, ABO obtains competitive performance on functions with

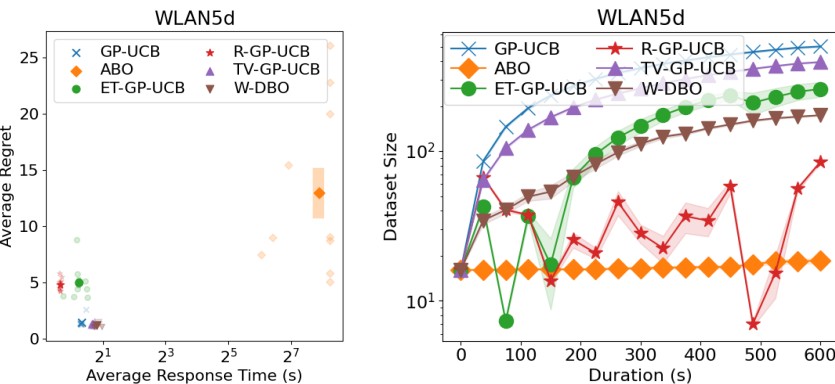

Figure 21: (Left) Average response time and average regrets of the DBO solutions during the WLAN real-world experiment. (Right) Dataset sizes of the DBO solutions during the WLAN real-world experiment.

smaller temporal lengthscales, e.g. Eggholder (see Figure 12) or Shekel (see Figure 16). These results highlight the lack of robustness of ABO.

**ET-GP-UCB.** Like GP-UCB, ET-GP-UCB does not take into account temporal correlations, and deals with stale observations by resetting its dataset each time a condition is met. Our experimental setting exposes the lack of robustness of ET-GP-UCB, since its performance is quite poor on most benchmarks. This is mainly due to the fact that, because the hyperparameters (including the observational noise level $\sigma^2$) are inferred on the fly, the MLE explains the variance in the observations with an increasingly large observational noise level $\sigma^2$ as time goes by. However, by construction of ET-GP-UCB, the greater $\sigma^2$, the less the dataset will be reset. As a consequence, on some benchmarks, the event is never triggered (or not triggered enough) and the performance of ET-GP-UCB is close to (sometimes worse than) the performance of GP-UCB. For examples, refer to Hartmann6d (see Figure 18), Shekel (see Figure 16) or Ackley (see Figure 13). Some other times, the variance in the observations cannot be explained by an increasingly large observational noise. In these cases, the triggering occurs properly and ET-GP-UCB obtains competitive performance, e.g., with Powell (see Figure 19).

**R-GP-UCB.** R-GP-UCB deals with stale data by resetting its dataset (like ET-GP-UCB). It indirectly accounts for temporal correlations by estimating an hyperparameter $\epsilon$, and the reset is triggered each time the dataset size exceeds $N(\epsilon)$ (given in [21]). More often than not, its performance is better than GP-UCB, because stale data is frequently removed from the dataset. As a consequence, R-GP-UCB has the lowest average response time of all the DBO solutions. Overall, because of its low response time, R-GP-UCB obtains very good performance on some benchmarks, e.g., Eggholder (see Figure 12), Hartmann3d (see Figure 17) or Temperature (see Figure 20).

**TV-GP-UCB.** TV-GP-UCB directly accounts for temporal correlations by computing a specific covariance function controlled by an hyperparameter $\epsilon$. Although the temporal covariance function is based on a distance between indices instead of a distance between points in time, the DBO solution turns out to be quite robust in our experimental setting. However, its response time is hampered by the irrelevant observations that are kept in the dataset. Because of them, TV-GP-UCB has one of the largest response time on many benchmarks, e.g., Rastrigin (see Figure 8), Schwefel (see Figure 9) or Shekel (see Figure 16). Nevertheless, its average performance is significantly better than the other state-of-the-art DBO solutions.

**W-DBO.** Because of its ability to measure the relevancy of its observations and to remove irrelevant observations, W-DBO achieves simultaneously good predictive performance and a low response time. Depending on the benchmark, its dataset size follows different patterns. When the objective function evolves smoothly, e.g., Powell (see Figure 19) or Rosenbrock (see Figure 15), W-DBO behaves roughly like GP-UCB and TV-GP-UCB and keeps most of its observations in its dataset (although it manages to identify and delete some irrelevant observations). When the objective function's

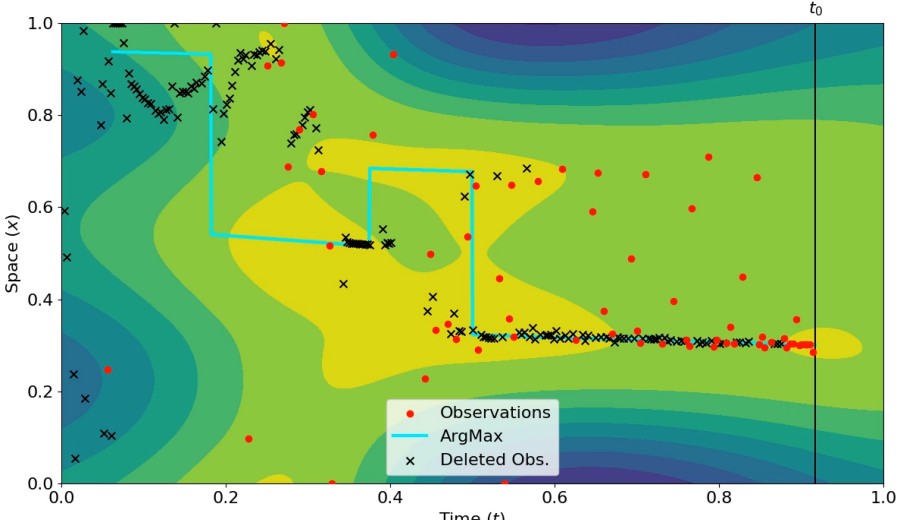

Figure 22: Snapshot from one of the videos showing the optimization conducted by W-DBO. The normalized temporal dimension is shown on the x-axis and the normalized spatial dimension is shown on the y-axis. The observations that are in the dataset are depicted as red dots, while the deleted observations are depicted as black crosses. The maximal arguments $\{\arg\max_{x \in S} f(x, t), t \in \mathcal{T}\}$ are depicted with a cyan curve. The predictions of W-DBO are shown with a contour plot. Finally, the present time is depicted as a black vertical line labelled $t_0$.

variations are more pronounced, the dataset size of W-DBO experiences sudden drops, as can be seen with Ackley (see Figure 13), Shekel (see Figure 16) or Temperature (see Figure 20). This suggests that W-DBO is also able to "reset" its dataset, although in a more refined way as it is able to keep the few observations still relevant for future predictions. Thanks to its ability to adapt in very different contexts, W-DBO outperforms state-of-the-art DBO solutions by a comfortable margin. This performance gap can be seen in its average performance across all benchmarks (see Figure 5), but also on most of the benchmarks themselves, e.g., Schwefel (see Figure 9), Ackley (see Figure 13), Shekel (see Figure 16), Hartmann-6 (see Figure 18) or Powell (see Figure 19).

## H.4 Animated Visualizations

In this section, we describe and discuss the two animated visualizations provided as supplementary material for the paper. These videos show W-DBO optimizing two 2-dimensional synthetic functions. They depict W-DBO's predictions, collected observations and deleted observations into the spatio-temporal domains of the functions.

One of the videos depict the optimization of the Six-Hump Camel function[4] on the domain $[-2, 2]^2$. The SHC function is

$$\text{SHC}(x, t) = \left(4 - 2.1x^2 + \frac{x^4}{3}\right)x^2 + xt + \left(-4 + 4t^2\right)t^2.$$

To study how W-DBO reacts to sudden changes in the objective function, the other video depict the optimization of the piecewise function

$$f(x, t) = \begin{cases} \text{SHC}(x, t) & \text{if } t < -\frac{1}{2}, \\ \text{SHC}(t, x) & \text{otherwise.} \end{cases}$$

A snapshot from the latter can be found in Figure 22. It illustrates that the benefits brought by W-DBO are substantial, since the algorithm is able to track $\max_{x \in S} f(x, t)$ over the time $t$ while

---

[4]The video is accessible at `https://abardou.github.io/assets/vid/PermSix-Hump_Camel_25.0_240.mp4`

simultaneously deleting a significant portion of collected observations. Indeed, many observations are deemed irrelevant, either because (i) they have become stale (there are only a few observations collected at the start of the experiment that have been kept in the dataset) or because (ii) they are redundant with observations that are already in the dataset (many observations are located near the maximal argument, and many of them are deleted soon after being collected).

# I  Limitations

For the sake of completeness, we explicitly discuss the limitations of W-DBO in this appendix. Four limitations were identified:

- As for any BO algorithm, W-DBO exploits a GP as a surrogate model (see Assumption 3.1). If the objective function $f$ cannot be properly approximated by a GP, we expect the performance of W-DBO to decline.

- As for any BO algorithm, W-DBO conducts GP inference, which causes it to manipulate inverses of Gram matrices that scale with the dataset size. Although the main motivation of introducing W-DBO is to reduce the dataset size, the cubic complexity of matrix inversion algorithms can still constitute a limitation if too many observations are kept in the dataset.

- We also introduce a structure for spatio-temporal correlations with Assumption 3.2. Although less restrictive than the one enforced by [21, 22], equivalent to the one in [20] and partially relaxed in Appendix E, this is still a limitation since we expect the performance of W-DBO to worsen if the objective function does not meet this assumption.

- Finally, W-DBO is not exempt from the effects of the sampling frequency. In fact, as for any DBO algorithm, the performance of W-DBO will drop if the function varies too much between observations. As an example, if $f$ evolves so rapidly that two successive observations become basically independent, W-DBO will not be able to infer anything meaningful about the objective function.

