# OpenReview forum: "This Too Shall Pass: Removing Stale Observations in Dynamic Bayesian Optimization"
_NeurIPS.cc/2024/Conference — NeurIPS 2024 poster_

### Official Review · Reviewer_NVX3 · 2024-06-18

**Soundness:** 3
**Presentation:** 3
**Contribution:** 3
**Rating:** 7
**Confidence:** 3

**Summary:**

The paper proposes a novel algorithm for dynamic Bayesian optimization (DBO). DBO defers from traditional BayesOpt as the black-box function to be optimized is changing in time, and the goal of the optimization procedure is to keep track of the optimum across a continuous time index. The continuous time component adds an interesting dimension to the problem: the black-box sampling frequency becomes important to be able to query the function frequently enough to keep track of the optimum. This means there is a necessity for the algorithm and acquisition function solve to be as efficient as possible.

One important important factor in the acquisition function solve time is the training of the GP surrogate model. In particular, the training is of order $O(N^3)$ on the number of training data-points. In a dynamical system where the function evolves with time, certain training-points can become redundant, meaning we can make the optimization more efficient if we discard them. The paper proposes a method to choose which points are worth discarding.

The main intuition behind the method, is that a point which does not change the *future* predictive distribution of the GP (with respect to time) is a point that is no longer needed in the data-set. Therefore the authors propose a criterion that looks at the predictive distribution under the full data-set $D$ and measures the 2-Wasserstein distance to a GP trained on $D \backslash (x_i, y_i)$ where $(x_i, y_i)$ is training point that could potentially be removed. To make the measure interpretable (in terms of the magnitude of the criterion), we can normalize by looking at the 2-Wasserstein distance between the full data-set and the prior GP, giving the final criterion. Unfortunately, actually calculating the criterion is too computationally expensive, especially when trying to reduce costs to have a high sampling frequency. The paper then shows that both 2-Wasserstein distances can be upper-bounded, and proposes using the upper-bounds to replace the actual distance, creating an approximate criterion.

In the appendix, the magnitude of the error of the approximation is investigated in practice. While it is shown to be non-negligible, the division in the criterion helps balance out the error in part, and most importantly, the empirical results of using the criterion are strong. Once the criterion is established, the paper introduces the algorithm: before every acquisition function solve, points are removed from the data-set in a greedy manner, i.e., the point with the least relevancy is compared against a budget, and if the budget allows it, then the points is removed and the budget is decreased, this is repeated until the budget is small enough. The budget is defined by the amount of relative error allowed by removing data-points.

Numerical results show that a trade-off between removing enough information to be computationally tractable, while keeping enough data to identify the optimum is required, and through empirics the authors are able to provide a recommended hyper-parameter value. Then the algorithms are tested on 10 synthetic functions and 2 real-world experiments. The proposed algorithm is the strongest in 10 of the 12 benchmarks.

**Strengths:**

- (Originality) The paper proposes a novel algorithm in an understudied area of Bayesian optimization. The paper provides a strong theoretical backing of the method and a good empirical study. Some important literature connections seem to have been missed though.

- (Quality) All claims in the paper seem backed-up and well justified. Details of the method are well reported, and important parts of the algorithm, such as sensitivity to hyper-parameters and approximation error are investigated.

- (Clarity) The paper is exceptionally written, and the method clear to understand.

- (Significance) As mentioned in the paper, Dynamical Bayesian optimization is an understudied area, however I can think of many possible applications of it. I believe the method to be potentially significant in many applications.

**Weaknesses:**

- Perhaps the biggest weakness is a lack of comparison, and overall lack of discussion of sparse Gaussian processes and their relationship to online learning. Indeed, methods for selecting inducing points share similarities with the proposed algorithm, e.g. by greedily making the data-set smaller based on information criterions (see Section 9 of [1]) and even with application to online learning and BO [2, 3, 4] (the final reference is very recent, so it was impossible for the authors to include, but it looks relevant). Such methods seem to have the potential of performing well in this setting, and if possible it should be compared against them, but at least they should be discussed.

- The proposed approximation of the 2-Wasserstein distance criterion is potentially loose, however, the authors recognize this and the empirical algorithmic performance is strong.

- A lot of the synthetic testing is carried out in non-dynamical benchmarks. While the importance of the method is that it is able to maintain a small data-set and still carry out BO well, I believe a larger pool of experiments that includes the main application domain is important.

[1] Quinonero-Candela, Joaquin, and Carl Edward Rasmussen. "A unifying view of sparse approximate Gaussian process regression." The Journal of Machine Learning Research 6 (2005): 1939-1959.

[2] Galy-Fajou, Théo, and Manfred Opper. "Adaptive inducing points selection for gaussian processes." arXiv preprint arXiv:2107.10066 (2021).

[3] Moss, Henry B., Sebastian W. Ober, and Victor Picheny. "Inducing point allocation for sparse Gaussian processes in high-throughput Bayesian optimisation." International Conference on Artificial Intelligence and Statistics. PMLR, 2023.

[4] Maus, Natalie, et al. "Approximation-Aware Bayesian Optimization." arXiv preprint arXiv:2406.04308 (2024).

**Questions:**

See weaknesses

**Limitations:**

Limitations are discussed at length in Appendix C and H.

---

> ### Author Rebuttal · Authors · 2024-08-06
>
> Dear Reviewer NVX3,
>
> Thank you for the detailed review. We are discussing below the weaknesses and questions you have raised. Also, please make sure to read the global response as we discuss some of your questions there.
>
> **Lack of discussion of sparse Gaussian processes.**
>
> Thank you for pointing out this very interesting connection between the sparse GP literature and our work.
>
> Although these works are definitely related to our problem (because they also seek to discard some observations while still preserving the quality of the GP inference), applying them to a dynamic setting would require some non-trivial modifications. More precisely,
>
> (i) In their current form, these works introduce solutions that take place into a spatial domain only. Consequently, they try to approximate an exact GP over a bounded domain, and assume that they can place inducing inputs arbitrarily in this very same domain. Conversely, in the dynamic setting, the domain is constantly changing and the domain where the inducing inputs can be located is restricted to the cartesian product of the spatial domain and the past $[0, t_0]$, with $t_0$ the present time. Moreover, the exact GP should be approximated on a different, complementary, unbounded domain, that is the cartesian product between the spatial domain and the future $[t_0, +\infty]$.
>
> (ii) Most sparse GP solutions have an hyperparameter that controls the number of inducing inputs to place. Unfortunately, this hyperparameter should be set specifically for each objective function. Conversely, W-DBO can adapt its dataset size depending on the nature of the objective function, without the need to adjust its hyperparameter.
>
> Still, we acknowledge the connections described by the reviewer. Upon acceptance, we will definitely discuss the mentioned papers in Section 2 of the camera-ready version.
>
> **The proposed approximation of the 2-Wasserstein distance criterion is potentially loose.**
>
> We plan to study more the approximation of the ratio of Wasserstein distances in future works, in order to better understand it theoretically.

---

> > ### Comment · Reviewer_NVX3 · 2024-08-07
> >
> > Thanks for your response. I agree that the sparse GP methodology is relevant but cannot be applied in the current version of the papers and non-trivial work would need to be carried out. Adding a discussion of these challenges and the connections with the current problem would be sufficient.
> >
> > Thanks further for the clarification about the synthetic benchmarks being dynamic, I do agree it should be made clearer.
> >
> > I remain positive about the work, and keep my recommendation of acceptance.

---

### Official Review · Reviewer_iNzn · 2024-07-04

**Soundness:** 1
**Presentation:** 2
**Contribution:** 2
**Rating:** 3
**Confidence:** 5

**Summary:**

The authors propose W-DBO, a statistical distance-based criterion for removing "stale" observations in the dynamic BO setting. Observations are removed based on their impact on the GP globally, as measured by an approximate integrated Wasserstein distance, for which the authors prove the approximation quality. Results show that w-DBO outperforms the competition in terms of regret over time.

**Strengths:**

__Relevant problem__: The optimization of time-varying functions appears highly relevant, and seems underexplored.

__Intuitive solution__: Measuring staleness by the impact on the GP globally is a good idea.

__Theory__: There is a good amount of theory included on the proposed algorithm.

**Weaknesses:**

Unfortunately, I have a substantial concern about the evaluation, which I believe requires the authors to (ideally) release code of their experimental setup and unfortunately, re-run experiments with a more conventional setup.

__Benchmarking__: The authors do not appear to use an ARD kernel in their work. As far as I can tell, the method is not restricted to a single lengthscale for all dimensions, so is there a reason for this unconventional choice?

Notably, using a non-ARD kernel is an issue of fairness in benchmarking. Since almost all test functions (Rastrigin, Schwefel, Shekel, Ackley, Styblinski-Tang and to a lesser degree Rosenbrock), that are chosen are symmetric, this choice is rather convenient. Specifically, as soon as one lengthscale is estimated correctly, all of them will be, which is _very_ beneficial for performance. For one, ABO estimates all of the lengthscales, and will as such be as a large disadvantage. The authors mention that standard GP-UCB is non-ARD as well (App. G3), which could very well be the explanation for its good performance on the symmetric functions. To emphasize this even more, the obvious non-symmetric function (Hartmann-3) is one one where W-DBO and GP-UCB perform worse. (Sidenote: Hartmann-6 is fairly uniform in its lengthscales.)

In short: Can the authors please run their experiments with an ARD kernel on the symmetric test functions? Right now, the non-ARD provides a alrge advantage, specifically in relation to the algorithms that run ARD (ABO at least). I do not believe running a non-ARD kernel is warranted, since it is not a design choice that is relevant to the dynamic setting and provides an outsized and unrealistic advantage.

__Relevance of "stale" observations__: It is not apparent to me that "stale" observations are a large problem if there is a time-varying component to the kernel which naturally decreases its relevance with time. As such, the algorithm seems to strictly target high-throughput settings due to the computational saving, which is more restrictive setting than dynamic BO generally.

__Evidence for removing stale observations__: While the criterion makes sense, I fail to see evidence that the removed observations are indeed "stale". What is good evidence for this? Simply plotting downstream regret performance (especially when the benchmarking has other flaws) is not a good metric for staleness.

An Illustrative example of which observations are being removed are a must. However, I also encourage the authors to assess what a good metric for staleness is, and present that in their results.

__Benchmarking, part 2__: The authors propose a dynamic BO algorithm, but the synthetic benchmarks appear to all be non-dynamic (there is no time-dependence in any of the benchmarks in App. G2). I would have expected these to have a time-varying component in order for the algorithm to be relevant. If the benchmarks indeed do not have a time-varying component, shouldn't standard GP-UCB be the gold standard for performance?

__Benchmarking, part 3__: Please add standard errors to the tables.

__Content in Appendices__: There is a lot of important content that has been moved to the supplementary material, most importantly the variable definitions in 4.1 (App. A & B), intuition for the Wasserstein ratio in 3.2 (App. E) . I believe the authors should try and make room for these important pieces in the main paper, possibly by shortening the background section or moving some of it to SM.

__Normalization by empty GP__: The idea is novel and appears intuitive, but it is not thoroughly explained insofar as why the metric is appropriate. I recommend the authors to elaborate and add a 1D illustrative example, possibly compared to simply using the Wasserstein distance.

Minor:
- "BO inference" is used numerous times in place of "GP inference', which is the more precise description of what is occuring.
- L160: "uses" -> "used"

**Questions:**

Questions are apparent from the "Weaknesses" section.

**Limitations:**

Limitations have been addressed.

---

> ### Author Rebuttal · Authors · 2024-08-06
>
> Dear Reviewer iNzn,
>
> Thank you for the detailed review. We are discussing below the weaknesses and questions you have raised. Also, please make sure to read the global response as we discuss some of your questions there.
>
> **On the usage of an ARD kernel.**
>
> We make no mention of setting an ARD kernel for ABO, nor for any other DBO algorithms. Appendix G.1 lists the kernels used for each solution. Let us detail precisely the number of hyperparameters that each solution has to infer:
>
> _GP-UCB and ET-GP-UCB_.
> * Spatial cov.: Matérn-5/2
> * Temporal cov.: N/A
> * Number of parameters: 3 (scale $\lambda$, spatial lengthscale $l_S$, noise level $\sigma^2_0$)
>
> _R-GP-UCB and TV-GP-UCB_.
> * Spatial cov.: Matérn-5/2
> * Temporal cov.: fixed kernel with decay parameter $\epsilon$
> * Number of parameters: 4 (scale $\lambda$, spatial lengthscale $l_S$, decay $\epsilon$, noise level $\sigma^2_0$)
>
> _ABO and W-DBO_.
> * Spatial cov.: Matérn-5/2
> * Temporal cov.: Matérn-3/2
> * Number of parameters: 4 (scale $\lambda$, spatial lengthscale $l_S$, temporal lengthscale $l_T$, noise level $\sigma^2_0$)
>
> Although we acknowledge the usefulness of anisotropic kernels in practice, we would like to address (i) the issue of “conventionality” of ARD kernels in benchmarking, and (ii) the performance of W-DBO with an ARD kernel.
>
> (i) Among the DBO papers (*i.e.*, [1, 2, 3]), the vast majority of the benchmarking was done with a non-ARD kernel. More specifically, [1] explicitly uses a non-ARD kernel, [2] uses an ARD kernel on 1 experiment out of 7 (cf. Tables 4-10 in [2]), and [3] makes no mention of an ARD kernel. This makes us consider that using a non-ARD kernel for benchmarking is fairly standard, at least in the DBO literature. We do not understand therefore why the comprehensive benchmarking would be unconventional and/or unfair towards some solutions. Upon acceptance, we propose to explicitly list the kernel hyperparameters for each DBO solution in Appendix G.1 to prevent any confusion about our benchmarking.
>
> (ii) Let us discuss how our framework can be applied to anisotropic kernels. Most of our analysis remains unchanged, but we need additional convolution formulas because the ones provided in Tables 3 and 4 only hold under Assumption 3.2. We now illustrate this point by providing additional results with the anisotropic SE kernel. Consider a $d \times d$ diagonal matrix $\Sigma = \text{diag}\left(l_1, \cdots, l_d\right)$ that gathers the $d$ lengthscales in its diagonal. It can be shown that the anisotropic convolution formula is $(k_S * k_S)(\mathbf x) = \pi^{d / 2} \det\left(\Sigma\right) e^{-\mathbf x^\top \Sigma^{-2} \mathbf x / 4}$. Observe that the formula reduces to the isotropic formula described in Table 3 when $l_1 = \cdots = l_d = l_S$. Additional experiments with the ARD SE kernel are listed on the PDF, which had to be uploaded in the global rebuttal (where they are discussed too).
>
> **I fail to see that the removed observations are stale. Provide an illustrative example of which observations are removed and a metric for staleness.**
>
> The concept of staleness is used in the title, as well as in Sections 1 and 2 to introduce our work because it is a term commonly used in the literature, but we replace it with the broader concept of “relevancy” from Section 3 onwards. Relevancy can be temporal (an observation that is too distant in time to be relevant for GP inference, a.k.a staleness) but also spatial (an observation that is too close from another to bring substantial information for GP inference). In other words, staleness coincides with temporal relevancy, whereas W-DBO accounts for both temporal and spatial relevancy. We provide a metric for the relevancy of an observation $\mathbf x_i$, namely the Wasserstein distance between the posterior GP conditioned on the entire dataset $\mathcal{D}$ and the posterior GP conditioned on $\tilde{\mathcal{D}} = \mathcal{D} \setminus \\{\mathbf x_i\\}$, defined on the cartesian product of the spatial domain $\mathcal{S}$ and the future $[t_0, +\infty]$, with $t_0$ the present time.
>
> We use this metric because it is intuitive and it captures properly the concept of observation relevancy (e.g., it is 0 when removing an observation does not change the GP posterior in any way, and it can be arbitrarily large otherwise, depending on how the removal of an observation affects the posterior). We remove observations according to a normalized version of this metric, to ensure that the removed observations are indeed “stale” or, more broadly speaking, “irrelevant”.
>
> We also provide an example illustrating which observations are being removed. Animated visualizations are provided in the supplementary material, and are discussed in Appendix G.4.
>
> **Please add standard errors to the tables.**
>
> Due to lack of space, we did not report the standard errors directly in Table 2. However, they are provided graphically whenever applicable, namely in Figures 1-3 and 6-17.
>
> To further address this weakness while respecting the NeurIPS template, we propose to replicate Table 2 in a dedicated appendix of the camera-ready version, where we will also provide the standard errors.
>
> **There is a lot of important content that has been moved to the SM.**
>
> Upon acceptance, we will use the additional page allowed to move some important content from the SM to the main paper, and by possibly also reducing the Background section to make room for this material.
>
> **Typos.**
>
> Thank you for pointing out these typos. Upon acceptance, we will fix them in the camera-ready version.
>
> **References**
>
> [1] I. Bogunovic, J. Scarlett, and V. Cevher. Time-varying gaussian process bandit optimization. In AISTATS, pages 314–323. PMLR, 2016.
>
> [2] P. Brunzema, A. von Rohr, F. Solowjow, and S. Trimpe. Event-triggered time-varying bayesian optimization. arXiv preprint arXiv:2208.10790, 2022.
>
> [3] Nyikosa, F. M., Osborne, M. A., & Roberts, S. J. (2018). Bayesian optimization for dynamic problems. arXiv preprint arXiv:1803.03432.

---

> > ### Comment · Reviewer_iNzn · 2024-08-10
> > **Response to Rebuttal**
> >
> > Thanks to the authors for their response.
> >
> > __Benchmarking:__ Thanks to the authors for the clarification. Ultimately, while ARD is clearly the convention in BO (I cannot speak for DBO specifically, so I thank the authors for clarifying) the issue is ultimately one of fairness in evaluation. As mentioned previously, ARD kernels will be disadvantaged on symmetric functions. I was under the impression that ABO ran an ARD kernel (their paper seems to suggest it), but if the authors say that all methods run a non-ARD kernel, I will take their word for it.
> >
> >
> > __Evidence for removal:__ There has not been evidence that suggests that the removal criterion is actually efficient. One would expect there to be a comparison between not removing at all (should be ideal) and removing stale observations (should be close to ideal if the method is potent). As stated previously, regret performance as a measure of this is not convincing enough, in my view. The ablation plot on $\alpha$ is a start, but running the D-BO without removing any observations (and an otherwise identical setup to W-DBO) as a gold standard is, in my view, a must have for all experiments.
> >
> > __Wasserstein Distance figure:__ I think this is a good addition. I would also consider adding where the removed observation was located.
> > ______
> > Given unconvincing experimental evidence that the proposed method does what it seeks to do (namely remove unimportant observations), otherwise unconvincing experimental results, and a presentation that could use some polish, I will maintain my rating.

---

> > > ### Author Response · Authors · 2024-08-10
> > > **More Clarifications**
> > >
> > > Dear Reviewer iNzn,
> > >
> > > Thank you for your feedback on our response. We further address your points below.
> > >
> > > **Benchmarking**
> > >
> > > We agree with the reviewer that this discussion boils down to an alleged unfairness issue in the benchmarking. We think that this issue has been put to rest in three ways: (i) we have shown that most benchmarking in the *DBO* literature was conducted with a non-ARD kernel, (ii) we have made it clear that, for each experiment in the paper, each DBO solution uses a non-ARD kernel and (iii) we have run additional experiments with an ARD kernel to make sure that W-DBO was still the best-performing solution with ARD kernels.
> > >
> > > **Evidence for removal**
> > >
> > > An instance of W-DBO that does not remove any point would have a behavior very close to the behavior of TV-GP-UCB. In fact, when it does not remove any point, W-DBO is simply TV-GP-UCB with a different temporal kernel. This suggested instance of W-DBO would therefore be quite redundant with TV-GP-UCB, and would not exhibit superior performance for the same reason TV-GP-UCB does not.
> > >
> > > As time goes by, TV-GP-UCB and similar solutions that do not remove irrelevant observations (e.g., ABO and the instance of W-DBO suggested by the reviewer) experience a larger and larger GP inference time because irrelevant observations accumulate in their datasets. This increasingly large inference time prevents these solutions from querying the objective function as often as W-DBO, which regularly removes observations that are deemed irrelevant according to the definition in the paper (which all four reviewers find intuitive and reasonable). Because we benchmark the solutions in a continuous time setting, failing to query the objective function often enough ultimately hinders the optimization of the dynamic objective function by limiting how much two consecutive observations can be correlated. In other words, W-DBO outperforms TV-GP-UCB and ABO precisely because it removes some observations. At the same time, recall that W-DBO also outperforms R-GP-UCB and ET-GP-UCB, two solutions that also remove observations but in a less refined way.
> > >
> > > This leads us to make two conclusions: (i) a version of W-DBO that does not remove any irrelevant observation would **not** be the gold standard suggested by the reviewer and (ii) the very fact that W-DBO exhibits better performance than all the other DBO solutions (ABO and TV-GP-UCB on the one hand, R-GP-UCB and ET-GP-UCB on the other hand) is a strong evidence that the observations that it removes are indeed irrelevant, or at least, irrelevant enough so that removing them is a better strategy than keeping them in the dataset.
> > >
> > > Finally, we acknowledge that our paper opens multiple interesting research questions about observation relevancy, and does not provide all the answers to these questions. In our opinion, this is more a strength than a weakness. W-DBO is an original answer to the DBO problem that outperforms the existing solutions while opening various research questions, which is why we believe our paper is of interest to the BO community.
> > >
> > > We hope our answers have lifted any remaining misunderstanding.

---

> > > > ### Comment · Reviewer_iNzn · 2024-08-11
> > > > **Further comments**
> > > >
> > > > __Benchmarking:__ I'm inclined to agree. I'm surprised that the setup in DBO deviates so substantially from the rest of the field, but I digress.
> > > >
> > > > __Evidence for removal:__ Thanks to the authors for clarifying. Unfortunately, I had missed the fact that runtime (and not the number of iterations) was on the x-axis.  Since this metric is heavily impacted by the implementation quality of each algorithm and runtime-impacting hyperparameters such as acquisition optimization budget.
> > > >
> > > > If runtime and throughput is the only relevant metric, scalable methods such as TurBO, which naturally discards observations, should have been part of the initial evaluation. If sample-efficiency is still important, plots with #iterations in the X-axis need to be included, at least in the appendix.
> > > >
> > > > As I hope the authors see, there are still substantial confounding factors that makes it very difficult to assess if the removal itself is actually good. Regret over time is a downstream metric, and so that by itself is used to assess the quality of the point removal metric, there can be just about no confounding factors. Otherwise, experiments that assess the query removal _in isolation_ (e.g., predictive performance before and after removal compared to previous methods) should certainly be included.

---

> > > > > ### Author Response · Authors · 2024-08-12
> > > > > **Response to Further Comments (1/2)**
> > > > >
> > > > > Dear Reviewer iNzn,
> > > > >
> > > > > Thank you for your once again prompt feedback. We further address your points below.
> > > > >
> > > > > > This metric is heavily impacted by the implementation quality of each algorithm and runtime-impacting hyperparameters such as acquisition optimization budget
> > > > >
> > > > > Indeed, and that is why we have been very careful to implement all algorithms with the same and well-established BO library (namely BOTorch [1]), as stated in Appendix G.1. The runtime-impacting tasks (e.g., acquisition function optimization, hyperparameters optimization) are all implemented with the same methods (namely `botorch.optim.optimize_acqf` and gradient descent on marginal log-likelihood, respectively) and with the same budget. We can emphasize this more in Appendix G.1 of the camera-ready version.
> > > > >
> > > > > > Scalable methods such as TurBO [...] should have been part of the initial evaluation
> > > > >
> > > > > TurBO [2] is a trust region (TR)-based BO algorithm designed to optimize a static objective function. Its core idea is to maintain TRs on a static domain, centered on the best solutions found so far, and iteratively update them until they shrink towards the global maximum of the static objective function.
> > > > >
> > > > > TurBO sets aside observations but does not discard them in the same way as DBO algorithms (e.g., R-GP-UCB, ET-GP-UCB, W-DBO) do. Indeed, DBO algorithms remove a discarded observation from memory, whereas TurBO simply does not exploit it, but may exploit it at a later stage if one of its TRs is expanded or is reset. More importantly, maintaining TRs in a dynamic environment, which includes both a dynamic domain for the queries (queries in the past cannot be made anymore, queries in the future cannot be made yet) and a global maximum that is constantly moving, requires to develop a substantially different TR-based BO algorithm than TurBO.
> > > > >
> > > > > Other solutions that could be interesting in the dynamic setting include sparse GPs, as pointed out by Reviewer NVX3. Unfortunately, these solutions were designed to approximate exact GPs on a static objective function, and would also require non-trivial work to be adapted to a dynamic setting. As Reviewer NVX3 puts it, “the sparse GP methodology is relevant but cannot be applied in the current version of the papers and non-trivial work would need to be carried out”.
> > > > >
> > > > > Following Reviewer NVX3’s remark, we will mention and discuss sparse GPs in the camera-ready version. We will also add TR-based BO algorithms to this discussion.
> > > > >
> > > > > > Plots with # iterations in the X-axis need to be included
> > > > >
> > > > > The number of iterations of each algorithm, for each experiment, can be deduced from its average response time given on the x-axes of Figures 6-17: given an experiment duration $d$ and an average response time $r$, the number of iterations is $d / r$. Note that, as stated in Appendix G.1, $d = 600$ seconds in our experimental setting.
> > > > >
> > > > > In general, W-DBO has a lower response time (*i.e.*, it makes more queries in a given amount of time) than TV-GP-UCB and ABO, but has a larger response time (*i.e.*, it makes fewer queries in a given amount of time) than R-GP-UCB.
> > > > >
> > > > > Figure 2 shows the running average regret w.r.t. time for the Ackley experiment, and W-DBO’s average regret rapidly becomes significantly lower than the average regret of the other solutions. This value remains lower during the whole experiment, showing the sample efficiency of W-DBO.
> > > > >
> > > > > Although we are not sure if adding plots with the number of iterations on the x-axis would provide much more additional information, we can add them in a dedicated appendix of the camera-ready version.

---

> > > > > > ### Author Response · Authors · 2024-08-12
> > > > > > **Response to Further Comments (2/2)**
> > > > > >
> > > > > > > Regret is not enough to assess the quality of the point removal metric
> > > > > >
> > > > > > In our previous response, we misinterpreted the reviewer’s point about the removal metric. Allow us to address this concern with a more adequate answer.
> > > > > >
> > > > > > As far as we understand, the reviewer asks for evidence that the removed points are indeed irrelevant. In the paper, we clearly define the relevancy of an observation $\mathbf x_i$ as the Wasserstein distance between two GP posteriors, one conditioned on the whole dataset $\mathcal{D}$ and the other conditioned on $\tilde{\mathcal{D}} = \mathcal{D} \setminus \\{\mathbf x_i\\}$. All four reviewers agree that this definition is intuitive and reasonable.
> > > > > >
> > > > > > We have further illustrated this metric of relevancy in the rebuttal. In fact, in the PDF uploaded with the rebuttal, Figures 2 and 3 bring evidence that the metric is indeed aligned with our intuition. Concretely, it is low (respectively, large) when the two GP posteriors are similar (resp., dissimilar).
> > > > > >
> > > > > > A corollary to this definition is that observation relevancy is a continuous non-negative variable; it is zero when the GP posteriors are identical and can be arbitrarily large depending on how the removal of an observation changes the posterior. However, removing an observation from a dataset is a binary decision, since it boils down to labeling each observation as “kept in the dataset” or “removed from the dataset”. To do so, W-DBO (described in Algorithm 1) consumes a removal budget (discussed in Section 4 and Appendix F) by greedily removing the observation with the **smallest value of relevancy** in its dataset, according to the definition of relevancy introduced in the paper.
> > > > > >
> > > > > > That is why we were confused when the reviewer asked us for a proof that the observations removed by W-DBO were indeed irrelevant. Because the metric used by W-DBO follows directly from the very definition of observation relevancy, the observations removed by W-DBO are **by definition** the ones minimizing relevancy.
> > > > > >
> > > > > > We do not claim that the removal policy advertised in the paper (*i.e.*, the point removal metric, the removal budget and the greedy removal of observations) is the optimal policy to clean a dataset from unwanted observations. Our claim and main result is that a DBO algorithm that uses this policy (that is, W-DBO) outperforms, by a significant margin, state-of-the-art DBO algorithms that either propose to (i) not remove any observation (TV-GP-UCB), (ii) sample the objective function less frequently (ABO) or (iii) reset the dataset according to some condition (R-GP-UCB, ET-GP-UCB). This is discussed and demonstrated at length in Section 5 and in Appendix G.2 and G.3. Note that regret is then the correct metric to support this claim.
> > > > > >
> > > > > > > Assess the query removal in isolation (e.g., predictive performance before and after removal compared to previous methods)
> > > > > >
> > > > > > Such an assessment has been provided in the PDF uploaded with the rebuttal. Figures 2 and 3 show that our relevancy metric about an observation takes a small (respectively, large) value when the GP posteriors obtained before and after the observation is removed are similar (resp., dissimilar). Because the predictive performance is a distance function applied to the GP posterior and to the objective function, if the two GP posteriors are similar (i.e., the relevancy metric is low), then their predictive performance will be similar as well.
> > > > > >
> > > > > > We can make this more explicit by plotting the objective function when we will include Figures 2 and 3 in the camera-ready version.
> > > > > >
> > > > > > **In conclusion**
> > > > > >
> > > > > > We hope our answers have put the last concerns of the reviewer about potential confounding factors at rest. Indeed, on the one hand, the better performance of W-DBO cannot be explained by implementation quality. On the other hand, we clarified that the main goal of the paper is to introduce a new DBO algorithm that outperforms state-of-the-art DBO algorithms. No claim is made on the optimality of the relevancy metric nor on the observation removal policy, except that it is this metric that gives a decisive advantage to W-DBO.
> > > > > >
> > > > > > We remain available to answer further questions from the reviewer before the discussion period ends.
> > > > > >
> > > > > > **References**
> > > > > >
> > > > > > [1] Balandat, M., Karrer, B., Jiang, D., Daulton, S., Letham, B., Wilson, A. G., & Bakshy, E. (2020). BoTorch: A framework for efficient Monte-Carlo Bayesian optimization. Advances in neural information processing systems, 33, 21524-21538.
> > > > > >
> > > > > > [2] Eriksson, D., Pearce, M., Gardner, J., Turner, R. D., & Poloczek, M. (2019). Scalable global optimization via local Bayesian optimization. Advances in neural information processing systems, 32.

---

### Official Review · Reviewer_JgwK · 2024-07-10

**Soundness:** 3
**Presentation:** 3
**Contribution:** 3
**Rating:** 6
**Confidence:** 3

**Summary:**

The paper proposes a new algorithm for dynamic Bayesian optimization (DBO). To develop this new algorithm, the authors first derive a Wasserstein distance-based criterion that is a way of measuring how relevant a given collected data point is during optimization. Since dynamic functions change over time, each observation collected during DBO becomes less relevant over time, until eventually the computational cost of continuing to consider the observation outweighs any benefit it provides. When this becomes the case it therefore makes sense to remove these older observations from the dataset, especially since we care a lot about reducing computational cost since having high sampling frequency is especially important in DBO. In order to define a principled way to remove these irrelevant observations during DBO, the authors use their Wasserstein distance-based criterion to measure the relevancy of collected data points and decide whether to remove them from the dataset during the course of optimization. This strategy leads to the author’s novel DBO algorithm (W-DBO), which the authors show to perform better than relevant baselines with a convincing set of experimental results.

**Strengths:**

Originality: The author’s proposed Wasserstein distance-based criterion and resultand W-DBO algorithm is clearly a novel approach to solve a relevant problem in DBO.
Quality: The paper is very well-written. Additionally, the figures and tables are all of good quality - they are both easy to parse and do a nice job of displaying relevant results.
Clarity: The paper is clear and easy to follow from start to finish. The figures and tables are clear and easy to read. The paper is also clearly motivated and it’s easy to understand what the authors did and why.
Significance: It is obvious to me that the problem the authors seek to solve here (DBO algorithms keeping around increasingly irrelevant data points despite the need to maintain high sample efficiency over time) is important and relevant to the community. The author’s W-DBO algorithm provides a reasonable and intuitive solution to this problem.
Convincing results: The experimental results do indeed show that W-DBO outperforms other methods on a large number of BO tasks.

**Weaknesses:**

Typo (Line 305): “to the best of your knowledge” should instead be “to the best of our knowledge”.

One ablation I would’ve like to see in this paper: The paper makes it clear that leveraging quantification of relevancy of observations can be used to improve DBO performance with their W-DBO algorithm. However, it is less clear to me from the experimental results that using the author’s proposed Wasserstein distance-based criterion method for relevancy quantification is necessarily better than another strategy for quantifying relevancy. While the Wasserstein distance-based criterion method the authors propose is intuitive and clearly works well in this setting, it would be interesting to see a direct comparison against other simpler (or even ad-hoc) methods for attempting to quantify how relevant a given observation is. For example, rather than using the Wasserstein distance-based criterion, one could just assume some constant rate of decline in quantitative relevancy over time. I do not think that this additional experiment is necessary for this paper to be accepted, but I do think it would strengthen the paper by showing the importance of using the author's proposed Wasserstein distance-based criterion method specifically for relevancy quantification in this setting.

**Questions:**

The authors state that computational biology is a field that makes “heavy use of DBO”. I am curious what scenarios in comp bio require the use of DBO rather than just traditional BO? In what scenarios do we have biological black-box functions that are changing over time? It might be useful to have some additional citations and/or discussion of this in the paper.

**Limitations:**

Yes.

---

> ### Author Rebuttal · Authors · 2024-08-06
>
> Dear Reviewer JgwK,
>
> Thank you for the detailed review. We are discussing below the weaknesses and questions you have raised. Also, please make sure to read the global response as we discuss some of your questions there.
>
> **Direct comparison against other simpler (or even ad-hoc) methods for relevancy quantification (e.g., constant rate of decline over time).**
>
> TV-GP-UCB assumes a constant decreasing rate of the covariance function over time. Neglecting spatial relevancy, this precisely translates into a constant decreasing rate of the relevancy of an observation over time. Our approach (W-DBO) differs from TV-GP-UCB in three ways:
>
> (i) W-DBO can use an arbitrary time kernel instead of a fixed kernel for TV-GP-UCB (see [1]),
>
> (ii) W-DBO has a more sophisticated quantification of relevancy (*i.e.*, the Wasserstein distance) that involves the relevancy both in the time dimension (staleness) and in the spatial domain,
>
> (iii) W-DBO is able to remove observations when they are deemed irrelevant.
>
> Together, (i), (ii) and (iii) explain the difference in the performance of TV-GP-UCB and the performance of W-DBO.
>
> One could use the very same core idea of our framework (a distance function on stochastic processes) and come up with other definitions of relevancy. As an example, one could define the relevancy of an observation using the KL-divergence on the two GP posteriors. However, this would require a very different analysis to find an approximation of this distance, which would lead more to a new paper than to an ablation study.
>
> **The authors state that computational biology is a field that makes “heavy use of DBO”.**
>
> In Section 1, we cite computational biology as an application of BO, not of DBO. However, it is true that in our concluding remarks of Section 6, we cite computational biology as a field that could apply DBO. This is a typo that will be removed from the camera-ready version upon acceptance. Thank you for pointing it out.
>
> **References**
>
> [1] Ilija Bogunovic, Jonathan Scarlett, and Volkan Cevher. Time-varying gaussian process bandit optimization. In Artificial Intelligence and Statistics, pages 314–323. PMLR, 2016.

---

> > ### Comment · Reviewer_JgwK · 2024-08-12
> > **Rebuttal Acknowledgement**
> >
> > I would like to thank authors for their response and addressing the points I raised in my review. I am happy to keep my assessment of their work the same.

---

### Official Review · Reviewer_mGHg · 2024-07-11

**Soundness:** 3
**Presentation:** 3
**Contribution:** 3
**Rating:** 6
**Confidence:** 3

**Summary:**

This paper addresses the challenge of optimizing a time-varying black-box function using GP-based Dynamic Bayesian Optimization (DBO). Unlike traditional Bayesian Optimization (BO), DBO seeks to handle dynamic functions where the optimum changes over time by incorporating time in the GP model covariance function. With the goal of expediting DBO, this paper proposes a strategy to remove ‘irrelevant’ data from the GP model, specifically by removing points that minimally affect the distribution over future predictions. To this end, the authors introduce a Wasserstein distance-based criterion (and associated bounds/approximations) and propose the W-DBO algorithm. This algorithm dynamically removes irrelevant observations, maintaining good predictive performance and high sampling frequency. Numerical experiments demonstrate W-DBO's superiority over state-of-the-art methods.

**Strengths:**

- The introduction of a Wasserstein distance-based criterion to measure the relevancy of observations is an intuitive and interesting that effectively addresses the introduced challenge of minimally altering the GP predictions.
- The proposed W-DBO algorithm appears to perform well on the extensive numerical experiments, demonstrating the importance of removing irrelevant observations. The algorithm remains computationally efficient, avoiding the prohibitive growth of dataset size over time.
- The strategy is relatively generalizable; a Wasserstein distance-based approach can be integrated with any BO algorithm to identify candidates for removal from the dataset, enhancing applicability of this paper across various domains.

**Weaknesses:**

- The motivation for this challenge largely is based on GP-based Bayesian optimization, which might not always be applicable for this problem setting. Specifically, GP-based BO is most relevant when samples are expensive (and are limited as a result). When samples are cheap to obtain, derivative-free optimization methods such as evolutionary strategies are more relevant, but these are not compared.
- While the numerical experiments are extensive, they are limited to synthetic examples—most of which are not time-dependent (Appendix G.2). The inclusion of some real-world examples and or dynamic optimization case studies could strengthen the motivation and highlight applicability of the proposed method.
- The scalability of the W-DBO algorithm itself (e.g., checking the proposed metric for all points) in high-dimensional spaces or with very large datasets is not fully addressed, which could be a limitation in some applications.

**Questions:**

- Is there ever a downside with the ‘greedy’ of removal of points? E.g., removing the first selected point prevents otherwise being able to remove two points?
- Can the algorithm remove observations that become relevant again in the future? This is difficult to track, since $\mathcal{GP}_\mathcal{D}$ is updated with the removed sample in Algorithm 1, meaning there is no tracking of the original full model with no data removed.

**Limitations:**

Yes, discussed in appendices.

---

> ### Author Rebuttal · Authors · 2024-08-06
>
> Dear Reviewer mGHg,
>
> Thank you for the detailed review. We are discussing below the weaknesses and questions you have raised. Also, please make sure to read the global response as we discuss some of your questions there.
>
> **The scalability of W-DBO (e.g., in high-dimensional spaces and/or with large datasets) is not discussed.**
>
> The Wasserstein distance is applied in the output space, which is one-dimensional regardless of the dimensionality of the input space, *i.e.*, the objective function domain.
>
> Our framework, which measures the observations relevancy and removes the irrelevant observations, can be seen as a simple post-processing stage running at the end of each iteration. As a consequence, it can be used in conjunction with any BO algorithm. The W-DBO algorithm presented in this paper exploits vanilla GP-UCB, and as such, will struggle in high-dimensional input spaces, precisely because GP-UCB struggles in high-dimensional input spaces. However, our framework could be paired with any state-of-the-art high-dimensional BO algorithm (e.g., [1]) and show good performance when optimizing high-dimensional dynamic black-boxes. This is an interesting future work, thank you for suggesting it.
>
> Regarding very large dataset sizes, W-DBO suffers from the same limitations as any BO algorithm. This is because it also manipulates inverses of Gram matrices, which scale with the dataset size. That is precisely why we argue that removing irrelevant observations gives W-DBO a decisive advantage upon other DBO solutions. Nevertheless, upon acceptance, we will discuss this limitation explicitly in the Appendix H of the camera-ready version.
>
> **Is there ever a downside with the ‘greedy’ of removal of points?**
>
> Our framework could easily be extended to the power set $2^\mathcal{D}$ of the dataset $\mathcal{D}$, in order to measure the relevancy of a subset of observations, instead of only quantifying the relevancy of a single observation.
>
> The greedy approach causes W-DBO to overestimate the impact of the observation removals on the GP posterior. To rapidly get an intuition explaining this, let us describe a simple example. Consider a dataset that contains an irrelevant pair of observations $\mathcal{P} = \\{\mathbf x_i, \mathbf x_j\\}$, in the sense that $W_2\left(\mathcal{GP}\_\mathcal{D}, \mathcal{GP}\_{\mathcal{D} \setminus \mathcal{P}}\right) = 0$. Furthermore, let us assume that $W_2\left(\mathcal{GP}\_\mathcal{D}, \mathcal{GP}\_{\mathcal{D} \setminus \\{\mathbf x_i\\}}\right) = W_2\left(\mathcal{GP}\_{\mathcal{D} \setminus \\{\mathbf x_i\\}}, \mathcal{GP}\_{\mathcal{D} \setminus \mathcal{P}}\right) = \epsilon > 0$. If W-DBO were able to directly capture the irrelevancy of the couple $\mathcal{P}$, it would have removed it without consuming any of its removal budget. However, the greedy removal procedure as described in Algorithm 1 causes W-DBO to remove first $\mathbf x_i$ and next $\mathbf x_j$. Consequently, it will consume $2\epsilon$ from its removal budget and, in that sense, it will overestimate the impact of removing $\mathbf x_i$ and $\mathbf x_j$.
>
> Although working with the power set $2^\mathcal{D}$ of the dataset $\mathcal{D}$ is clearly more advantageous to avoid such budget depletion caused by greedy removals, the combinatorial explosion of $|2^\mathcal{D}|$ makes this strategy prohibitive in practice, even for moderately-sized datasets. That is why it is not advertised in the paper. Upon acceptance, we will include this remark in Section 4 of the camera-ready version. Thank you for pointing it out.
>
> **Can the algorithm remove observations that become relevant again in the future?**
>
> For usual stationary, decreasing kernels (e.g., Squared-Exponential or Matérn) and because of Assumption 3.2, the covariance between a function value at a time $t$, *i.e.*, $f(\mathbf x, t)$, and a function value at a future time $t’$, *i.e.*, $f(\mathbf x’, t’)$ with $t < t’$, can only decrease as $t’$ gets further and further away from $t$. Consequently, the relevancy of the observing $f(\mathbf x, t)$ will only decrease as time goes by.
>
> An interesting case that is not discussed in the paper, is when the time kernel $k_T$ is a periodic kernel, e.g., $k_T(t, t’) = \exp\left(-\frac{2 \sin^2\left(\pi |t - t’| / p\right)}{l^2}\right)$. The objective function is thus periodic along its time dimension. In that case, the concept of stale observation vanishes completely since the observation of $f(\mathbf x, t)$ will always be useful to predict $f(\mathbf x, t’)$, even when $t’ \gg t$. The Wasserstein distance between the two posterior GPs with this time-periodic kernel can be shown to diverge to $+\infty$, and so does our approximation, as it indeed should.
>
> **References**
>
> [1] Bardou, A., Thiran, P., & Begin, T. Relaxing the Additivity Constraints in Decentralized No-Regret High-Dimensional Bayesian Optimization. In The Twelfth International Conference on Learning Representations. 2024.

---

> ### Comment · Reviewer_mGHg · 2024-08-09
> **Response to Rebuttal**
>
> Thanks for the response. In particular, I see I have misunderstood how the experimental case studies were defined (see authors' global response), and so this stated weakness is largely alleviated. I have raised my score correspondingly, and look forward to the authors' stated clarifications in the camera-ready (or otherwise future) version.

---

### Author Rebuttal · Authors · 2024-08-06

Dear Reviewers,

We thank you all for your detailed reviews. In this global response, we address some weaknesses and questions that were raised in more than one review. We also discuss the additional figures shown in the PDF uploaded with the global rebuttal.

**This is a high-throughput problem, BO is not the ideal candidate for this setting.**

Most evaluations of DBO algorithms are performed in a setting that assumes a constant time step between two consecutive iterations (e.g., see [1, 2]). The underlying assumption is that the objective function is expensive to evaluate, making the GP inference time complexity negligible.

In Section 2, we argue that this is an unreasonable assumption to make for two reasons, regardless of how expensive an evaluation of the objective function is:

(i) Unlike static BO, the GP inference time in DBO limits how much two consecutive observations can be correlated. This in turn impacts the quality of the GP inference.

(ii) Considering optimization tasks with very large (e.g., infinite) time horizons, the GP inference time will eventually become more expensive than the evaluation of the objective function itself, if nothing is done to prevent the dataset size from diverging.

Now, if the objective function is expensive to evaluate, then a sample-efficient strategy should still be used, and BO is indeed the gold standard in this area. In other words, our problem setting still calls for sample-efficient optimization techniques (e.g., BO) because the objective function is expensive to evaluate. Our problem setting is also more general than the one considered in [1, 2], since it relaxes the assumption of a fixed time step between two consecutive iterations.

**The benchmarks are synthetic and not time-dependent.**

Among 12 benchmarks, 10 are synthetic while two are real-world experiments (WLAN and Temperature).

Although the dynamic nature of the two real-world experiments is quite explicit, we agree that we did not stress enough the fact that all synthetic examples are also time-dependent. Time is simply taken as the last dimension (*i.e.*, the $d$-th dimension) of each  $d$-dimensional synthetic function, while the first $(d-1)$ dimensions form the spatial domain. This explains why GP-UCB is not the best-performing strategy on the synthetic benchmarks.

This is stated in Appendix G.1, but not clearly enough in the main text. We will make this point clear in the camera-ready version, by mentioning at the beginning of Section 5 that the temporal domain is always the $d$th dimension of each $d$-dimensional objective function. To further avoid confusion, we will also rewrite Appendix G.2 to make the time variable $t$ explicitly appear in the definition of each synthetic function.

**Additional experiments with an ARD kernel**

In the PDF uploaded with the rebuttal, you can find additional experiments run with the anisotropic SE kernel (see Figure 1). The results are quite similar to the isotropic case, although the average response time and the average regret of each solution are larger. Both of these observations can be explained by the fact that each DBO solution has to infer more hyperparameters (more precisely, three spatial lengthscales instead of one) at each iteration.

We plan to add these results to a dedicated Appendix in the camera-ready version. Thank you for pointing out that it would be beneficial to extend our work to the anisotropic case.

**1D example motivating the normalization by an empty GP.**

A 1D example could indeed help motivating the normalization by an empty GP. You can find it in the PDF uploaded with the rebuttal. We discuss it below.

Figure 2 illustrates that the very same non-normalized Wasserstein distance ($W_2(\mathcal{GP}\_\mathcal{D}, \mathcal{GP}\_{\tilde{\mathcal{D}}}) = 0.46$ in both cases) can lead to different results on the posteriors, depending on their covariance parameters (here, only the lengthscale is considered). The left part of Figure 2 depicts two widely different posteriors, while the right part of Figure 2 depicts two similar posteriors. A good metric of relevancy should capture this difference.

Figure 3 depicts four different scenarios illustrating that the normalized Wasserstein distance is able to capture this difference. In fact, when the metric is low (*i.e.*, close to $0$), it always implies that the two posteriors are similar, regardless of the covariance hyperparameters (*i.e.*, the lengthscale in this example). Similarly, when the metric is large (*i.e.*, close to $1$), the two posteriors are very different, regardless of the covariance hyperparameters.

Upon acceptance, we plan to add this example to Appendix E of the camera-ready version. Thank you for pointing out that a 1-dimensional example would be beneficial.

**References**

[1] Ilija Bogunovic, Jonathan Scarlett, and Volkan Cevher. Time-varying gaussian process bandit optimization. In Artificial Intelligence and Statistics, pages 314–323. PMLR, 2016.

[2] Paul Brunzema, Alexander von Rohr, Friedrich Solowjow, and Sebastian Trimpe. Event-triggered time-varying bayesian optimization. arXiv preprint arXiv:2208.10790, 2022.

---

### Decision · Program_Chairs · 2024-09-25

**Decision:**

Accept (poster)

**Comment:**

All except one of the reviewers are positive about this work. The reviewer with a negative rating acknowledges a number of the work's strengths, such as solving a relevant problem, proposing an intuitive solution, and having good theory, but opted for a very low rating in spite of this due to concerns with benchmarking.

After fairly extensive discussion with other reviewers, no-one else found these concerns sufficiently significant to change their scores, and the paper's other reviewers have all opted in favor of acceptance. Between the two camps, I am inclined to follow the majority's recommendation.